Manuscript prepared for Biogeosciences
with the LATEX class copernicus.cls.
Date: 24 January 2017

# Manganese in the West Atlantic Ocean in context of the first global ocean circulation model of manganese

Marco van Hulten[1,6], Rob Middag[2,3,4], Jean-Claude Dutay[1], Hein de Baar[4,5], Matthieu Roy-Barman[1], Marion Gehlen[1], Alessandro Tagliabue[7], and Andreas Sterl[6]

[1]Laboratoire des Sciences du Climat et de l'Environnement (LSCE), IPSL, CEA–Orme des Merisiers, 91191 Gif-sur-Yvette, France
[2]University of Otago, 364 Leith Walk, Dunedin, 9016, New Zealand
[3]University of California Santa Cruz (UCSC), 1156 High Street, Santa Cruz, CA 95064, USA
[4]NIOZ Royal Netherlands Institute for Sea Research, Department of Ocean Systems, and Utrecht University, P.O. Box 59, 1790 AB Den Burg, Texel, the Netherlands
[5]University of Groningen (RUG), Postbus 72, 9700 AB Groningen, the Netherlands
[6]Royal Netherlands Meteorological Institute (KNMI), Utrechtseweg 297, 3731 GA De Bilt, the Netherlands
[7]University of Liverpool, 4 Brownlow Street, Liverpool L69 3GP, UK

*Correspondence to:* M. M. P. van Hulten (mvhulten@lsce.ipsl.fr)*

**Abstract.** Dissolved manganese (Mn) is a biologically essential element. Moreover, its oxidised form is involved in removing itself and several other trace elements from ocean waters. Here we report the longest thus far 17 500 km length full-depth ocean section of dissolved Mn in the West Atlantic Ocean, comprising 1320 data values of high accuracy. This is the GA02 transect that is part of the GEOTRACES programme, which aims to understand trace element distributions. The goal of this study is

5   to combine these new observations with a new, state-of-the-art, modelling to give a first assessment of the main sources and redistribution of Mn throughout the ocean. To this end, we simulate the distribution of dissolved Mn using a global-scale circulation model. This first model includes simple parameterisations to account for the sources, processes and sinks of Mn in the ocean. Oxidation and (photo)reduction, aggregation, settling, as well as biological uptake and remineralisation by plankton, are included in the model. Our model provides, together with the observations, the following insights:

10   – The high surface concentrations of manganese are caused by the combination of photoreduction and sources to the upper ocean. The most important sources are sediments, dust, and, more locally, rivers.

– Observations and model simulations suggest that surface Mn in the Atlantic Ocean moves downwards into the southward flowing North Atlantic Deep Water (NADW), but because of strong removal rates there is no elevated concentration of Mn visible any more in the NADW south of 40° N.

---

*Postprints may be found at arXiv:1606.07128.

- ~~The biological cycle is not well understood: when incorporating Mn in plankton, similarly to phosphorus, Mn becomes extremely low in large parts of the surface of the Pacific Ocean. This is inconsistent with observed dissolved Mn concentrations in these Pacific surface waters.~~

- The complete model predicts lower dissolved Mn in surface waters of the Pacific Ocean than the observed concentrations. The intense Oxygen Minimum Zone (OMZ) in subsurface waters is deemed to be a major source of dissolved Mn also mixing upwards into surface waters, but the OMZ is not well represented by the model. Improved high resolution simulation of the OMZ may solve this problem.

- There is a mainly homogeneous background concentration of dissolved Mn of about $0.10$ nM to $0.15$ nM throughout most of the deep ocean. The model reproduces this by means of a threshold on particulate manganese oxides of $25$ pM, suggesting that a minimal concentration of particulate Mn is needed before aggregation and removal become efficient.

- The observed distinct hydrothermal signals are produced by assuming both a strong source and a strong removal of Mn near hydrothermal vents.

## 1 Introduction

Dissolved manganese ($Mn_{diss}$) is taken up by phytoplankton, because Mn is crucial for photosynthesis and other biological functions (Raven, 1990). Furthermore, its oxidised form ($Mn_{ox}$) plays an important role in the removal of several other trace metals from seawater (Yamagata and Iwashima, 1963). While in the open ocean manganese (Mn) exists in small concentrations, it is the twelfth most plentiful element in the Earth's crust (Wedepohl, 1995). In seawater, Mn occurs in many forms, among which the bioavailable dissolved form. After phytoplankton death, incorporated Mn sinks downwards together with the dead material, but most of the organic material is remineralised before reaching the sea floor (Froelich et al., 1979), releasing Mn back to the water.

Another important mechanism of storing Mn in particles, besides biological incorporation, is the removal of dissolved Mn via larger colloids on which the Mn is oxidised to insoluble Mn(IV) (and possibly the other Mn(III) oxidation state), and the subsequent aggregation by particulate matter. Oxidation occurs everywhere in the ocean where oxygen is available. This process can be strongly accelerated by Mn(II)-oxidising microorganisms, primarily bacteria and fungi (Sunda and Huntsman, 1988, 1994; Tebo et al., 2005), but we do not understand the role of these organisms quite well (Nealson, 2006). The reverse process is the reduction of Mn oxides to bioavailable dissolved Mn(II), i.e. $Mn^{2+}(aq)$. The full oxidation/reduction (redox) equilibrium reaction, in its most simple form, is given by (Froelich et al., 1979):

$$Mn^{2+} + \frac{1}{2}O_2 + H_2O \quad \leftrightharpoons \quad MnO_2 + 2H^+. \tag{R1}$$

Reduction is significantly faster under the influence of sunlight. It is hence referred to as *photoreduction* when irradiance is the major contributor. The relative rate of reduction compared to oxidation is important for Mn(II) availability. An overall net higher oxidation rate implies more particle formation, hence more Mn export.

Manganese enters the open ocean through lithogenic dust deposition (Baker et al., 2016) and lateral advection from reducing sediments (Homoky et al., 2016). Sediments along the relatively shallow (<1000 m depth) ocean margins tend to receive more organic deposition, hence by enhanced microbial decomposition are more anoxic and a stronger source of reduced Mn than deep sea sediments. Surface $[\mathrm{Mn_{diss}}]$ is especially high in the central Atlantic Ocean and up to at least $30°$ N because of high dust input from the Sahara in combination with photoreduction (Landing and Bruland, 1987; Jickells, 1995; Guieu et al., 1994; Baker et al., 2006; De Jong et al., 2007; Wu et al., 2014). Similarly, $[\mathrm{Mn_{diss}}]$ is high in the northern Indian Ocean (Vu and Sohrin, 2013). Dissolved Mn diffuses out of oxygen deprived sediments, because sediment microorganisms reduce $\mathrm{Mn_{ox}}$ if there is no more oxygen or nitrate left (Li et al., 1969; Landing and Bruland, 1980; Sundby and Silverberg, 1985; Middag et al., 2012). As long as there is oxygen in the sediment, the organic carbon is remineralised by using this oxygen as an electron acceptor (Froelich et al., 1979).

Rivers are another source of Mn to the ocean (Elderfield, 1976; Aguilar-Islas and Bruland, 2006). While much of the fluvial Mn is removed within the estuaries into the sediments by scavenging and aggregation, a large part may finally be transported to the ocean by diffusion from and resuspension of the sediments (Jeandel, 2016; Charette et al., 2016). Typically, the smaller sediment particles ($0.5\,\mathrm{\mu m}$ to $4\,\mathrm{\mu m}$) have a high Mn content, and, because of their small size, are able to reach the open ocean (Yeats et al., 1979; Sundby et al., 1981; Trefry and Presley, 1982). Manganese may also flux into the ocean by melting sea ice (Middag et al., 2011b). Finally, overwhelming evidence is found of manganese fluxing out of hydrothermal vents (Klinkhammer et al., 1977, 1985; Hydes et al., 1986; Klinkhammer et al., 2001; Middag et al., 2011a, b; German et al., 2016).

Downward fluxes of settling particles that have been collected in sediment traps show a strong correlation between lithogenic particles and authigenic Mn (Roy-Barman et al., 2005). Therefore, lithogenic particles are likely to play a significant role in the removal (oxidation, scavenging and aggregation) of Mn (Dutay et al., 2015). Here the ballast effect of lithogenic particles, which typically have a density of about twice of that of seawater, is likely playing a major role for rapid settling of agglomerates of biogenic and lithogenic particles including Mn-oxides coatings (Balistrieri et al., 1981). This is consistent with the fact that most suspended particles are small (less than $2\,\mathrm{\mu m}$), but the larger aggregates are deemed to be the significant contributors to the vertical flux (McCave, 1975; Bishop and Fleisher, 1987). The complete process may be more complicated than described above (Boyle et al., 2005), e.g. because Mn binds to dissolved ligands such that more of it may stay in solution (Sander and Koschinsky, 2011; Madison et al., 2013; Luther et al., 2015). Manganese oxides are important scavengers of other trace metals like iron, cobalt, nickel and zinc (Yamagata and Iwashima, 1963; Murray, 1975; Means et al., 1978; Saito et al., 2004; Tonkin et al., 2004), as well as insoluble radionuclides such as thorium and protactinium (e.g Reid et al., 1979; Hayes et al., 2015; Jeandel et al., 2015). Therefore, Mn availability does not only directly impact primary production but may also play a role in removing other elements from the surface ocean. These elements include biologically essential trace metals as well as many more trace elements.

Published observational studies, like the results of Wu et al. (2014), show low, relatively constant $\mathrm{Mn_{diss}}$ concentrations of around $0.15\,\mathrm{nM}$ away from the boundaries. Towards the surface there is a sharp increase of $[\mathrm{Mn_{diss}}]$. Other notable elevations are near oceanic ridges. These elevations of $[\mathrm{Mn_{diss}}]$ at mid-depths can be ascribed to hydrothermal activity near those regions.

The hydrothermal plumes typically extend in the order of $1000\,km$ (Middag et al., 2011a, b), or reach even up to $3000\,km$ (Resing et al., 2015).

What makes the distribution of $Mn_{diss}$ relatively homogeneous in the interior of the ocean? How can this be reconciled with localised features where $[Mn_{diss}]$ is very elevated compared to that stable "background concentration"? Until now only local model simulations of the Mn ocean cycle have been performed that focus on the processes most relevant for the respective regions. In other words, the different features of the $Mn_{diss}$ distribution have not been brought together in a unified model.

The case of hydrothermal activity has been studied by Lavelle et al. (1992) who modelled Mn in the deep ocean near hydrothermal vents. They included four Mn tracers, namely, a dissolved form, small particles associated with bacteria, larger aggregate particles, and one in sediments as the model includes benthic fluxes. They found that "more than $80\,\%$ of the hydrothermal Mn is deposited within several hundred kilometres of the ridge crest though dissolved Mn concentrations beyond that distance exceed background levels by many times". This illustrates a high Mn input and removal from hydrothermal vents, as well as the already mentioned large plume extend.

The $Mn_{diss}$ distribution in the North Pacific Ocean has been modelled by Johnson et al. (1996), in a 1-D vertical model neglecting horizontal transport. Their oxidation model depends on the oxygen concentration and the hydroxide activity. Their region of interest was on the upper and intermediate depth ocean, and their goal was to reproduce the $[Mn_{diss}]$ maximum in the Oxygen Minimum Zone (OMZ). Besides the North Pacific, the OMZ is also present in other basins, amongst which the northwestern Indian Ocean (Saager et al., 1989; Lewis and Luther, 2000). Johnson et al. (1996) found that the combination of remineralisation rates and decreased oxidation in the OMZ explained the $[Mn_{diss}]$ subsurface maximum in their profiles, whereas a flux from the continental margin sediments appeared not responsible. In the euphotic zone $Mn_{diss}$ was incorporated in phytoplankton, while remineralised $Mn_{diss}$ in the aphotic (and disphotic) zone was lost by oxidation and scavenging.

While these modelling efforts are useful for their purposes, no studies exist in which the global ocean $Mn_{diss}$ distribution is modelled. To arrive at a more integrated understanding of the behaviour of pelagic Mn, we include a Mn model in a global ocean general circulation model. Specifically, we will test a simple mechanism that should be able to give insight into the apparently stable $Mn_{diss}$ concentration, and its contrast to strong Mn sources. This is the first time that a global ocean model for manganese has been written and assessed. It is a basic model that should give a starting point for further studies. At the moment there is no mechanistic evidence for typical uptake-remineralisation processes as is the case for, e.g., iron. However, as there is of course uptake of Mn by phytoplankton, and sometimes it can even be a limiting nutrient (Middag et al., 2013), we include a biological cycle of Mn in the model. While we perform a model simulation on a global scale, we give more attention to the $Mn_{diss}$ distribution in the Atlantic Ocean. We will compare our simulations in most detail, and quantitatively, with a here reported highly accurate dataset from the GEOTRACES programme, namely the GA02 section in the West Atlantic Ocean. Here *accuracy* refers to the proximity of the measurements to the true values, as confirmed by (i) the compliance of reference samples with international consensus values (Table 1), and (ii) the agreement at the crossover station (Fig. 1) with the US section GA03, and (iii) the agreement of the two independent methods FIA and ICPMS for the our data of section GA02 (Fig. 2). In addition, the North Atlantic GA03 (Wu et al., 2014) and the Zero Meridian Southern Ocean GIPY5 (Middag et al., 2011a) transects will be used for further detailed visual comparison. Furthermore, in those regions we can study important

properties of the ocean geochemistry of manganese and the interaction with circulation, among which the Atlantic overturning circulation and hydrothermal activity.

In this study our goal is to assess the fundamental processes that are the most important to veraciously simulate $[\mathrm{Mn_{diss}}]$, including the aforementioned properties of the dissolved Mn distribution. Here *veracity* refers to the proximity of the model to the observations. To this end, we will first introduce our Mn model with its processes, sources and sinks. We will show the results of a reference simulation, which will be compared with recent high-accuracy observations. Also several sensitivity simulations will be presented, studying the effects of the biological cycle, the intense nature of hydrothermal vents and the strong removal of manganese from the seawater.

## 2 Methods

### 2.1 Observations in the West Atlantic Ocean

#### 2.1.1 Sample collection

For the determination of trace metal concentrations, samples were collected along the GEOTRACES Atlantic Meridional GA02 transect of the Netherlands (Fig. 5). Sampling was done with an all-titanium ultraclean CTD sampling system for trace metals (De Baar et al., 2008) with novel PVDF samplers (Rijkenberg et al., 2015; Middag et al., 2015a, b). Immediately upon recovery, the complete titanium frame with its 24 PVDF samplers was placed inside a clean room environment where the sub-samples for trace metal analysis were collected. The water was filtered from the PVDF samplers over a $0.2\,\mu\mathrm{m}$ filter cartridge (Sartobran-300, Sartorius) under pressure (1.5 atm) of (inline prefiltered) nitrogen gas. Sub-samples for dissolved metals were taken in cleaned (Middag et al., 2009, for cleaning procedure) LDPE sample bottles. All sample bottles were rinsed five times with the sample seawater. Seawater samples were acidified with HCl to a concentration of 0.024 M HCl which results in a pH of 1.7 to 1.8 with Baseline® Hydrochloric Acid (Seastar Chemicals Inc.).

#### 2.1.2 Analysis of dissolved Mn

Analyses of dissolved manganese were performed shipboard with the method developed by Doi et al. (2004) with some slight modifications in the preparation and brands of the chemicals used. Notably, samples were buffered in-line with an ammonium borate sample buffer to a pH of $8.5\pm0.2$. This buffer was produced by dissolving 30.9 g of boric acid (Suprapure, Merck) in 1 L MQ water (Millipore Milli-Q) deionised water $R>18.2\,\mathrm{M\Omega\,cm^{-1}}$ and adjusting the pH to 9.4 with ammonium hydroxide (Suprapure, Merck).

The buffered sample was pre-concentrated during 150 s on a Toyopearl AF-Chelate 650M (TosoHaas, Germany) column. Hereafter the column was rinsed for 60 s with MQ water to remove interfering salts. The Mn was subsequently eluted from the column for 200 seconds with a solution of 0.1 M three times quartz distilled formic acid (reagent grade, Merck) containing 0.1 M hydrogen peroxide (Suprapure, Merck) and 12 mM ammonium hydroxide (Suprapure, Merck). The pH of this carrier solution was adjusted to $2.9\pm0.05$. The eluate with the dissolved Mn passed a second column of immobilised 8-hydroxyquinoline

| $Mn_{diss}$ (nmol kg$^{-1}$) | Consensus | Middag et al. (2015b) |
|---|---|---|
| SAFe D2 | $0.35 \pm 0.05$ | $0.33 \pm 0.01$ ($n = 24$) |
| GEOTRACES S | $1.46 \pm 0.14$ | $1.47 \pm 0.03$ ($n = 10$) |
| GEOTRACES D | $0.21 \pm 0.03$ | $0.18 \pm 0.01$ ($n = 5$) |

**Table 1.** Compliance with the international GEOTRACES reference samples program. Left column are the international consensus values. Right column are the values reported here as part of the GA02 dataset.

(Landing et al., 1986) to remove interfering iron ions in the carrier solution (Doi et al., 2004). Hereafter the carrier mixed with 0.7 M ammonium hydroxide (Suprapure Merck) and a luminol solution. The latter luminol solution was made by diluting 600 µl luminol stock solution and 10 µl TETA (triethylenetetramine, Merck) in 1 dm$^3$ MQ. The luminol stock solution was made by diluting 270 mg luminol (3-aminophtalhydrazide, Aldrich) and 500 mg potassium carbonate in 15 ml MQ. The resulting mixture of carrier solution, ammonium hydroxide and luminol solution had a pH of $10.2 \pm 0.05$ and entered a 3 m length mixing coil placed in a water bath of 25 °C. Hereafter the chemiluminescence was detected with a Hamamatsu HC135 Photon counter. Concentrations of dissolved Mn were calculated in nanomole per litre (nM) from the photon emission peak height of triplicate measurements.

The system was calibrated using standard additions from a 5000 nM Mn stock solution (Fluka) to filtered acidified seawater of low Mn concentration that was collected in the sampling region. A five-point calibration line (0, 0.1, 0.2, 0.6 and 1.2 nM standard additions) and blank determination were made every day. The three lowest points (0, 0.1 and 0.2 nM) of the calibration line were measured in triplicate and the two highest points (0.6 and 1.2 nM) in duplicate in order to add more weight to the lower part of the calibration line. The blank was determined by measuring acidified MQ which was below the detection limit and subsequently no blank was substracted. The limit of detection defined as three times the standard deviation of the lowest value observed was $< 0.01$ nM. The flow injection system was rinsed every day with a 0.5 M HCl solution.

An internal reference sample was measured in triplicate every day. This was a sub-sample of a 25 dm$^3$ volume of filtered seawater that was taken at the beginning of Leg 1 (also used during Leg 2, i.e. Iceland to the equator) and Leg 3 (Punta Arenas, Chile, to the equator). The relative standard deviation (i.e. the precision) of this replicate analysis seawater sample that was analysed 40 times on different days in triplicate was 2.57 % (Leg 1 and 2) and 1.21 % for 17 analyses during Leg 3. The relative standard deviation on single days was on average 1.37 % and the absolute values were 0.45 and 0.61 nM for the first two legs and Leg 3, respectively.

As external comparison, the international reference samples collected on the GEOTRACES Intercalibration Cruise (www. geotraces.org) as well as from the SAFe cruise (Johnson et al., 2007) were analysed for Mn. At different locations, large volumes of seawater were sampled of which subsamples were analysed by various laboratories, resulting in different reference values. These labs include the Royal Netherlands Institute for Sea Research (NIOZ), who also analysed the GA02 transect data. An independent referee removed outliers and averaged the reference values, resulting for each of the sample locations in a consensus value, which is here considered as the true value. These consensus values are listed in Table 1, together with

the values determined by NIOZ. The distributions of the measurements from NIOZ lie within the precision of the consensus values.

The vertical profiles of the BATS station for the Netherlands sampling at 13 June 2010 and the US sampling at 19 November 2011 are presented in Fig. 1. For both occupations the samples were analysed by ICP-MS by Rob Middag at UCSC. Moreover, the same analyst had done shipboard FIA analyses during the 2010 cruise. There was no statistical difference within analytical uncertainty between the three datasets in the deep waters (Middag et al., 2015b). Small differences within the upper ∼500 m between the Netherlands sampling and the US sampling are deemed to be real due to the seasonal variability over the more than one year time difference of sampling. Furthermore, these Mn profiles are also consistent with the profiles determined by Landing et al. (1995).

Besides the shipboard Flow Injection Analysis (FIA), we also analysed samples using mass spectrometer analysis. Figure 2 shows the correlation between the two methods of analysis for the determination of $[\text{Mn}_{\text{diss}}]$, shipboard and laboratory measurements. There is a very good agreement between the shipboard and mass spectrometer analyses, which strongly suggests a high observational accuracy. For the GIPY5 transect in the Southern Ocean, we analysed $[\text{Mn}_{\text{diss}}]$ only through flow injection (FIA). Since we want to plot GA02 together with the GIPY5 in this study, we use the observational data obtained from the shipboard FIA for the comparison with the model simulations.

## 2.2 Model description

In order to simulate the three-dimensional (3-D) distribution of dissolved Mn, we use the general circulation model *Océan PArallélisé* (OPA) that is part of NEMO, a framework for ocean models (Madec and the NEMO team, 2016). We use the ORCA2-LIM configuration of NEMO. The spatial resolution is 2° by 2° $\cos(\phi)$ (where $\phi$ is the latitude) with an increased meridional resolution to 0.5° in the equatorial domain (Madec and Imbard, 1996). The model has 30 vertical layers, with an increased vertical thickness from 10 m at the surface to 500 m at 5000 m depth. Representation of the topography is based on the partial-step thickness (Barnier et al., 2006). Lateral mixing along isopycnal surfaces is performed both on tracers and momentum (Lengaigne et al., 2003). The parameterisation of Gent and McWilliams (1990) is applied from 10° poleward to represent the effects of non-resolved mesoscale eddies. Vertical mixing is modelled using the turbulent kinetic energy (TKE) scheme of Gaspar et al. (1990), as modified by Madec and the NEMO team (2016). The fluid dynamics used to drive our model is identical to that used in Aumont et al. (2015).

The model contains two tracers of Mn, referred to as dissolved ($\text{Mn}_{\text{diss}}$) and oxidised ($\text{Mn}_{\text{ox}}$) manganese. These tracers are driven by the equations set out in this section, as well as the velocity fields obtained from OPA. Instead of calculating the dynamical variables of the model (velocity and mixing), we run it off-line, using a climatology with a resolution of five days.

Figure 3 presents the conceptual scheme of our manganese model. The internal processes include biological uptake and remineralisation (green arrows), reduction, oxidation, aggregation and burial (blue arrows in the figure), and are described in the following subsections. Manganese sources from rivers, the atmosphere, sediments and hydrothermal vents are presented as arrows at the top and bottom of the figure. These four Mn sources are presented in Fig. 4, and Table 2 lists the absolute contributions to the different basins by each of these sources, as well as the relative contribution of every source to the world

| Basin | Dust | Rivers | Sediment | Hydrothermal |
|---|---|---|---|---|
| Atlantic Ocean | 2200 | 127 | 924 | 13917 |
| Pacific Ocean | 1183 | 94 | 1237 | 59846 |
| Indian Ocean | 1506 | 25 | 442 | 18653 |
| Southern Ocean | 14 | 0 | 206 | 7601 |
| Arctic Ocean | 23 | 19 | 463 | 2269 |
| Mediterranean Sea | 673 | 12 | 91 | 0 |
| total amount (Mmol yr$^{-1}$) | 5598 | 277 | 3363 | 102286 |
| relative amount (%) | 5.0 | 0.2 | 3.0 | 91.7 |

**Table 2.** Absolute amount of effective annual input by an imposed Mn flux from sediments, rivers and hydrothermal vents into each basin (Mmol yr$^{-1}$). The Southern Ocean is defined as the ocean south of 58.7° S. The line "total amount" denotes how much world-wide Mn is added to the ocean due to a specific flux; "relative amount" is normalised to the total Mn input flux.

ocean. The model parameters are summarised in Table 3. In the following subsections we will describe how the different sources and processes are included in the model.

### 2.2.1 Atmospheric source

Manganese is added to the pool of $Mn_{diss}$ in the upper model layer, according to

$$\left. \frac{\partial \mathcal{D}}{\partial t} \right|_{\text{dust}} = \frac{\alpha \cdot f_{\text{Mn,dust}}}{m \cdot \Delta z_1} \cdot \Phi_{\text{dust}}, \tag{1}$$

where $\mathcal{D}$ is the dissolved Mn concentration, $\alpha$ is the solubility of Mn in dust, $f_{\text{Mn,dust}}$ is the mass fraction of Mn in dust, $m$ is the molar mass of Mn, and $\Delta z_1 = 10\,\text{m}$ is the upper model layer thickness. The lithogenic dust deposition flux, $\Phi_{\text{dust}}$, is derived from the Interaction with Chemistry and Aerosols (INCA) model (Hauglustaine et al., 2004). Here we use a 12 month climatology of INCA's output as a forcing.

The average mass fraction of Mn in the Earth's upper crust is 527 ppm (Wedepohl, 1995). However, the fraction measured in Saharan dust is 880 ppm (Mendez et al., 2010), consistent with Guieu et al. (1994) and Statham et al. (1998). Since most of the dust deposited on the Atlantic Ocean originates from the Sahara and our focus is the Atlantic Ocean, the value of 880 ppm is used for $f_{\text{Mn,dust}}$.

The solubility of Mn from dust is uncertain and relatively high compared to most other trace metals. Here $\alpha = 40\,\%$ of the Mn in dust is assumed to dissolve, largely consistent with the values reported by Guieu et al. (1994); Jickells (1995); Baker et al. (2006); De Jong et al. (2007); Buck et al. (2010). Several studies report even higher values ($> 50\,\%$), which are, however, mainly from anthropogenic or otherwise processed dust, while the lower reported values ($< 50\,\%$) are from natural dust, mainly of Saharan origin.

Figure 4a presents the average Mn dissolution flux of the 12 months climatology. Globally this is $5.6\,\text{Gmol yr}^{-1}$ of Mn.

| Parameter | Symbol | Value used | Known range | Reference |
|---|---|---|---|---|
| Mass fraction of Mn in dust | $f_{\mathrm{Mn,dust}}$ | 880 ppm | 696 to 880 ppm | Wedepohl (1995); Mendez et al. (2010) |
| Dust Mn solubility | $\alpha$ | 40 % | 10 to 70 % | Baker et al. (2006) |
| Sediment source Mn/Fe ratio | $r_{\mathrm{Mn:Fe,sed}}$ | 0.2 | uncertain | Bortleson and Lee (1974); Slomp et al. (1997) |
| River source Mn/Fe ratio | $r_{\mathrm{Mn:Fe,riv}}$ | 0.214 | uncertain | Sarmiento and Gruber (2006) |
| Hydrothermal Mn/$^3$He ratio | $r_{\mathrm{Mn:^3He,hydro}}$ | $0.10 \times 10^9$ | uncertain | - |
| Settling speed of $\mathrm{Mn_{ox}}$ | $w_{\mathrm{ox}}$ | $1\text{--}10 \, \mathrm{m \, d^{-1}}$ | $0.9\text{--}1.4 \, \mathrm{m \, d^{-1}}$ | Roy-Barman (2009) |
| Oxidation rate constant | $k_{\mathrm{ox}}$ | $0.341 \times 10^{-3} \, \mathrm{h^{-1}}$ | uncertain | Bruland et al. (1994); Sunda and Huntsman (1994) |
| Photoreduction rate constant | $k_{\mathrm{red,light}}$ | $98 \times 10^{-3} \, \mathrm{h^{-1}}$ | $50\text{--}150 \times 10^{-3} \, \mathrm{h^{-1}}$ | Sunda and Huntsman (1994) |
| Aphotic reduction rate | $k_{\mathrm{red,dark}}$ | $1.70 \times 10^{-3} \, \mathrm{h^{-1}}$ | $0.98\text{--}14.3 \times 10^{-3} \, \mathrm{h^{-1}}$ | Bruland et al. (1994); Sunda and Huntsman (1994) |
| Aggregation threshold | $\mathcal{X}_{\mathrm{thr}}$ | 25 pM | hypothetical | - |
| Mn/P incorporation ratio | $R_{\mathrm{Mn:P}}$ | $0.36 \times 10^{-3}$ | $0.2 \times 10^{-3}$ to $1.5 \times 10^{-3}$ | Middag et al. (2011a); Twining and Baines (2013) |

**Table 3.** Mn model parameters for the *Reference* simulation. The settling velocity $w_{\mathrm{ox}}$ is given by Eq. 11. In *LowHydro* the hydrothermal flux and the maximum settling speed are both reduced by a factor of 10 ($r_{\mathrm{Mn:^3He,hydro}} = 0.01 \times 10^9$; and $w_{\mathrm{ox}} \equiv 1 \, \mathrm{m \, d^{-1}}$ or, equivalently, $\mathcal{X}_{\mathrm{thr}} \to \infty$). In the sensitivity simulation *NoThreshold* the aggregation threshold $\mathcal{X}_{\mathrm{thr}}$ is set to zero.

### 2.2.2 River source

25 The manganese river source is modelled analogously to iron, which is part of the biogeochemical model PISCES (Aumont et al., 2015). This means that our manganese influx is proportional to the total dissolved (organic and inorganic) carbon flux, just like for iron (Aumont et al., 2015). Hence, the modelled concentration change of $\mathrm{Mn_{diss}}$ caused by river input is given by

$$\left. \frac{\partial \mathcal{D}}{\partial t} \right|_{\mathrm{rivers}} = r_{\mathrm{Mn:Fe,river}} \cdot \left. \frac{\partial [\mathrm{Fe_{diss}}]}{\partial t} \right|_{\mathrm{rivers}}, \tag{2}$$

where $[\mathrm{Fe_{diss}}]$ is the dissolved iron concentration, and $r_{\mathrm{Mn:Fe,river}}$ is the effective manganese/iron flux ratio, here set to 0.214, based on river dissolved concentrations (Sarmiento and Gruber, 2006, p. 2). For comparison, the Mn/Fe flux ratio is much higher than the crustal ratio, which is around 0.02 (Wedepohl, 1995), but probably underestimated as we do not consider external sources of particulate manganese. The effective $\mathrm{Mn_{diss}}$ input into the ocean by rivers is presented in Fig. 4b. The global, effective river flux of Mn is $0.28 \, \mathrm{Gmol \, yr^{-1}}$.

### 2.2.3 Sediment source

The largest contribution of Mn to the upper ocean is dust deposition, but over large shelf and slope regions (e.g. polar oceans) the flux of $Mn_{diss}$ from the sediment can be of the same order of magnitude as, or higher than, the dust deposition flux (e.g. Middag et al., 2013, for the Southern Ocean). The (redox) reactions in the sediment are not explicitly modelled, since the sediment is not part of our model domain. Therefore, Mn addition from the sediment is modelled as a prescribed source. Since we do not have global maps of $Mn_{diss}$ sediment–seawater flux, we parameterise the flux based on existing parameterisations of nitrate and iron.

Middelburg et al. (1996) derived an empirical model for calculating the denitrification rate as a function of the seafloor depth:

$$\zeta_{Fsed} = -0.9543 + 0.7662 \cdot \ln(z_{Fsed}) - 0.235 \cdot \ln(z_{Fsed})^2, \tag{3}$$

where $\zeta_{Fsed}$ is the natural logarithm of the denitrification flux ($\mu mol\,cm^{-2}\,d^{-1}$ of carbon) and $z_{Fsed}$ is a function of bathymetry. Aumont et al. (2015) used this model for their sediment source of dissolved Fe in the PISCES model. They used a high-resolution bathymetric map to account for the shallow shelves, and modulated the seafloor depth ($z \to z_{Fsed}$) as follows:

$$z_{Fsed} = \min\left(8, \left(\frac{z}{500\,\mathrm{m}}\right)^{-1.5}\right). \tag{4}$$

We set the maximum iron flux to $1\,\mu mol\,m^{-2}\,d^{-1}$, following Aumont and Bopp (2006). We assume a porewater ratio $r_{Mn:Fe,sed} = 0.2$, and follow the same method as Aumont et al. (2015). Our final sediment addition of $Mn_{diss}$ into bottom water is given by:

$$\frac{\partial \mathcal{D}}{\partial t}\bigg|_{sediments} = \frac{0.2\,\mu mol\,m^{-2}\,d^{-1}}{\Delta z_{sed}} \cdot \min\left(1, 2e^{\zeta_{Fsed}}\right), \tag{5}$$

where $z_{sed}$ is the gridbox thickness of the bottom gridbox (just above the seafloor) in metre. With this prescribed source, the Mn flux is limited to a maximum of about $75\,\mu mol\,m^{-2}\,yr^{-1}$. This can be seen in Fig. 4c: the higher-than-70 $\mu mol\,m^{-2}\,yr^{-1}$ regions on the shelves is especially notable as it falls in the upper (red) part of the colour scale. The global sediment flux of Mn is $3.4\,Gmol\,yr^{-1}$.

### 2.2.4 Hydrothermal source

Hydrothermal vent $Mn_{diss}$ fluxes are modelled proportional to that of $^3He$ in Rüth et al. (2000) and Dutay et al. (2004). This approach is shown to have worked for iron (Bowers et al., 1988; Douville et al., 2002; Tagliabue et al., 2010; Resing et al., 2015). The basic equation for the change of $Mn_{diss}$ from hydrothermal vent influx is

$$\frac{\partial \mathcal{D}}{\partial t}\bigg|_{hydrothermal} = r_{Mn:^3He,hydro} \cdot \frac{\partial [^3He]}{\partial t}, \tag{6}$$

where $r_{Mn:^3He,hydro} = 0.1 \times 10^9$ is the ratio between the $Mn_{diss}$ and $^3He$ effective inflow from hydrothermal vents into the model domain. Recent observational studies found $Mn_{diss}/^3He$ concentration ratios of $4.0 \times 10^6$ (Kawagucci et al., 2008) and

$3.5 \times 10^6$ (Resing et al., 2015) at observational sites close to hydrothermal outflux regions. To satisfy this, we assume that a dissolved fraction of 4 % is left when the hydrothermal plume reaches these two observational sites.

This high "solubility fraction" means that the hydrothermal vents are a large source of $Mn_{diss}$ compared to Fe (Tagliabue et al., 2010; Resing et al., 2015). The integrated Mn flux is $102 \, Gmol \, yr^{-1}$. In our simulations we will show that we need to assume such a large flux to explain the observations. This choice also relates to the fast modelled removal rate of Mn near the depth of hydrothermal vents (explained later). The hydrothermal $Mn_{diss}$ source is presented in Fig. 4d.

### 2.2.5 Redox processes

Reduction and oxidation of Mn within the water column is a combination of several processes. Here we assume that $Mn_{ox}$ is subject to reduction with a constant rate, but significantly stimulated by sunlight (e.g. Sunda and Huntsman, 1994). This is taken into account in the model by using different $k_{red}$ for the euphotic and aphotic zones of the ocean. However, there are other processes playing a role that can locally be important. Those include the microbial enhancement of the rate of oxidation in regions where $Mn_{diss}$ supply is high (Sunda and Huntsman, 1994; Tebo et al., 2005), and the dependence on the $O_2$ concentration and pH (Johnson et al., 1996; Rijkenberg et al., 2014, for Fe at GA02). However, at this stage we decide to not include a dependency on $[O_2]$ to the model. Hence, here we choose to model Mn following a pseudo-first-order reaction where $k_{ox}$ and $k_{red}$ are pseudo-first-order rate constants (conventional primes omitted, e.g. Stone and Morgan, 1990):

$$\frac{\partial \mathcal{D}}{\partial t}\bigg|_{redox} = -k_{ox}\mathcal{D} + k_{red}\mathcal{X} \tag{7}$$

$$\frac{\partial \mathcal{X}}{\partial t}\bigg|_{redox} = k_{ox}\mathcal{D} - k_{red}\mathcal{X} \, , \tag{8}$$

where $\mathcal{X}$ is the particulate oxidised Mn concentration, and

$$k_{red} = \begin{cases} k_{red,light} & \text{in the euphotic zone} \\ k_{red,dark} & \text{elsewhere} \, , \end{cases} \tag{9}$$

where $k_{red,light}$ is the reduction rate in the euphotic zone and $k_{red,dark}$ in the aphotic zone. The euphotic zone is defined as the depths where the sunlight intensity is at least 1 %. The value for $k_{red,light}$ is taken from Sunda and Huntsman (1994) who found a mean dissolution rate of natural Mn oxides of $(98 \pm 23) \times 10^{-3} \, h^{-1}$. The $k_{red,dark}$ is much smaller and much more uncertain (varying a factor of fifteen in Sunda and Huntsman (1994)). As we have both dissolved and particulate Mn in our model, we chose to fit the $k_{ox}/k_{red,light}$ to $[Mn_{ox}]/[Mn_{diss}]$ from observations, resulting in a value within the observational range of Sunda and Huntsman (1994). At the time of this study only Bruland et al. (1994) was known to us as reporting accurate dissolved and particulate Mn for the same samples (at the VERTEX-IV station). From this $k_{ox}$ was calculated, and as Bruland et al. (1994) sampled the deep ocean as well, $k_{red,dark}$ was derived from that study as well (Table 3).

For one simulation, we will introduce a threshold on the oxidation process of $Mn_{diss}$. In that case, oxidation only takes place when $[Mn_{diss}]$.higher than a certain threshold value $\mathcal{D}_{thr}$; in other words, $k_{ox}$ is multiplied by $H(\mathcal{D} - \mathcal{D}_{thr})$. The Heaviside step function $H(x)$ equals zero where $x \leq 0$ and one where $x > 0$. Based on observations, we have estimated $\mathcal{D}_{thr} = 0.125 \, nM$,

corresponding to the observed deep ocean background value away from the influence of hydrothermal sources. The value can be reproduced by first sorting the $Mn_{diss}$ concentrations. Then we cut off the third quartile to remove the high values (above 0.42 nM), and chose a low value for a typical background concentration (0.13 nM is the first quartile). The reason for not choosing simply the minimum value is that values close to that can be lower than the 'background value' because of local removal processes.

### 2.2.6 Settling and burial

Manganese oxides settle, resulting in a concentration change according to

$$\left.\frac{\partial \mathcal{X}}{\partial t}\right|_{settling} = -w_{ox} \cdot \frac{\partial \mathcal{X}}{\partial z}, \tag{10}$$

where $w_{ox}$ is the settling velocity, set to a constant $1\,\mathrm{m\,d^{-1}}$ as long as $[Mn_{ox}] \leq 25\,\mathrm{pM}$. If $[Mn_{ox}] > 25\,\mathrm{pM}$, the settling velocity is not a constant any more but a function of depth. Still, in the mixed layer $w_{ox} = 1\,\mathrm{m\,d^{-1}}$, but if $[Mn_{ox}] > 25\,\mathrm{pM}$ the sinking speed increases linearly such that it reaches a value of $10\,\mathrm{m\,d^{-1}}$ at 2.5 km depth. The manganese oxide is buried when arriving at the ocean floor, which means that it is removed from the model domain.

The rationale for increasing the settling velocity above a certain threshold is that if $Mn_{ox}$ is large enough, dense aggregates form that have faster sinking rates. This is not unlike increasing velocities of detritus in some models (Kriest and Oschlies, 2011; Aumont et al., 2015). Notably mineral particles (sand, clay, carbonate) of high density in the order of 2–3 times that of seawater are responsible for this (McCave, 1975). Particulate organic carbon may play an important role as well (Passow and De La Rocha, 2006). In this way, $[Mn_{diss}]$ does not go to zero while still providing a deep ocean sink. As long as this critical particulate Mn concentration of 25 pM is not reached, aggregation of small $Mn_{ox}$ particles does not yet occur. The choice of the critical concentration of 25 pM, also referred to as the *aggregation threshold*, is derived from the redox rate constants in combination with a typical, low value of the observed dissolved Mn concentration (here chosen as 0.125 nM): $\mathcal{X}_{thr} = (k_{ox}/k_{red,dark}) \cdot 0.125\,\mathrm{nM} = 25\,\mathrm{pM}$. More precisely, settling of $Mn_{ox}$ follows Eq. (10) with the settling velocity

$$w_{ox}/(\mathrm{m\,d^{-1}}) = 1 + 9 \cdot \max\left(0, \frac{z - MLD}{2.5\,\mathrm{km}}\right) \cdot H(\mathcal{X} - \mathcal{X}_{thr}), \tag{11}$$

where $z$ is the depth, $MLD$ is the mixed layer depth, and $\mathcal{X}_{thr}$ is the aggregation threshold set to 25 pM.

### 2.2.7 Biological cycle

Manganese is incorporated into phytoplankton during growth. To this end, we run the PISCES-v2 biogeochemical ocean model (Aumont et al., 2015) together with our manganese model. The biological processes where Mn is involved are modelled in proportion to the change in phosphate concentration, thus it is given by:

$$\left.\frac{\partial \mathcal{D}}{\partial t}\right|_{biology} = R_{Mn:P} \cdot \left.\frac{\partial [PO_4^{3-}]}{\partial t}\right|_{biology}, \tag{12}$$

where $[PO_4^{3-}]$ is the phosphate concentration. The extended Redfield ratio for Mn, $R_{Mn:P}$, is set to $0.36 \times 10^{-3}$, the value that was determined from data at the GIPY5 Zero Meridian section (Middag et al., 2011a). ~~This is close to the value 0.39e-3~~

derived by~~Middag et al. (2013)~~ ~~from dissolved distributions.~~ This is a typical value for the manganese to phosphorus ratio in phytoplankton, though the full range of synchrotron X-ray fluorescence determined ratios, i.e. those that correspond to intracellular concentrations, is $0.2 \times 10^{-3}$ to $1.5 \times 10^{-3}$ (Twining and Baines, 2013).

There is no growth limitation of phytoplankton by shortage of dissolved Mn, i.e., in the model, manganese does not affect the biological carbon cycle.

There are four types of plankton in PISCES: nanophytoplankton, diatoms, microzooplankton and mesozooplankton (presented in Fig. 3 as the box "living"). There are two detrital pools, namely small particles settling with $2 \, \mathrm{m \, d^{-1}}$ and large particles settling in our model configuration with $50 \, \mathrm{m \, d^{-1}}$ (box "detrital"). Manganese is incorporated in the four living pools according to the PISCES equations for phosphorus. Through other processes they become part of detrital material or particulate organic matter. These pools are, however, not explicitly followed. Only the biological sources and sinks of $\mathrm{Mn_{diss}}$ are modelled, entailing the conversion of $\mathrm{Mn_{diss}}$ to the two phytoplankton pools, and from all particle pools to $\mathrm{Mn_{diss}}$.

## 2.3 Simulations

The *Reference* simulation was spun up for 600 yr to reach a steady state. From year 100 onwards two sensitivity simulations were forked off to run in parallel with *Reference* for another 500 yr. These simulations are variations of the *Reference* simulation that uses the parameters listed in Table 3. Table 4 lists the simulations and their key parameters.

| Simulation name | $r_{\mathrm{Mn}:^3\mathrm{He,hydro}}$ | $w_{\mathrm{ox}}/(\mathrm{m \, d^{-1}})$ | $\mathcal{X}_{\mathrm{thr}}/\mathrm{pM}$ | $\mathcal{D}_{\mathrm{thr}}/\mathrm{nM}$ | $R_{\mathrm{Mn:P}}$ |
|---|---|---|---|---|---|
| *Reference* | $0.10 \times 10^9$ | 1–10 | 25 | 0 | $0.36 \times 10^{-3}$ |
| *NoBio* | $0.10 \times 10^9$ | 1–10 | 25 | 0 | **0** |
| *LowHydro* | $\mathbf{0.01 \times 10^9}$ | **1** | – | 0 | $0.36 \times 10^{-3}$ |
| *NoThreshold* | $0.10 \times 10^9$ | 1–10 | **0** | 0 | $0.36 \times 10^{-3}$ |
| *OxidThreshold* | $0.10 \times 10^9$ | 1–10 | **0** | **0.125** | $0.36 \times 10^{-3}$ |

**Table 4.** List of simulations with the parameters changed compared to the reference simulation in boldface.

Our second simulation is without a biological cycle of Mn (*NoBio*). In this simulation Eq. 12 is removed from the model, or, equivalently, $R_{\mathrm{Mn:P}}$ is set to zero. This simulation should illustrate the consequences of the (lack of) biological incorporation and subsequent remineralisation of Mn.

The goal of *LowHydro* is to investigate whether the combination of the high hydrothermal input of $\mathrm{Mn_{diss}}$, and strong aggregation (modelled as a high settling velocity), is needed to obtain a veracious representation of the distribution of $\mathrm{Mn_{diss}}$, i.e. one where the predicted concentrations compare well with the observed concentrations. Specifically, we want to explain the sharp observed Mn plumes. To this end, we first decreased the settling velocity to a constant $1 \, \mathrm{m \, d^{-1}}$ (or, equivalently, $\mathcal{X}_{\mathrm{thr}} \to \infty$), which, as expected, resulted in a wide spreading of $\mathrm{Mn_{diss}}$ and a too high $[\mathrm{Mn_{diss}}]$ almost everywhere in the deep ocean (not presented). Decreasing only hydrothermal input would trivially result in a proportionally smaller $[\mathrm{Mn_{diss}}]$ near the

vents. For this reason *LowHydro* contains two changes compared to the *Reference* simulation: a tenfold decrease in both the hydrothermal flux and the maximum settling velocity.

In our fourth simulation, *NoThreshold*, we want to see if an aggregation threshold is needed for a veracious $[\mathrm{Mn_{diss}}]$ simulation. The threshold is removed by setting $\mathcal{X}_{\mathrm{thr}}$ in Eq. 11 to zero.

The fifth and final simulation, *OxidThreshold*, is one that has a threshold in the oxidation process instead of the particle sinking, setting effectively a minimum concentration of dissolved Mn.

## 2.4 Model validation

In this study we will mainly use data from the GEOTRACES programme, but for a worldwide global ocean comparison one has to rely also on data that were collected in the era before the reference samples of SAFe and GEOTRACES were available. Table 5 lists the datasets from GEOTRACES expeditions, as well as several other datasets; Fig. 5 shows the coordinates of the stations.

Most of these GEOTRACES datasets are part of the publicly available GEOTRACES Intermediate Data Product 2014 (IDP)
(Mawji et al., 2015), except for GP16 (Resing et al., 2015) that will be released as part of the GEOTRACES Intermediate Data Product 2017.

These observations are used for a visual global ocean data-model comparison. Details on the statistical and visual model–data comparison are presented in Appendix A. Of these observations, only the West Atlantic Ocean data (the GEOTRACES GA02 transect, consisting of 64 PE 319, PE 321 and JC 057; ~~1440~~1320 points) and the Zero-Meridian Southern Ocean data
(GIPY5; 468 points) have been used for statistical comparison with the model. Moreover, only the data of the shipboard FIA have been used, which have five stations less than the on-shore mass spectrometry determinations (ICP-MS having 120 more measurement, thus 1440 in total). The focus of this study is the West Atlantic Ocean for several reasons. Firstly, recent measurements have been carried out in that region, resulting in a large consistent (one method) dataset. Other regions generally contain fewer measurements and are mainly based on different methods by different analysts. Secondly, the West Atlantic
Ocean is of importance to the Atlantic meridional overturning circulation, and hence the deep ocean cycling of for example the major nutrients. Therefore the West Atlantic Ocean was chosen as a key site for the $\sim 18\,000$ km long GEOTRACES GA02 transect for the collection of data for dissolved Mn and a suite of other trace elements and isotopes. The GEOTRACES GIPY5 transect at the Zero Meridian in the Southern Ocean complements the GA02 transect up to Antarctica, giving a more complete picture of the ocean circulation. For these reasons in this study we focus on the GA02 and GIPY5 transects.
The particulate $[\mathrm{Mn_{ox}}]$ measurements from Landing and Bruland (1987) were used to tune the redox model. The data from the CLIVAR P16 cruise are unpublished; sampling is from the surface up to $1000$ m depth; its methods of analysis are described by Milne et al. (2010).

| Transect | Year | Expedition | Ocean basin | Citation | # |
|---|---|---|---|---|---|
| GEOTRACES transects | | | | | |
| GIPY11 | 2007 | ARK XXII/2 | Arctic Ocean | Middag et al. (2011b) | 773 |
| GIPY4 | 2008 | MD166 BONUS-GoodHope | Southern Ocean | Boye et al. (2012) | 233 |
| GIPY5 | 2008 | ANT XXIV/3 | Southern Ocean | | |
| | | | a) Zero Meridian | Middag et al. (2011a) | 468 |
| | | | b) Weddell Sea | Middag et al. (2013) | 176 |
| | | | c) Drake Passage | Middag et al. (2012) | 221 |
| GI04 | 2009/2010 | KH-09-5 | Indian Ocean | Vu and Sohrin (2013) | 233 |
| GA02 | 2010 | 64 PE 319 | Northwest Atlantic Ocean | *this study* | 384 |
| GA02 | 2010 | 64 PE 321 | Northwest Atlantic Ocean | *this study* | 504 |
| GA02 | 2011 | JC 057 | Southwest Atlantic Ocean | *this study* | 432 |
| GA03 | 2010 | US GT10 | North Atlantic | Wu et al. (2014) | 91 |
| GA03 | 2011 | US GT11 | North Atlantic | Wu et al. (2014) | 578 |
| GP16 | 2013 | US EPZT | South Pacific | Resing et al. (2015) | 874 |
| Other expeditions and datasets | | | | | |
| - | 1983 | VERTEX-IV | North Pacific Ocean | Landing and Bruland (1987) | 27 |
| - | 2006 | EUCFe (*Kilo Moana*) | Pacific Ocean | Slemons et al. (2010) | 349 |
| - | 2005/2006 | CLIVAR P16 | Pacific Ocean | Milne and Landing (unpublished data) | 174 |
| - | 2007 | CoFeMUG | South Atlantic | Noble et al. (2012) | 429 |
| Total number of dissolved Mn measurements: | | | | | 5742 |

**Table 5.** Observational $[\mathrm{Mn_{diss}}]$ used for comparison with the model simulations. GEOTRACES datasets are indicated by GEOTRACES transect codes. Their accuracies are approved by GEOTRACES on the basis of results of reference samples and cross-over stations. At the time there were no reference samples available for the other datasets, or they were not used for Mn.

## 3 Results

### 3.1 Observations in the West Atlantic Ocean

Figure 6 shows the West Atlantic GEOTRACES transect at the top 500 m. The concentration of $\mathrm{Mn_{diss}}$ near the ocean surface is high, going up to 4 nM and beyond. The highest surface concentrations are observed in the regions of high dust deposition as previously observed for aluminium (Middag et al., 2015a) and consistent with dust as the main source of Mn to the surface open ocean. The $\mathrm{Mn_{diss}}$ distribution is mainly homogeneous in the intermediate and deep ocean, where dissolved Mn has a concentration of about 0.10 nM to 0.15 nM.

North of 40° N, high concentrations of dissolved Mn reach deeper than the top couple of hundred metres, apparently because of vertical mixing. As can be seen in Fig. 7, some of it seems to go all the way down to about 3000 m. However, other sources

may be (partly) responsible as well, among which advection from diffusive sediments and hydrothermal vents. By plotting all deep waters at enhanced resolution of concentration (Fig. 7), the Mn maxima of hydrothermal plumes and the $O_2$ minimum zones are better discernible. Since Mn and Fe have a very similar redox chemistry in the oceans, not surprisingly the maxima of Mn and Fe (Rijkenberg et al., 2014) generally overlap (the red dashed and black solid ellipses in Fig. 7). Some of the Mn may be transported southwards by North Atlantic Deep Water (NADW), but this cannot be deduced from this transect: already around $35°$ N, $[Mn_{diss}]$ approaches near-constant deep background concentrations such that the NADW plume is no longer discernible. Similarly, at about $35°$ S, around 1000 m depth, we observe elevated concentrations in the northward advecting Antarctic Intermediate Water (AAIW), but once again concentrations already reach the typical background concentration around $20°$ S. In the Northern Hemisphere, the subsurface waters underlying the high dust deposition region near the equator have relatively elevated $Mn_{diss}$ concentrations down to 750–1000 m depth, implying some influence of the dust deposition and particle export on the subsurface Mn concentrations, but yet again, the typical background concentration is reached further down. Elevated local features are located at almost 2 km depth on the Zero Meridian at $50°$ S (Middag et al., 2011a, their Fig. 2; also presented as dots from $52°$ S on our Fig. 9) and almost 3 km depth in the West Atlantic transect at, and just south of, the equator, and at the Denmark Strait overflow (Fig. 7). The elevated Mn in the Denmark Strait Overflow Water (DSOW) for a small part coincides with elevated aluminium attributed to sediment resuspension (Van Hulten et al., 2014; Middag et al., 2015a). However, there is a mismatch between the highest Al concentrations observed around $45°$ N and the highest Mn in the northernmost part of the transect (Middag et al., 2015a). This implies the source of Mn is related to the DSOW rather than the sediment resuspension occurring while the DSOW advects into the North Atlantic Ocean. The elevated deep Mn at the other locations match with elevated Fe and is most likely of hydrothermal origin (Klunder et al., 2011; Gerringa et al., 2015). Finally, at low latitudes, in the very deep ocean, $[Mn_{diss}]$ decreases with depth, suggesting a slower circulation in that region or an additional/enforced export of Mn to the sediment.

## 3.2 Reference simulation

After 600 yr, both the upper 100 m and the deep ocean relative Mn content changed by less than 25 ppm over a period of 10 yr in the *Reference* simulation. This is about a factor of four less compared to the decadal change of 100 yr ago. The small and reducing drift suggests that the simulation is practically in a steady state. Figure 8 shows the modelled and measured dissolved Mn concentrations at four depths; observations as coloured dots (same scale). The dissolved Mn concentration is high in the surface of the Atlantic, Indian and Arctic oceans, mostly consistent with the observations (Fig. 8a). The model also reproduces the latitudinal gradient of $[Mn_{diss}]$ in the Atlantic and Indian oceans, reflecting dust deposition patterns (Fig. 4a).

Figure 9 shows the dissolved Mn concentrations at full depth in the Zero-Meridian Southern Ocean and the West Atlantic Ocean from the *Reference* simulation; observations as coloured dots. Lower concentrations in the deep ocean are reproduced by the model (Figs 8b–d and 9). Both the model and observations present a mainly homogeneous distribution of just over 0.1 nM, though there are a number of sub-0.1 nM measurements in the polar oceans. The $Mn_{diss}$ concentrations are generally higher near the surface compared to the deep ocean, both in the observations and the model. This is caused by a combination of dust deposition and photoreduction. Also the penetration of $Mn_{diss}$ from Mn-rich surface waters into the deep ocean at around

| Simulation | Correlation | RMS deviation (nM) | Reliability index | Mn (Gmol) | Percentage $Mn_{ox}$ |
|---|---|---|---|---|---|
| *Reference* | 0.78 ($\pm$0.05) | 0.46 ($\pm$0.05) | 1.76 ($\pm$0.08) | 440 | 14.1 |
| *NoBio* | 0.79 ($\pm$0.05) | 0.45 ($\pm$0.05) | 1.76 ($\pm$0.08) | 409 | 14.0 |
| *LowHydro* | **0.64** ($\pm$0.07) | **0.60** ($\pm$0.05) | **2.90** ($\pm$0.08) | 1023 | 15.8 |
| *NoThreshold* | 0.78 ($\pm$0.05) | 0.46 ($\pm$0.05) | **2.03** ($\pm$0.11) | 373 | 13.7 |
| *OxidThreshold* | 0.78 ($\pm$0.05) | 0.46 ($\pm$0.05) | 1.82 ($\pm$0.09) | 405 | 12.6 |

**Table 6.** Statistical model–data comparison for $[Mn_{diss}]$ at the GEOTRACES West Atlantic GA02 transect. The significance errors in the entries of *Reference* are $\pm 2\sigma$ from Monte Carlo samplings (Appendix A). Bold face values denote significant worsening compared with *Reference*; deviations of other values from the *Reference* value are insignificant. The last two columns are the total modelled Mn in seawater, and the Mn oxides as a portion of the total.

50° N is reproduced by the model, but this is scavenged quickly before traversing southward in the NADW, which is consistent
with early studies (e.g. Bender et al., 1977). Finally, the *Reference* simulation reproduces the measurements near hydrothermal
vents in the Atlantic and Indian oceans (Figs 8c and 9).

Still, at many places, the $Mn_{diss}$ concentration is underestimated by the model, with the most notable exceptions of the
Southern and Arctic oceans. The underestimation is especially pronounced at the surface of the Pacific Ocean (Fig. 8a). Fur-
thermore, in the model the Mn-rich water from the Amazon does not reach the GA02 transect as opposed to the observations.
This potentially explains the underestimated concentration of $Mn_{diss}$ in that region. The simulation overestimates measured
$[Mn_{diss}]$ in the Arctic Ocean, except for some coordinates like near the Gakkel Ridge (the dark red dot in Fig. 8c at 90° E)
(Middag et al., 2011b). The Gakkel Ridge is not in our hydrothermal forcing field (Fig. 4d).

Figure 10 presents $[Mn_{diss}]$ of the GA03 west to east transect. Observations from Wu et al. (2014) are presented as coloured
dots. Again, the general patterns are captured, but the concentration of $Mn_{diss}$ very close to the Mid-Atlantic Ridge is underes-
timated, while above the ridge (at about 2–3 km depth) it is overestimated. In other words, the modelled hydrothermal $Mn_{diss}$
gradient is not as sharp as in the observations. It is difficult to improve this feature because of the low vertical model resolution
at that depth.

To more objectively compare the different simulations to each other, we list several goodness-of-fit statistics in Table 6. They
compare the model simulations to observational data from the GEOTRACES GA02 transect. The model–data $[Mn_{diss}]$ Pearson
correlation coefficient $r$ has a value of 0.78 for the *Reference* simulation. The reliability index, the average factor by which
model predictions differ from observations, shows that on average the model differs by a factor of 1.76 from observations
(Stow et al., 2009, and Appendix A).

### 3.3 Biological cycle disabled

As Mn plays an important role as a trace nutrient, it is likely that biology has an impact on $[Mn_{diss}]$ at the ocean surface. Therefore, a simulation has been performed where biological incorporation and remineralisation of $Mn_{diss}$ was not included, henceforth named *NoBio*.

Across much of the surface ocean, the biological cycle of Mn in *Reference* causes a notable decrease of $[Mn_{diss}]$ compared to *NoBio*, though in the Pacific Ocean there is an increase at around $20°$ N and especially around $20°$ S (Fig. 11). The decrease in the Pacific Ocean is up to $100\%$ near the equator and up to $80\%$ at $40°$ S. In the Atlantic Ocean, $[Mn_{diss}]$ has not changed much between the two model simulations, explaining the insignificant change in the model–data comparison (Table 6). Deeper in the Pacific Ocean, $[Mn_{diss}]$ is higher in *Reference* compared to *NoBio*.

Figure 12 shows $[Mn_{diss}]$ of the model compared with the observations at two depths for the *NoBio* simulation. The changes between the simulations are only big in the Pacific Ocean (Fig. 12). Elsewhere biology does not appear to have a big effect on the $Mn_{diss}$ concentration. At the surface of the equatorial Pacific Ocean, concentrations are higher in the *NoBio* simulation. Since in *Reference* $[Mn_{diss}]$ is too low compared to the US East Pacific Zonal Transect (EPZT) (GP16), *NoBio* compares better with the observations. Deeper in the Pacific Ocean, *Reference* has higher $[Mn_{diss}]$, hence is better than in *NoBio*.

### 3.4 Hydrothermal flux and export reduced

We have shown that the *Reference* simulation gives a reasonably veracious representation of the effects of hydrothermal vents and the background concentration in the deep ocean. This is achieved by setting a large $Mn_{diss}$ flux from hydrothermal vents and a big maximum settling velocity at the depth of the vents to remove the hydrothermal Mn away from the source. The simulation presented in this section, *LowHydro*, is meant to test if the high flux is necessary for a veracious $Mn_{diss}$ distribution.

Figure 13 shows the relative change in $[Mn_{diss}]$ at four depths when both the hydrothermal input and the maximum settling velocity are decreased by a factor of 10 (*LowHydro*). In the surface ocean (Fig. 13a) there are both moderate increases and decreases. However, there are very large increases in places with deep convection that connect with the deep ocean (over $500\%$ in the Weddell Sea). In the deep ocean below about $1\,km$ depth, such a large increase is not limited to the Weddell Sea but stretches over much of the Atlantic and Indian oceans (Fig. 13c and d). Exceptions are the locations near the oceanic ridges, 25 where there is hydrothermal activity. At those locations $[Mn_{diss}]$ decreased by up to $80\%$.

      The $[Mn_{diss}]$ transects in Fig. 14 show that *LowHydro* performs worse than the *Reference* simulation. For instance, the high $[Mn_{diss}]$ with a clear hydrothermal origin at $54°$ S has disappeared, while at the same time the deep ocean filled up with $Mn_{diss}$ resulting in a consistent overestimation of about a factor of five in most of the deep West Atlantic Ocean. In the South Pacific Ocean and the Indian Ocean the signature of hydrothermal input of $Mn_{diss}$ is – though smaller and worse than in *Reference* – 30 still clearly present, with $[Mn_{diss}]$ values near the ridges distinctly different from the "background" concentration (results not presented). Furthermore, the statistics on the points at the West Atlantic GA02 transect of *LowHydro* compared to *Reference* unambiguously show that $[Mn_{diss}]$ worsened (Table 6). The *LowHydro* simulation is significantly worse in all three statistics:

the gradients from hydrothermal vents have disappeared, and [$\text{Mn}_\text{diss}$] is much too high throughout the ocean compared to *Reference*.

## 3.5 Aggregation threshold disabled

The simulation *NoThreshold* does not impose the aggregation threshold, meaning that settling is unconstrained in this simulation. Figure 15 presents [$\text{Mn}_\text{diss}$] from *NoThreshold* at the GA02 and GIPY5 transects. In the intermediate and deep ocean south of $40°$ N the concentration of $\text{Mn}_\text{diss}$ is generally more than $50\%$ smaller than that in *Reference* (Fig. 9).

The homogeneous, already low background concentration of $\text{Mn}_\text{diss}$ is reduced to close to zero (at least in the deep South Atlantic Ocean), while the hydrothermal signals are still correctly represented in the *NoThreshold* simulation. This explains that neither the correlation coefficient nor the root-mean-square deviation, based on the West Atlantic GA02 data, of [$\text{Mn}_\text{diss}$] in *NoThreshold* differ significantly from those of *Reference*. This means that away from ocean ridges the spatial variation of the modelled [$\text{Mn}_\text{diss}$] is similar to that of the observations (Stow et al., 2009). However, the reliability index, the average factor by which model predictions differ from observations, here also based on the GA02 data, has changed from 1.76 to 2.03, a change that is significant by six standard deviations (Table 6). Therefore, the *NoThreshold* simulation is much worse than the *Reference* simulation.

## 3.6 Oxidation threshold enabled

As a final simulation, we limited the oxidation, rather than the settling of the manganese oxides. The resulting dissolved Mn concentrations are similar to those of *Reference* which has an aggregation threshold. Figure 16 shows the $\text{Mn}_\text{ox}/\text{Mn}_\text{diss}$ concentration ratio of the two simulations as well as observations at ($155°$ W, $28°$ N). The blue squares represent the ratios between the particulate and dissolved Mn measurements taken during the VERTEX-IV cruise (Bruland et al., 1994). The left panel shows that this ratio lies near 0.005 in the upper $100$ m, and the right panel shows the deep ocean that has higher values. The green line is the $\text{Mn}_\text{ox}/\text{Mn}_\text{diss}$ concentration ratio from the *Reference* simulation, and the red, dashed line is the *OxidThreshold* simulation. For the *Reference* simulation, the modelled $\mathcal{X}/\mathcal{D}$ generally lies at the lower end of the observed ratio, while only between 200 and 400 m particles are overestimated compared to dissolved Mn. Whereas in the upper $100$ m of the ocean both simulations are consistent with those from the VERTEX-IV cruise, in the deep ocean only the *Reference* simulation compares well with the observations. In *OxidThreshold* the ratio is underestimated.

## 4 Discussion

Many of the properties of the $\text{Mn}_\text{diss}$ distribution in the world ocean are reproduced by our *Reference* simulation (Sect. 3.2). However, some discussion on the assumptions of the underlying processes in the model is required.

## 4.1 Margin sediments

We have chosen to use a simple sediment flux parameterisation for this study. In our model, $Mn_{diss}$ is added to bottom water from anoxic sediments analogously to the iron flux in the model PISCES (Aumont et al., 2015).

On the one hand, our model may underestimate the flux. In fact, with Mn reduction a larger free energy is released compared to Fe reduction, so that Mn oxides reduce more easily (Froelich et al., 1979). Furthermore, high benthic Mn fluxes have been observed from eastern North Pacific marine sediments (McManus et al., 2012). On the other hand, at some large-shelf regions, like in parts of the Arctic Ocean, the Mn flux is overestimated, because the low model resolution does not handle ~~large~~ shelf regions well. This problem is similar for iron (Aumont et al., 2015, their Figure 8c, d).

Slomp et al. (1997) measured pore water concentrations of dissolved Fe and Mn. The Mn/Fe ratios based on their pore water concentrations typically range from 0.2 to 0.5, but sometimes up to 2.5. The reduction rate found by their reaction–diffusion model yields a Mn/Fe reduction rate ratio of 0.02 to 0.2. Balzer (1982) reports fluxes from sediment to bottom water of dissolved Mn and Fe that yield an average ratio of 0.5. The value that we used, 0.2, lies between the intermediate and lower end of the reported values. This choice is mainly due to the fact that the crude iron flux parameterisation results in increases of the Mn flux that are out of proportion in some regions of the ocean like the eastern Arctic Ocean and the Indonesian Throughflow. This is a known shortcoming of this parameterisation that can also be seen in the modelled iron distribution (Tagliabue and Völker, 2011). In the Arctic Ocean, a higher sedimentary Mn input would induce an even stronger overestimation than it already does with $Mn/Fe = 0.2$ (Fig. 4c). The shortcoming had also been concluded by Van Hulten et al. (2013) who presented a sensitivity study that used the same parameterisation for aluminium sediment input as for Mn here.

Nevertheless, a higher $Mn_{diss}$ input would be beneficial for the simulation in the Atlantic; and especially the Pacific Ocean, where $[Mn_{diss}]$ is underestimated everywhere but south of 50° S. This general underestimation may be due to underestimation of the enhanced dissolved Mn as previously measured in coastal upwelling regions and their underlying Oxygen Minimum Zones (OMZs). Reports of elevated dissolved Mn in major Pacific coastal upwelling regions are in the California Current of the Northeast Pacific (Landing and Bruland, 1987; Chase et al., 2005), and the Peru Humboldt Current of the Southeast Pacific (GP16, Resing et al. (2015); Hawco et al. (2016)). The mobilisation of Mn in these two Pacific upwelling systems is comparable to the elevated dissolved Mn in two other classical upwelling systems, off Namibia in the Southeast Atlantic (CoFeMUG, Noble et al. (2012)), and in the Northwest Indian Ocean (GI04, Saager et al. (1989); Vu and Sohrin (2013)). Indeed in the surface waters (Fig. 8a) the model does predict elevated dissolved Mn off California and off Peru, but simulated concentrations in the $\sim 0.6$ to $\sim 2$ nM range are lower than measured values up to 4 or even 10 nM off California, and up to $\sim 3.4$ nM off Peru (Fig. 8a). Otherwise off Namibia there appears to be more agreement between the simulated and the measured dissolved Mn in surface waters. The upwelling regions also are characterised by an extensive OMZ in the subsurface waters. In the eastern Pacific a tongue of very low dissolved $O_2$ is most apparent at around 300 m depth (Hawco et al., 2016, their Figure 1) and would yield elevated dissolved Mn. In our model at the 500 m depth horizon (Fig. 8b) this is indeed seen off California and off Namibia, but not reproduced very well off Peru where some higher measured Mn up to $\sim 2$ nM was in

fact observed. The very low dissolved $O_2$ and high dissolved Mn in these OMZ waters, lead to strong spatial gradients, and a
more fine resolution of the model in upwelling and OMZ regions would be required.

To summarise, we had chosen to use a simple sediment flux parameterisation, which is not completely adequate for representing $Mn_{diss}$ fluxes from OMZ sediments. For future work a finer regional resolution of the model in upwelling/OMZ regions, better parameterisation, or even an explicit sediment submodel, should be taken into consideration.

## 4.2 Biological cycle

### 4.2.1 General discussion and the GIPY5 transect

Our model includes biological processes involving manganese in a way very similar to that of phosphorus, but the rate variables are multiplied by a typical Mn/P ratio based on the measured plankton content ratio. We had chosen to follow all phosphorus cycling as described by Aumont et al. (2015) for consistency reasons. We did not want to evaluate all potential details of biology but rather take a simple approach.

At this stage, we lack observational constraints allowing us to develop a more complex representation of the biological Mn cycle. Specifically, even though in the upper waters of the Southern Ocean the uptake and remineralisation cycle of $Mn_{diss}$ correlates well with the nutrients $PO_4^{3-}$ and $NO_3^-$ (Middag et al., 2011a), there is no mechanistic evidence for how the uptake-remineralisation process exactly should work for Mn. Of course, Mn plays an important role in biology (Coale, 1991; Middag et al., 2011a, 2013; Browning et al., 2014). Whereas the trace nutrient iron often appears more significant (Buma et al., 1991), manganese might even be limiting to primary production in parts of the Southern Ocean. This is at least suggested by the observational data: at the Zero-Meridian GIPY5 transect south of $58°$ S, the dissolved Mn concentration is very low at the surface, and from $\sim50$ m to $250$ m there is a maximum of $[Mn_{diss}]$ (Fig. 9, left transect). This is probably due to remineralised particulate organic matter (Middag et al., 2011a). The maximum is not found in the model. For instance, near $66°$ S at $1000$ m upwards, the modelled $[Mn_{diss}]$ decreases monotonically, which suggests that biology does not explain the subsurface maximum. However, our model may not be adequate. Firstly, it may be unreasonable to treat Mn biologically identical to P. Secondly, the subsurface maximum can be a direct consequence of a low oxygen concentration, because that would decrease the oxidation rate $k_{ox}$, and our redox model does not depend on $[O_2]$. This is not the most probable hypothesis, because at least during the ANT XXIV/3 expedition only a slight oxygen minimum occurred around $300$–$400$ m depth which lies well below the Mn maximum (Middag et al., 2011a). Thirdly, there could be problems with the underlying model. For instance, if the vertical mixing in the underlying circulation model is too strong, low-$[Mn_{diss}]$ waters mix from the deep Southern Ocean upwards through the $100$ m to $200$ m region, possibly removing the vertical gradient of $[Mn_{diss}]$.

The model suggests that a Mn biological cycle following P may have an important contribution to the Southern Hemisphere, especially in the Pacific Ocean (Fig. 11). This should be tested further through field experiments and more sophisticated model simulations.

The biologically incorporated portion of the particulate Mn fraction contributes most to the sinking Mn flux. The oxidised fraction has only a significant, though still smaller, contribution near hydrothermal vents compared to incorporated Mn (not

presented). It appears that, especially for low latitudes, biological incorporation is more important than oxidation. The flux
patterns at 100 m depth for incorporated (organic) and oxidised Mn are quite different (Fig. 17). The organic Mn flux pattern
is similar to that of POC and the major nutrients, whereas the oxidised Mn flux is typically higher, especially in the northern
seas. However, in the low-latitude Pacific Ocean, the $Mn_{ox}$ flux is much smaller than the organic Mn flux. So biology strongly
influences the already low $[Mn_{diss}]$ at the surface of the Pacific Ocean.

### 4.2.2 Relation to redox and settling

Our model yields small $[Mn_{diss}]$ in surface waters of the eastern Pacific Ocean. Either (i) the supply of dissolved Mn from
the extensive OMZ in subsurface waters of this region (Hawco et al., 2016, their Figure 1) is too low, thus not adequately
simulated by the model, or (ii) the loss due to biological uptake is too high. For the OMZ the observed vertical and lateral
gradients of both dissolved $O_2$ and dissolved Mn are very large, and would require a higher resolution of the model grid in the
OMZ regions. This would require major revisions of the model and for the time being is merely recommended for future work.
On the other hand, the hypothesis of perhaps too much biological removal has been tested by simply turning off the biological
module. The difference in East Pacific surface waters (Fig. 11) shows a large negative offset in exactly those surface waters that
overly the strong OMZ at 300 m depth (Hawco et al., 2016, their Figure 1). Also in Fig. 12 we see that a model run without
biological incorporation shows surface water Mn off California and off Peru that tend to better agree with the observations,
than in the complete model (Fig. 8), and also again the lateral distribution mimics the distribution of the underlying OMZ
subsurface waters. Thus overall it appears that the discrepancy is more due to inadequate simulation of the OMZ regions, than
due to too strong removal by the simulated biology. ~~This appears to be due to the biological cycle (Fig. 11a). The model does
not include the effects of oxygen minima, especially those at reductive sediments and upwelling regions in the eastern Pacific
Ocean. The remineralisation signal generates a more accurate simulation at intermediate depths (Figs 8, 11, 12: panels b).~~

Overall, the amount of manganese ($Mn_{diss} + Mn_{ox}$) in the world ocean is slightly larger with biology (440 Gmol) compared
to without (409 Gmol, Table 6). This is not expected, because biological incorporation would give an extra sink of Mn incor-
porated in particulate organic matter. The explanation of this paradox lies in the interaction between the biological cycle and
the settling process of $Mn_{ox}$ in the model.

We apply a concentration-dependent sinking speed, which generates an approximately homogeneous distribution of dis-
solved manganese of around 0.125 nM in much of the deep ocean. There is a finite range of input fluxes where this works. If
the input flux gets above a certain value, $[Mn_{diss}]$ exceeds this background value, as it occurs in the Atlantic Ocean north of
$35°$ N. However, if the input flux is low enough, $[Mn_{diss}]$ decreases below this background value because of slow $Mn_{ox}$ settling,
like it does in the East Pacific Ocean. Inflow of higher $[Mn_{diss}]$ waters by large-scale circulation also counts as a source that
can elevate Mn, but the currents in the deep Pacific Ocean are slow.

The biological processes modify this behaviour. In PISCES there are two types of particulate carbon: one of large particles
that sinks with $50 \, \mathrm{m\,d^{-1}}$ (in our version) and one of small particles that sinks with $2 \, \mathrm{m\,d^{-1}}$. Most carbon, thus also manganese,
is present in the pool of small particles. Even though, at least in the model, the Mn sources in the Pacific Ocean are small, there
is a lot of $Mn_{diss}$ in the surface ocean that would ultimately be $Mn_{ox}$ if it were mixed down below the photic zone depth.

However, with the biological cycle enabled, phytoplankton incorporates a part of the $Mn_{diss}$. Particulate organic Mn (just like POC) sinks down into the deep ocean. During the downward propagation of the organic Mn particles, they remineralise, releasing $Mn_{diss}$ in the deep ocean. A part of this $Mn_{diss}$ is converted to $Mn_{ox}$ that sinks down inefficiently ($1\,\mathrm{m\,d^{-1}}$) when $Mn_{ox}$ has not yet reached the equilibrium concentration $\mathcal{X}_{thr}$. The remineralised Mn fills up, through oxidation, the $Mn_{ox}$ pool in the deep ocean.

Biological processes store Mn in plankton, which is preserved from strong Mn oxide settling (in the model). This new pool of manganese fills the pools of dissolved and oxidised Mn in the deep ocean. This new pool is provided by the plankton that took it from the $Mn_{diss}$ pool in the photic zone. Without biology this would be available for oxidation to $Mn_{ox}$ during night. Adding this extra Mn to the integrated total yields a higher Mn quantity in *Reference* than in *NoBio*.

### 4.2.3 Atlantic–Pacific contrast

We saw a strong relative effect of biological incorporation in the Pacific Ocean but not in the Atlantic Ocean. This is because of the fact that $[Mn_{diss}]$ is already quite low in the Pacific Ocean. The added modelled sink of biological incorporation in the surface ocean, simply following P uptake, quickly drains $Mn_{diss}$ concentrations towards zero. In reality, $[Mn_{diss}]$ is still quite high in the surface of the Pacific Ocean. The high observed surface concentration is primarily caused by strong Mn sources from anoxic sediments in the Southeast Pacific Ocean, in combination with low $[O_2]$ in juxtaposed waters inhibiting manganese oxidation. The discrepancy of the model versus the observations of $[Mn_{diss}]$ is primarily due to the underestimation of those sources, and because in the model oxidation does not explicitly depend on $[O_2]$. At the western boundary of the Atlantic Ocean the Mn source is higher, but one would actually expect a high Mn source at eastern boundaries where upwelling occurs.

As dust deposition and dissolution are uncertain, we can also not exclude the possibility that we underestimate the Mn dust flux into the (South) Pacific Ocean. A preliminary simulation with the Mahowald et al. (1999) dust flux that is especially higher in the Pacific Ocean, shows more realistic $[Mn_{diss}]$ throughout the Pacific Ocean. Nonetheless, we decided to use the flux from Hauglustaine et al. (2004) that is, supposedly, more state-of-the-art. Furthermore, the Mahowald et al. (1999) dust flux did not fully solve the issue of the low $Mn_{diss}$ concentrations near the upwelling regions.

### 4.3 Redox rates

Although we chose to model Mn reduction and oxidation as first-order-reaction kinetics, redox of Mn within the water column is a combination of several processes. Firstly, $Mn_{ox}$ is subject to non-biological reduction, significantly stimulated by sunlight (e.g. Sunda and Huntsman, 1994). This is taken into account by using different $k_{red}$ for the euphotic and aphotic zones of the ocean. Secondly, the rate of oxidation is enhanced by microbes (bacteria and fungi) in regions where $Mn_{diss}$ supply is high (Sunda and Huntsman, 1994; Tebo et al., 2005), and it depends on the $O_2$ concentration and pH. This has been observed in the North Pacific Ocean (Johnson et al., 1996, and references therein), as well as in some GEOTRACES transects in the Atlantic Ocean. Measurements from the West Atlantic Ocean and Zero-Meridian Southern Ocean GEOTRACES cruises show some variations with depth, e.g. near $10°$ N and near $40$–$47°$ N (top of Fig. 7). This corresponds to the maximum of dissolved Fe associated with the oxygen minimum as reported for the same cruise by Rijkenberg et al. (2014). Our model does not reproduce

such a maximum of dissolved Mn, because in the model the Mn oxidation rate does not depend on $[O_2]$. Furthermore, fluxes from reducing sediments are not expected to reproduce the feature either (Johnson et al., 1996). This means that at least for those regions it would be logical to include a dependence on $[O_2]$. However, this is not the pattern that attracts the most attention when looking at the full transect. The striking patterns are rather the high concentration at the surface and near the equator around 2.5 km to 3 km depth (Fig. 7). For this reason we have not included a dependency on $[O_2]$ to the model. In other words, while $k_{ox}$ depends on oxygen, we assumed here that the oxygen concentration is generally so high that its variation does not affect $k_{ox}$.

In our model, oxidation takes place everywhere with the same $k_{ox} = 0.341 \times 10^{-3}\,\mathrm{h}^{-1}$. Based on the underestimation of the concentrations in the surface of the Atlantic and Pacific oceans, one may think that $[Mn_{ox}]$ is too high compared to $[Mn_{diss}]$. In other words, assuming approximate equilibrium, the ratio $k_{ox}/k_{red}$ could be too high. However, the $k$ values in our model are chosen to get values consistent with the ratio of concentration measurements of $Mn_{ox}$ and $Mn_{diss}$. To this end we set $k_{red,light}$ to the mean value found by Sunda and Huntsman (1994), then used the dissolved and particulate profiles in Fig. 4 of Bruland et al. (1994) to derive $k_{ox}$ and $k_{red,dark}$. If we were to ignore that ratio, and increase the first-order rate constants for reduction with a factor of five (a factor of three higher than the upper value of the range found by Sunda and Huntsman (1994)), the model would still not yield sufficiently high $[Mn_{diss}]$ (results not presented).

Still, one may introduce a threshold on the oxidation process (instead of the settling of the particles). The purpose would be to find out if this would result in higher surface $[Mn_{diss}]$, without having a significant effect on the ratio between the dissolved manganese and the oxides. In such a model simulation, oxidation only takes place when $[Mn_{diss}]$ is higher than a certain threshold $\mathcal{D}_{thr}$, equal to 0.125 nM in *OxidThreshold*. ~~Analogous to Eq. 10, $k_{ox}$ is multiplied by $H(\mathcal{D} - \mathcal{D}_{thr})$.~~ While this yields a distribution of $Mn_{diss}$ not significantly worse in the West Atlantic Ocean than that in *Reference*, the $Mn_{ox}/Mn_{diss}$ concentration ratio around 500 m and below is then strongly underestimated compared to the VERTEX data (Fig. 16, right panel, red dashed line). While $[Mn_{diss}]$ stays at realistic values, $[Mn_{ox}]$ does not and approaches zero. This pleads for using an aggregation rather than an oxidation threshold.

## 4.4 Export dynamics

The very homogeneous low dissolved concentration of Mn at $\sim$0.1 nM in the deep waters of the West Atlantic GA02 section is remarkable (Fig. 8). Such deep water values of $\sim$0.1 nM were also found in the deep waters of the Antarctic Ocean (Middag et al., 2011a, 2012, 2013), as well as deep waters of the Mediterranean Sea (2013 cruise, data will be included in the GEOTRACES Intermediate Data Product 2017). For the Atlantic Ocean, these deep concentrations are much lower than the dissolved Mn values of $\sim$0.5 nM to 0.6 nM reported for deep samples at several Atlantic stations in the 1990s (Saager et al., 1997; Statham and Burton, 1986; Statham et al., 1998). The sampling resolution is too coarse to exclude spatial variation, but it appears the values produced by pioneering efforts to study the oceanic Mn distributions are overestimations.

The new ultraclean sampling methods (Rijkenberg et al., 2015), rigorous calibrations and excellent accuracy at the Bermuda crossover station for twelve trace metals (Middag et al., 2015b) now yield a much larger database of uniformly very low Mn $\sim$0.1 nM in deep waters. Given these very uniform and much lower background Mn concentrations, the hydrothermal Mn

plumes are better discernible and much larger widespread than realised in the 1990s. Exactly over or very near the ridge crest
30 this is recently shown dramatically in a plume of more than $500\,\mathrm{km}$ wide over the Gakkel Ridge in the high-latitude Arctic
Ocean (Middag et al., 2011b), and in the Antarctic Ocean with a plume that extends more than 1000 km, at a site near the
Bouvet triple junction where three ocean ridges meet (Middag et al., 2011a). Similar long range extent of the Mn hydrothermal
plume has recently been discerned in the Pacific Ocean (Resing et al., 2015).

Although the West Atlantic GA02 section is quite far west from the Mid-Atlantic Ridge, the impact of the hydrothermal
plume is still visible. Dissolved Mn reaches a maximum of $0.2\,\mathrm{nM}$ to $0.3\,\mathrm{nM}$ in the $2500\,\mathrm{m}$ to $3000\,\mathrm{m}$ depth range just
south of the equator. This is consistent with observations along two zonal (east-west) sections across the Mid-Atlantic Ridge.
These are (i) the part of the GEOTRACES GA03 section that passes over the TAG hydrothermal site (USGT11-16 at 26.14° N,
44.83° W) (Wu et al., 2014), and (ii) at $\sim$12° S in the South Atlantic Ocean (Noble et al., 2012, their Fig. 1). In both sections
a plume of dissolved Mn is visible that extends for $500\,\mathrm{km}$ in both zonal directions.

Remarkably, the background concentration of dissolved Mn is quite uniform in the 0.10 to 0.15 nM range. Preliminary
equilibrium calculations predict an equilibrium of 0.19 nM total dissolved Mn with the solid phase pyrochroite $\mathrm{Mn(OH)_2\,(s)}$.
This total dissolved equilibrium concentration is quite close to the observed 0.10 to 0.15 nM background range measured in
the oceans. Please notice that in pyrochroite the Mn is divalent Mn(II) just as the dissolved Mn forms. Disregarding organic
complexes, the latter dissolved equilibrium Mn(II) forms would comprise 0.148 nM (78 %) free $\mathrm{Mn^{2+}}$ ion, 0.027 nM ($\sim 14\,\%$)
$\mathrm{MnCl^+}$ ion, 0.011 nM ($\sim 5.8\,\%$) $\mathrm{MnCl_2^0\,(aq)}$ and a very low abundance 0.000028 nM ($\sim 0.0015\,\%$) $\mathrm{MnOH^+}$ ion. The above
most simple Reaction R1 for oxidation removal from seawater, may perhaps be envisioned to be split into several reaction steps
as follows:

$$\mathrm{Mn^{2+}} \underset{\text{equil.}}{\leftrightharpoons} \mathrm{MnOH^+\,(aq)} \underset{\text{slow}}{\longrightarrow} \mathrm{Mn(OH)_2\,(s)} \underset{\text{fast}}{\longrightarrow} \mathrm{Mn(III,IV) \text{solid state}(s)}, \tag{R2}$$

where the dominant (78 %) dissolved $\mathrm{Mn^{2+}}$ species is via the minor dissolved $\mathrm{MnOH^+}$ species converted to solid $\mathrm{Mn(OH)_2}$,
and next this is oxidised to one or several of various oxide forms, for example hausmannite $\mathrm{Mn_3O_4}$ that is a mixed valency
state (III, IV) Mn-oxide, or $\mathrm{Mn(IV)O_2}$ pyrolusite or birnessite. Please notice that in the *Reference* simulation an aggregation
threshold for particulate Mn at 25 pM was invoked, that via redox rate constants is based on typical background dissolved Mn
concentrations that was chosen at 0.125 nM. The latter threshold value of 0.125 nM was required to have the model simulate
the observed data, and is quite close to the equilibrium concentration of 0.19 nM versus pyrochroite. The notion of perhaps
an equilibrium control is not new. Previous pioneering measurements led to an apparent background concentration of Mn of
0.4 nM or higher, that is now shown to be too high, yet otherwise attempts were made to compare these values in terms of
equilibrium versus hausmannite $\mathrm{Mn_3O_4}$ (Klinkhammer and Bender, 1980). Von Langen et al. (1997) did propose a reaction
mechanism also involving conversion of $\mathrm{Mn^{2+}}$ via $\mathrm{Mn(OH)^+}$ as in above (R2). Further research is needed to verify and
25 confirm the here discussed hypothesis of equilibrium of dissolved Mn at $\sim 0.19$ nM versus pyrochroite.

In the *NoThreshold* simulation $[\mathrm{Mn_{diss}}]$ is underestimated in much of the ocean. When removing the threshold, the RMSD
significantly worsens: it appears that the threshold is needed to reasonably simulate $[\mathrm{Mn_{diss}}]$. But there are several alternatives
to the aggregation threshold that may keep $\mathrm{Mn_{diss}}$ at a relatively homogeneous concentration.

Firstly, instead of an aggregation threshold, a criterion on oxidation may be imposed, because, hypothetically, there may
be a lower limit below which oxidising microbes would not proliferate and the pseudo-first-order rate constant would then be lower. The simulation with an oxidation threshold based on this idea gives similar $[\mathrm{Mn_{diss}}]$ as the aggregation threshold; but as we saw in the previous section, it would influence significantly the $\mathrm{Mn_{ox}}/\mathrm{Mn_{diss}}$ concentration ratio in the deep ocean, making for a less veracious simulation. Therefore, this alternative should probably not be pursued.

Secondly, the process of oxidation is strongly mediated by adsorption onto particles, after which they form larger aggregates that settle faster. This is now parameterised by increasing the settling velocity of particles with depth. However, a more explicit model for adsorption/desorption could be useful. Possibly, much of the $\mathrm{Mn_{ox}}$ does not settle efficiently. In that case, another tracer of adsorbed Mn is needed that would export oxidised Mn (in adsorbed or aggregated form) but not too fast to account for the background $[\mathrm{Mn_{diss}}]$. The adsorption should not take place onto a homogeneous pool of particles as is effectively done in our
model, but rather onto a component, for example calcium carbonate ($\mathrm{CaCO_3}$) (Martin and Knauer, 1983; Fraústo da Silva and Williams, 2001) and lithogenic particles (Roy-Barman, 2009). Sediment samples show a strong correlation between authigenic manganese and lithogenic particles, though Mn does not show any correlation with biogenic silica, POC or $\mathrm{CaCO_3}$ according to recent observations and modelling (Roy-Barman, 2009). On the one hand, this suggests that only lithogenic particles are a scavenger of Mn. On the other hand, we do not know whether the correlation comes from Mn adsorption onto lithogenic
particles, or if it is lithogenic in itself.

Thirdly, oxidation is mediated by microbes (mainly bacteria) (Sunda and Huntsman, 1994; Von Langen et al., 1997; Tebo et al., 2005; Fraústo da Silva and Williams, 2001). Possibly they prolong the time of $\mathrm{Mn_{ox}}$ particles spent in the euphotic zone, but this has not been tested. In the model we assume that these processes are included in the redox rate constant. If we want the redox rate constants to depend on inhomogeneously distributed quantities like the bacterial distribution or the oxygen
concentration, the equations need to be modified, and our model may need to be extended. Subsequently, more Mn would stay suspended.

Fourthly, it has been suggested that dissolved Mn ligands keep Mn in solution (Sander and Koschinsky, 2011; Madison et al., 2013; Luther et al., 2015). That would result in less conversion to manganese oxide aggregates, and hence may solve the removal problem, but this would yield wrong $\mathrm{Mn_{ox}}/\mathrm{Mn_{diss}}$ concentration ratios like we saw for *OxidThreshold*. Similarly,
nanoparticles, which have a size that falls within the operationally defined dissolved Mn, keep Mn afloat as long as they do not aggregate.

### 4.4.1 Hydrothermal activity

In the model, hydrothermal $\mathrm{Mn_{diss}}$ is added to the ocean based on a model proxy of $^{3}\mathrm{He}$ (Dutay et al., 2004). For the *Reference* simulation the hydrothermal $\mathrm{Mn_{diss}}$ flux was set to $0.10 \times 10^{9}$ mole Mn for each mole of $^{3}\mathrm{He}$. There is no reference to literature
values in Table 3 since these are very uncertain. For instance, Klinkhammer and Bender (1980) note that "Corliss and Dymond (1979) reported that the $\mathrm{Mn} : ^{3}\mathrm{He}$ ratios vary by over a factor of 2 in the Galapagos vents, and as we discuss below, similar variations are found at $21^{\circ}\,\mathrm{N}$". We have chosen a value to yield an acceptable distribution of $\mathrm{Mn_{diss}}$.

The goal of *LowHydro* is to investigate whether the combination of the high hydrothermal input of $Mn_{diss}$, and strong aggregation is needed to get a more veracious simulation of the distribution of $Mn_{diss}$, namely that of *Reference*. To this end, we chose to decrease hydrothermal input and also to decrease the settling velocity in *LowHydro*. In *LowHydro* the spatial distribution of $[Mn_{diss}]$ becomes more homogeneous. Neither the low background concentration nor the high hydrothermal vent concentrations are reproduced any more (Fig. 14). Decreasing hydrothermal input and the settling velocity by a smaller factor than 10 results in a $Mn_{diss}$ distribution that is better than *LowHydro* but not as good as in *Reference* (not presented). In any case, the main result is that a high hydrothermal input very similar to or higher than that used in the *Reference* simulation is needed to reproduce $[Mn_{diss}]$.

According to observations, a negligible amount of Mn from hydrothermal vents reaches the surface ocean. As Bruland and Lohan (2006) put it: "The hydrothermal input of iron and manganese [...] is essentially all scavenged and removed in the deep sea prior to having a chance to mix back into the surface waters." However, this is what we have thought about iron while there are now doubts about this (Tagliabue et al., 2010). Similarly, hydrothermal Mn may not be completely removed from the ocean before reaching the surface by currents and vertical mixing. There are different potential reasons for this. ~~First, t~~The oxidation of Mn(II) is thermodynamically favoured, but the large activation energy of Mn(II) oxidation renders Mn(II) stable in aquatic environments (Nealson, 2006). Other potential reasons for $Mn_{diss}$ stability are already given on the previous page. ~~In our model we assumed that the activation energy is not an issue. Furthermore, the background "dissolved" Mn particles are not $Mn^{2+}$ ions but rather suspended colloids or complexes. Candidates include colloidal $Mn_{ox}$ and Mn bound to ligands, respectively the second and third potential reasons for the observed background "$Mn_{diss}$" concentration.~~ The low $Mn_{diss}$ concentrations away from hydrothermal vents are established in our model by removing $Mn_{ox}$ from the high hydrothermal Mn input down to 25 pM.

Interestingly, in this way, in *Reference*, hydrothermal flux accounts for 92 % of the total Mn input. Dust deposition is 18 times and sediment flux 31 times as small as hydrothermal input, and still $[Mn_{diss}]$ in the upper 1000 m of the ocean is dominated by $Mn_{diss}$ release from marginal sediments and dissolution from deposited dust. Nonetheless, some of the hydrothermally derived $Mn_{diss}$ reaches the ocean surface. In our model, 8 % of Mn in the surface is hydrothermal of origin. The reason for the importance of sediment and dust is that in the model the settling velocity of $Mn_{ox}$ is much higher near hydrothermal vents (up to $10\,\mathrm{m\,d^{-1}}$) than in the surface ocean ($w_s = 1\,\mathrm{m\,d^{-1}}$). The modelling study by Lavelle et al. (1992) suggests a settling velocity much higher than our $w_{ox}$. They used a model with an additional tracer of large particles (aggregate products) that had settling speeds of up to $175\,\mathrm{m\,d^{-1}}$ (near hydrothermal vents, as they model those regions). This shows that using an increasing settling velocity is justified. Moreover, it suggests that at least two particle tracers are needed in the model. It is worth investigating in future studies if this would make for a more veracious and reliable model.

## 5  Conclusions

This is the first study in which the 3-D distribution of $Mn_{diss}$ has been modelled and compared to the recent observations from the GEOTRACES programme. A combination of photoreduction and Mn sources to the upper ocean yields high surface

concentrations of dissolved manganese. However, the concentration is at several locations underestimated by the model. The Mn sources to the ocean surface and the sinks are uncertain and could therefore be adjusted for a more veracious simulation.

The most important sources for the upper ocean are sediments, dust, and, more locally, rivers, whereas hydrothermal vents are the most important in the deep ocean. The observed sharp hydrothermal signals are produced by assuming both a strong source and a strong removal of Mn near hydrothermal vents. Our model further shows that the Mn at the surface in the Atlantic Ocean moves downwards into the North Atlantic Deep Water, but because of strong removal the Mn signal does not propagate southwards.

    There is a mainly homogeneous background concentration of dissolved Mn of about $0.10$ nM to $0.15$ nM throughout most of the deep ocean. Our model reproduces this by means of a threshold on solid manganese oxides of $25$ pM, suggesting that a minimal concentration of particulate Mn is needed before aggregation and removal become efficient. An aggregation threshold,

as applied in our model, appears reasonable, and does not affect the modelled $Mn_{ox}/Mn_{diss}$ concentration ratio. An oxidation threshold is more troublesome as it affects this ratio in an undesirable way. Since the settling condition as a sole threshold already appears to mitigate most of the removal, it is reasonable to further develop the model with the aggregation threshold and without the oxidation threshold. Still, the simplified redox and subsequent settling of $Mn_{ox}$ is incomplete and could be too imprecise, as future studies might show.

Our model includes biology, and this has a big impact in the Pacific Ocean. However, we have at the moment no clear evidence for typical uptake-remineralisation processes as for iron. At the same time, we are missing dominant sources from anoxic sediments, especially in upwelling regions like at the East Pacific boundary. Those sources, once adequately represented in the simulation, might be large enough to compensate for the effect of our biological Mn uptake at the surface. Such a source, and possibly biological incorporation, should be improved in future versions of the model. Besides these processes, it may also

be needed to model microbial activity throughout the ocean, as that is not homogeneously distributed. ~~Our chemical first-order reaction may be insufficient for many modelling purposes.~~ The oceanwide chemical first-order reaction may be inadequate to represent such microbial activity in certain regions.

    Hydrothermal fluxes of Mn were set to such a high rate that we must assume $96\,\%$ is scavenged near the outflow of the vents. This choice was made to account for the local high $Mn_{diss}$ near the oceanic ridges. This was combined with a high $w_{ox}$

in the deep ocean to prevent hydrothermal $Mn_{diss}$ spreading far away from the oceanic ridges in a too high concentration. As an alternative, settling may not need to be set as high as $10\ \mathrm{m\,d^{-1}}$ in the deep ocean, but then it is unclear how to simulate the local nature of $[Mn_{diss}]$ anomalies near the ocean ridges. One possibility is to include extra species of Mn, which may include a very fast sinking particle and a very reactive Mn species.

    Process studies of Mn are necessary to determine the rate constants, and possibly thresholds, for redox, scavenging and

aggregation. More measurements of particulate $Mn_{ox}$ concentrations would be useful as well. When measuring particulate Mn, it needs to be clear what is measured exactly. It would for instance be useful if (1) a range of particle sizes were measured, and (2) a structural analysis of the particles were performed, such that one can unambiguously say onto which particle Mn were adsorbed or into which particle it were incorporated.

## Appendix A: Data–model comparison

We compare quantitatively our model results with the observations at the GA02 GEOTRACES transect in the West Atlantic Ocean. First the model output is horizontally interpolated onto the station coordinates, keeping the vertical model grid (10 m at the surface up to 500 m thickness near the bottom). Then the modelled $[\mathrm{Mn_{diss}}]$ that lie closest to each of the observations are associated with each other. To be precise, for each observation, we take the shallowest gridbox whose upper bound lies deeper than the observation, after which residuals can be defined as $P_i - O_i$, where $O_i$ is the observed and $P_i$ the modelled $[\mathrm{Mn_{diss}}]$, for each $i \in \{1, ..., N\}$, with $N$ the number of data points. The interpolation introduces a representative error that is taken to be part of the residual (Van Hulten, 2014). Then several statistics are determined, namely the Root Mean Square Deviation (RMSD), the Reliability Index (RI) and the Pearson correlation coefficient $r$.

In addition to classical statistical indices (Pearson correlation index $r$, root mean square error RMS), another additional performance indicator has been used as suggested in previous skill assessment studies (Stow et al., 2009; Vichi and Masina, 2009): the reliability index (Leggett and Williams, 1981). The reliability index "quantifies the average factor by which model predictions differ from observations" (Stow et al., 2009). It is in essence the root-mean-square deviation, but it uses the logarithm of the residual. This is useful when both large and small values need to be considered (as for this case). The RI is given by

$$RI = \exp \sqrt{\frac{1}{N} \sum_{i=1}^{N} \left( \log \frac{O_i}{P_i} \right)^2}. \tag{A1}$$

where $O_i$ is the measured concentration with index $i$, $P_i$ is the model prediction associated to the respective observation $i$, and $N$ is the number of observations.

Furthermore, for each sensitivity simulation the significance of the change in each goodness-of-fit statistic compared with the corresponding reference simulation is calculated. This is determined by means of a Monte Carlo simulation on the reference simulation for which a subsample of 400 is randomly selected from the original set of ~~1440~~1320 data–model points. They are the pairs of observations and model output, both on the model grid. This is done 50 000 times, and from this the $2\sigma$ confidence interval is calculated (the mean $\pm$ two times the standard deviation). Suppose that we want to simulate $q$, and assume $q$ is in steady state. Given a reference simulation $X$, for any model simulation $Y$ resulting in $q_Y(\mathbf{x})$, the RMSD of $q_Y(\mathbf{x})$ must be outside the $2\sigma$ confidence range of the RMSD distribution of $q_X(\mathbf{x})$ to say that $Y$ is a significant improvement or worsening compared to $X$.

The statistics tell us how veracious our simulations are~~, but it does not tell how reliable they are in a wider sense (Petersen, 2006). For this reason,~~ Moreover, these statistics are illustrated with comparison plots and an extensive discussion of those. For the visual comparison between model and observations, horizontal and vertical cross-sections of the model data are presented. Using the same colour scale, observations are plotted as coloured dots to directly compare the model with the observations. Horizontal $[\mathrm{Mn_{diss}}]$ sections are presented for four different depths, where "surface" signifies the average over the upper 45 m, "500 m" is 400–600 m averaged, "2500 m" is 2100–2900 m averaged and "4500 m" is 4000–5000 m averaged. The colour scale is not linear to better show the main features at both low and high concentrations of $\mathrm{Mn_{diss}}$. The vertical $[\mathrm{Mn_{diss}}]$ sections are calculated from the 3-D model data by converting the ORCA2 gridded model data to a rectilinear mapping and interpolating the rectilinear data onto the cruise track coordinates.

## Appendix B: Code availability

We used NEMO 3.6 svn 7036 that can be downloaded through this command:

```
$ svn checkout -r 7036 http://forge.ipsl.jussieu.fr/nemo/svn/branches/2015/nemo_v3_6_STABLE
```

However, the latest stable version of NEMO is usually advisable. A login will be asked that you can create on the NEMO website.

The manganese module comes as part of different Fortran 90/95 source code files that are to be found in the electronic supplement. Those should be put in `NEMOGCM/CONFIG/*/MY_SRC/` that adds and overrides symlinks in `NEMOGCM/CONFIG/*/WORK/`. The asterisk must be replaced with your configuration name that should be based on `ORCA2_OFF_PISCES`.

30      Both NEMO and the manganese model are available under the CeCILL licence.

*Acknowledgements.* We would like to thank several people in particular who helped in different ways in this study. Angela Milne, William Landing and Joseph Resing kindly provided their manganese data from the Pacific Ocean. We thank Caroline Slomp, Catherine Jeandel and Micha Rijkenberg for the useful discussion. Loes Gerringa has kindly provided speciation calculations on manganese. Loes Gerringa and Micha Rijkenberg organised and served as expedition leaders of the three cruises of the GA02 section. This research was funded by

35      the NWO (grant 839.08.410; GEOTRACES, Global Change and Microbial Oceanography in the West Atlantic Ocean and grant 820.01.014 GEOTRACES Netherlands-USA Joint Effort on Trace Metals in the Atlantic Ocean). This study was partly supported by a Swedish Research Council grant (349-2012-6287) in the framework of the French-Swedish cooperation in the common research training programme in the climate, environment and energy agreement between VR and LSCE, for the project "Particle transport derived from isotope tracers and its impact on ocean biogeochemistry: a GEOTRACES project in the Arctic Ocean". The sampling and analysis of the data of Milne and Landing (unpublished data) were supported by NSF grants OCE-0223378, OCE-0550317 and OCE-0649639.

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

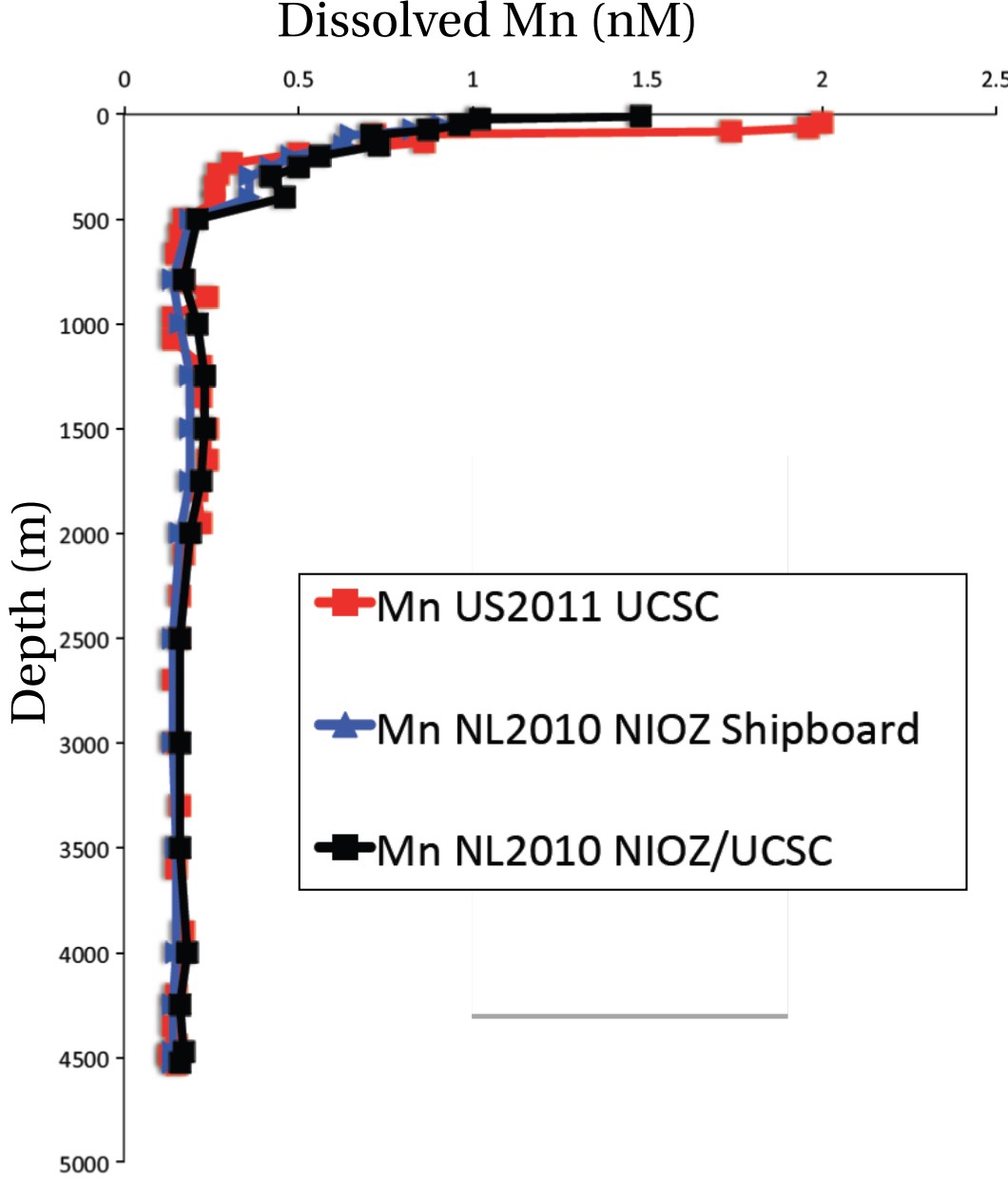

**Figure 1.** $[Mn_{diss}]$ at the 16 June 2010 (NL) and the 19 November 2011 (US) occupations of the crossover site of the Bermuda Atlantic Time Series (BATS) station. There is good agreement between the two sets of samples, as well as the two different analytical methods, the shipboard FIA by NIOZ for the NL2010 samples, and the lab ICP-MS at UCSC for both the NL2010 and the US2011 samples. Each lab had its own independently prepared lab standards, further confirming the overall accuracy. Analyses by Rob Middag at NIOZ in 2010 and in 2011 at UCSC.

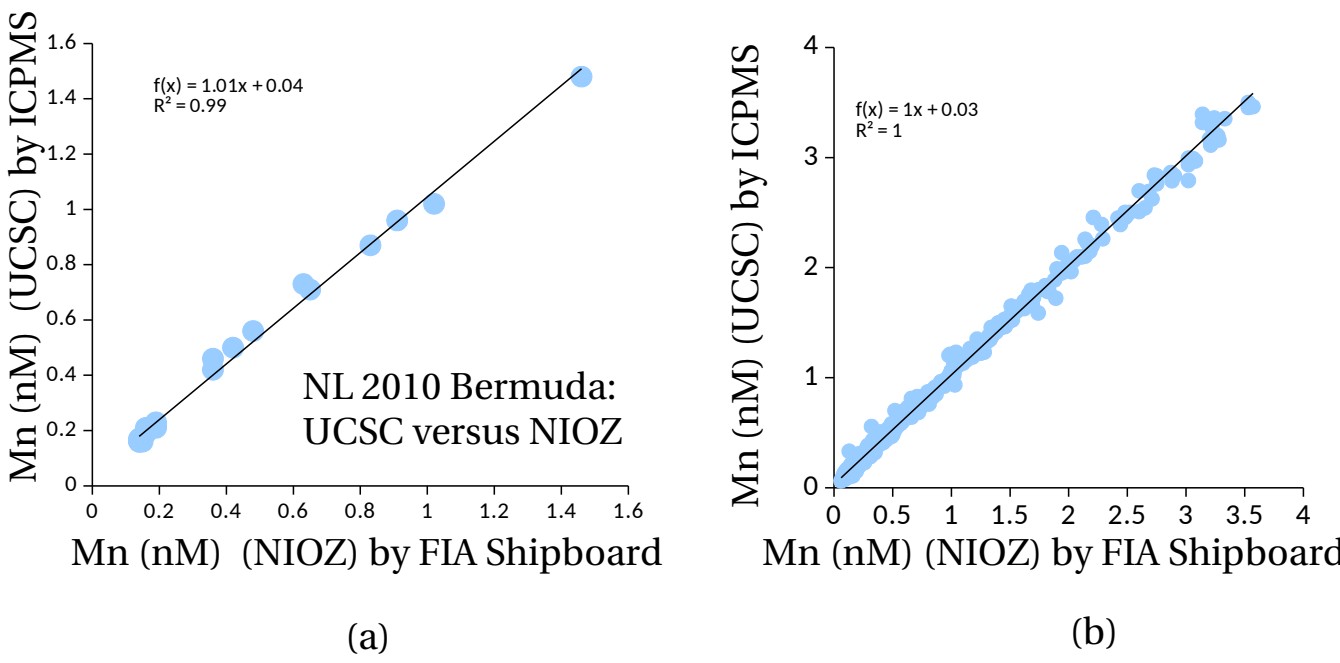

**Figure 2.** The correlation between the two methods of analysis for the determination of $[Mn_{diss}]$ at GA02 (shipboard and laboratory mass spectrometry): (a) at Bermuda; (b) all 55 West Atlantic stations.

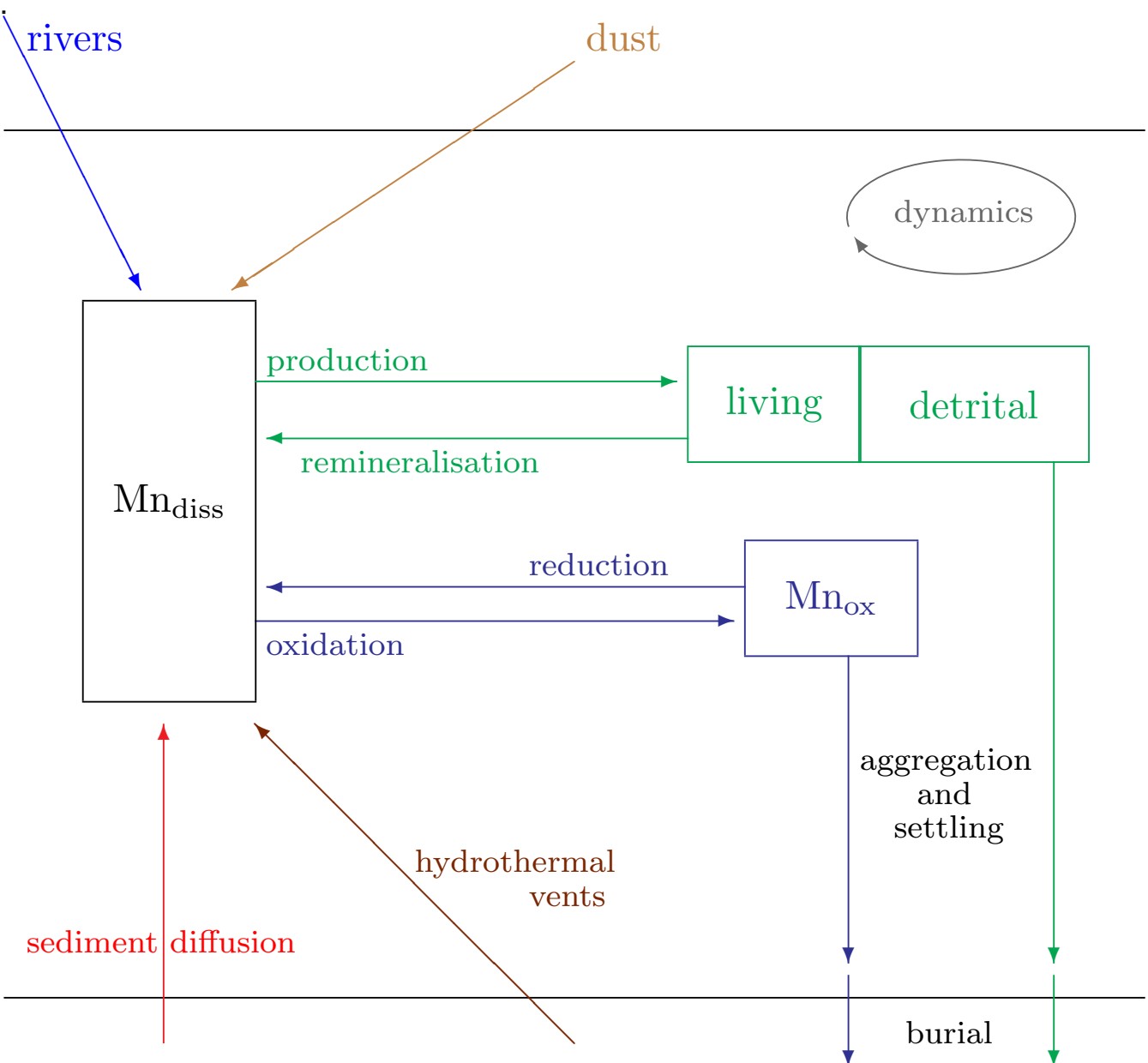

**Figure 3.** Model scheme: biology in green; redox and scavenging in blue; circulation and mixing (dynamics) in grey; and the sources dust, rivers, hydrothermal and sediment are in light brown, light blue, dark brown and red, respectively. $Mn_{diss}$ is the dissolved and $Mn_{ox}$ the oxidised Mn.

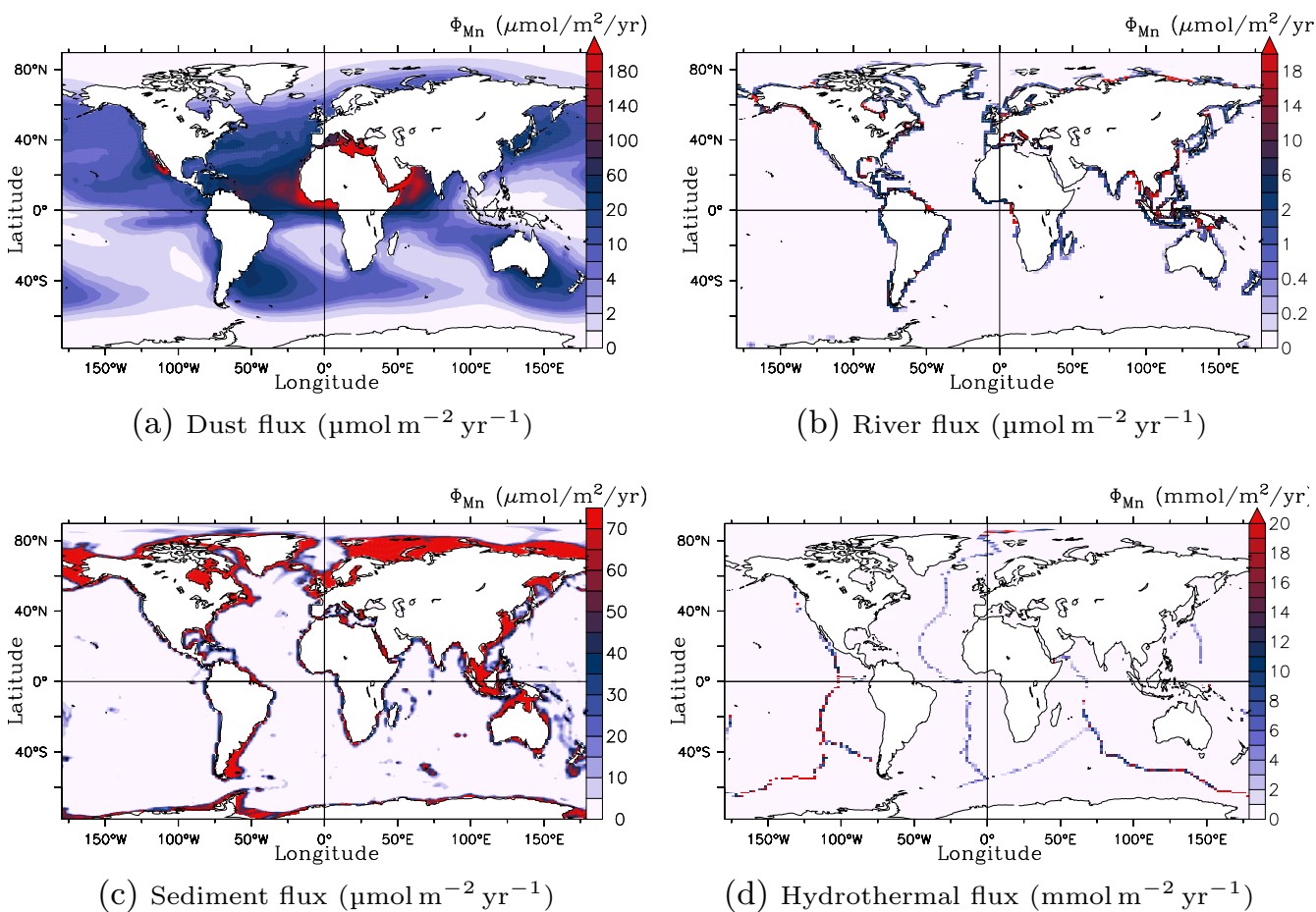

**Figure 4.** Sources of Mn to the ocean: effective $\mathrm{Mn_{diss}}$ input flux $\Phi$. Three-dimensional fields are vertically integrated, such that dimensions are molar fluxes. Ranges vary between the different sources.

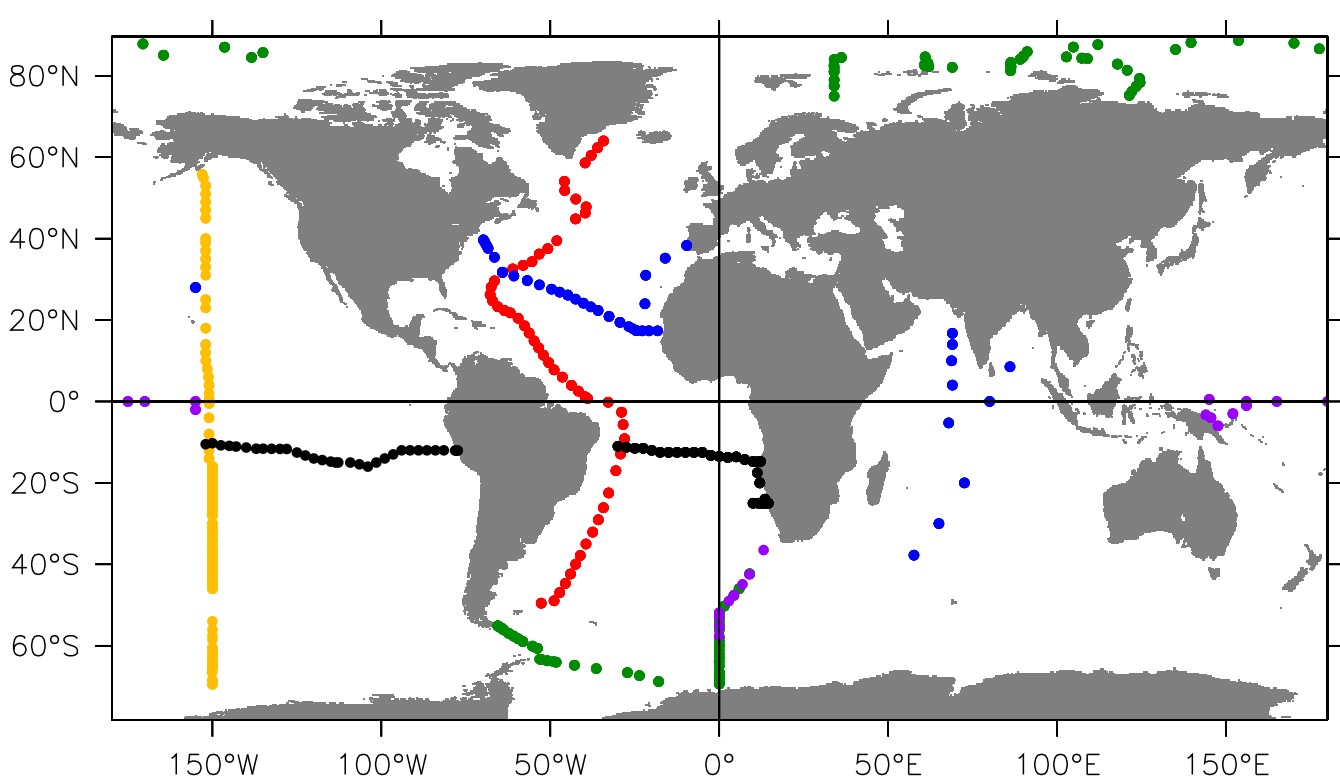

**Figure 5.** Transect (or expedition) names corresponding to station colours: GIPY11 in the Arctic Ocean; GIPY4 and GIPY5 in the Atlantic sector of the Southern Ocean; GI04 in the Indian Ocean; GA02 in the West Atlantic Ocean; GA03 in the North Atlantic Ocean; VERTEX-4 in the North Pacific Ocean; EUCFe in the equatorial Pacific Ocean; CLIVAR P16 in the Pacific Ocean; GP16 in the South Pacific; CoFeMUG in the South Atlantic Ocean. See Table 5 for an overview with references and the number of observations.

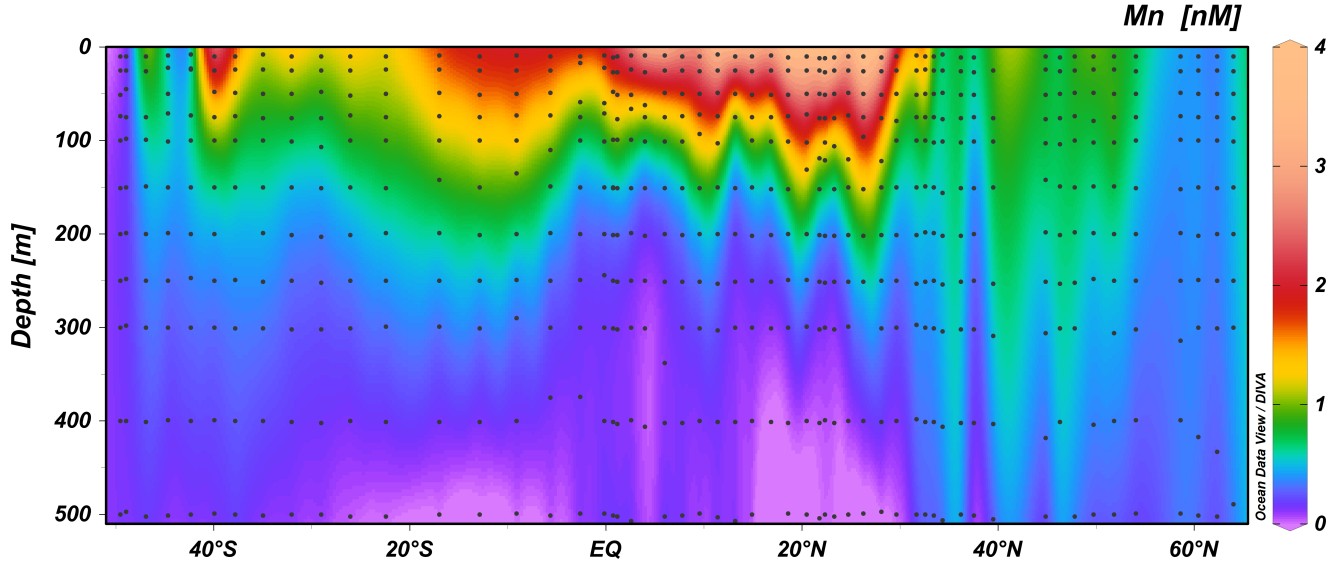

**Figure 6.** Observations of $[\mathrm{Mn_{diss}}]$ (nM) along the GEOTRACES GA02 transect in the West Atlantic Ocean. Dots are the locations of the measurements. Upper 500 m.

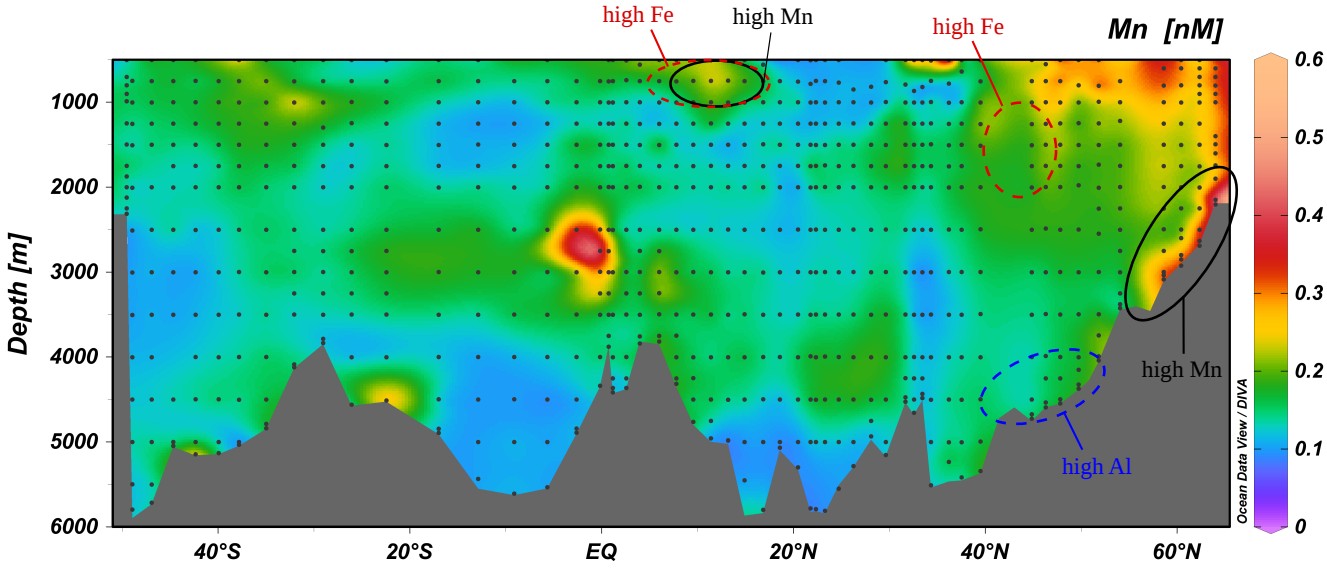

**Figure 7.** Observations of $[\mathrm{Mn_{diss}}]$ (nM) along the GEOTRACES GA02 transect in the West Atlantic Ocean. The dots denote the locations of the measurements. Below 500 m. The red, dashed lines indicate samples with high dissolved iron concentrations after Rijkenberg et al. (2014). Similarly, the blue, dashed lines represent high dissolved aluminium concentrations, and are after Middag et al. (2015a). Note that the concentrations are generally much smaller than in the surface, and that the colour scale is different from that of Fig. 6.

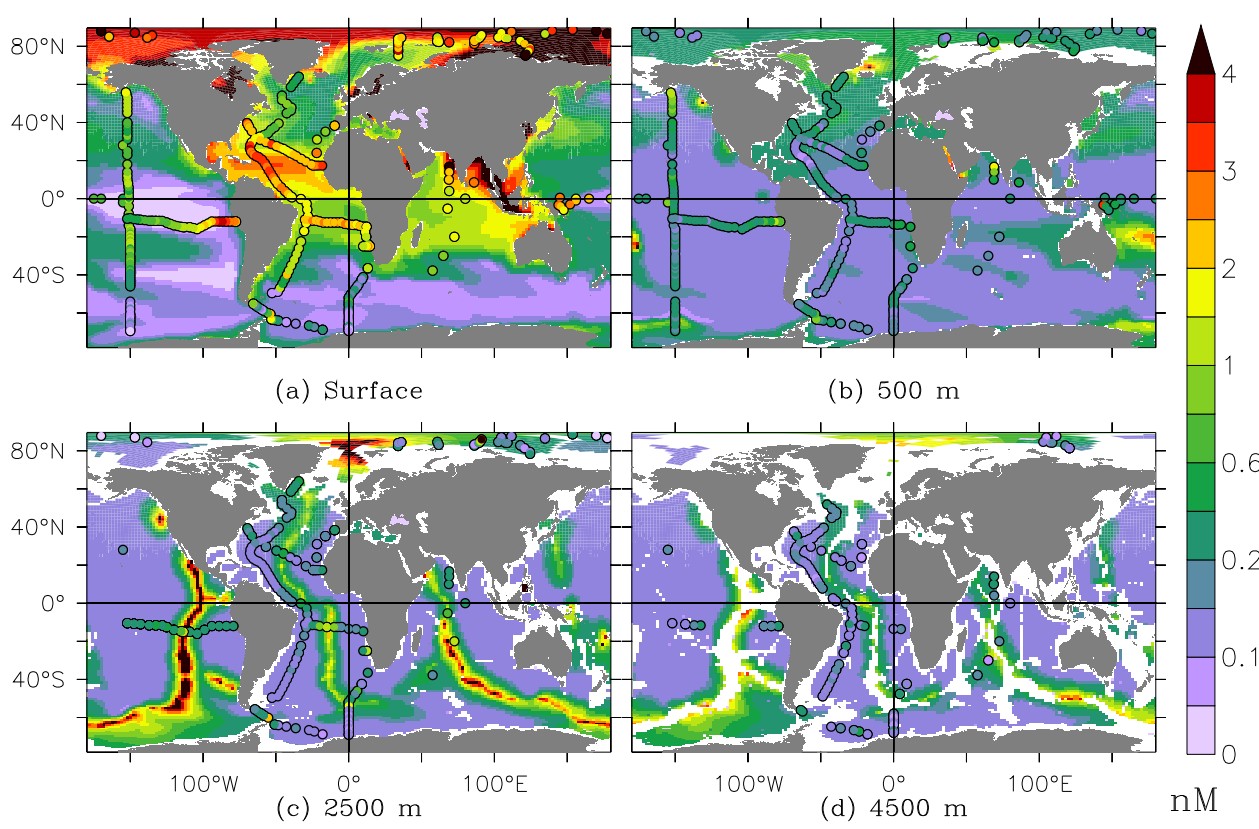

**Figure 8.** $[\mathrm{Mn_{diss}}]$ (nM) at four depth layers in the world ocean for the reference simulation (*Reference*) after 500 yr (annual average). Observations are presented as coloured dots; white is the land mask of the model grid.

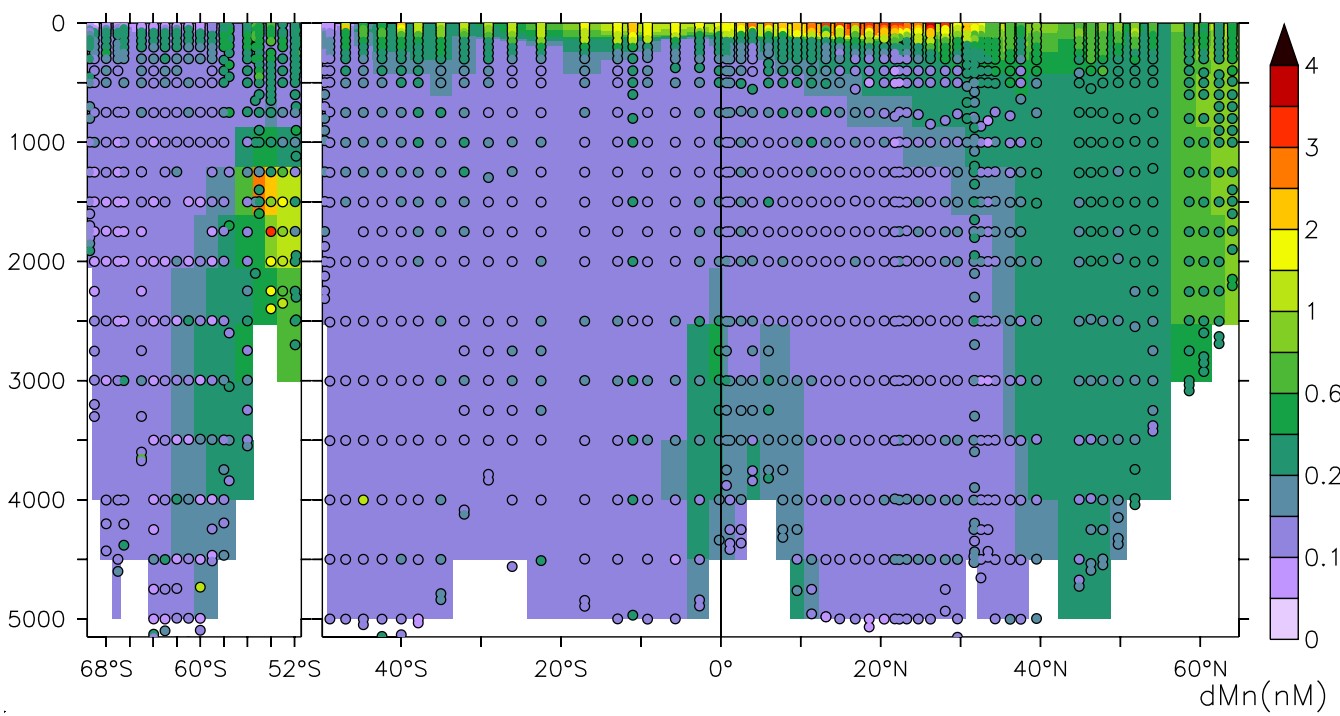

**Figure 9.** $[Mn_{diss}]$ (nM) at the Zero-Meridian section component of the GIPY5 dataset, and the West Atlantic GA02 GEOTRACES transects for *Reference* after 500 yr (annual average). Observations are presented as coloured dots.

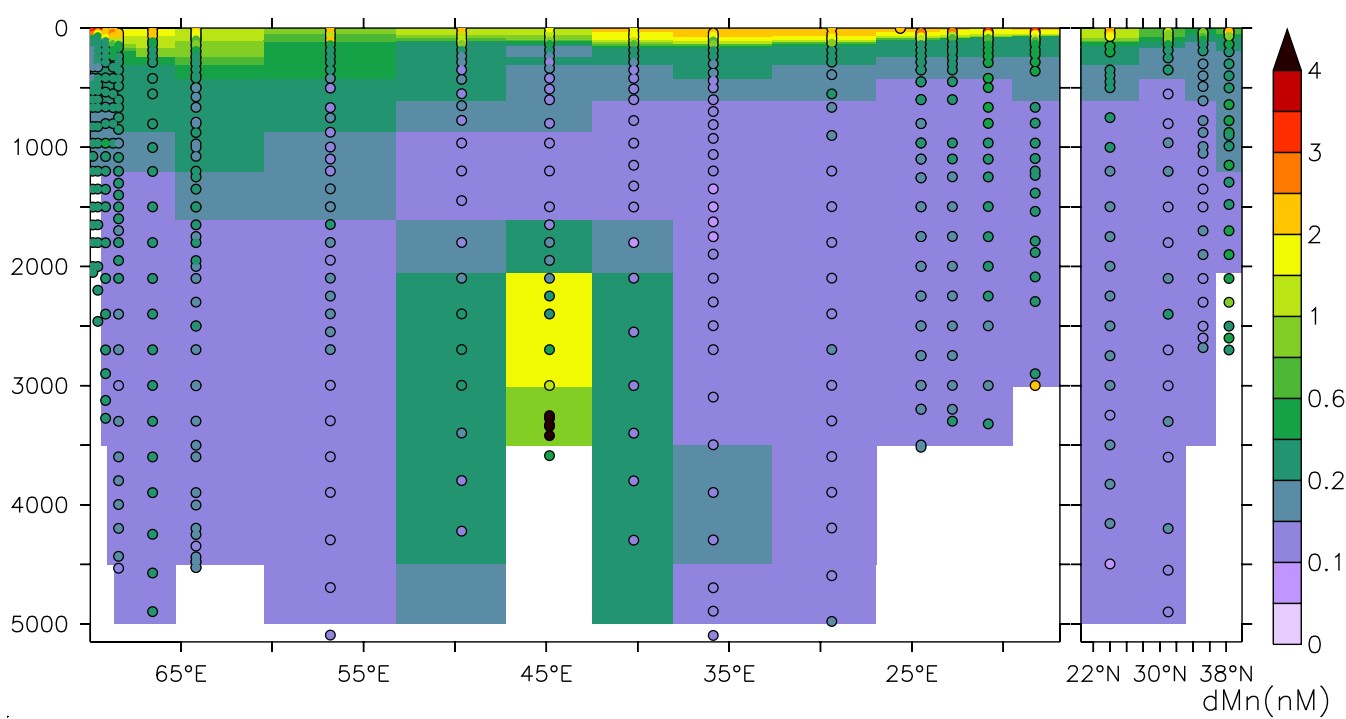

**Figure 10.** [Mn_diss] (nM) at the North Atlantic GA03 GEOTRACES transect for *Reference* after 600 yr (annual average). Observations are presented as coloured dots.

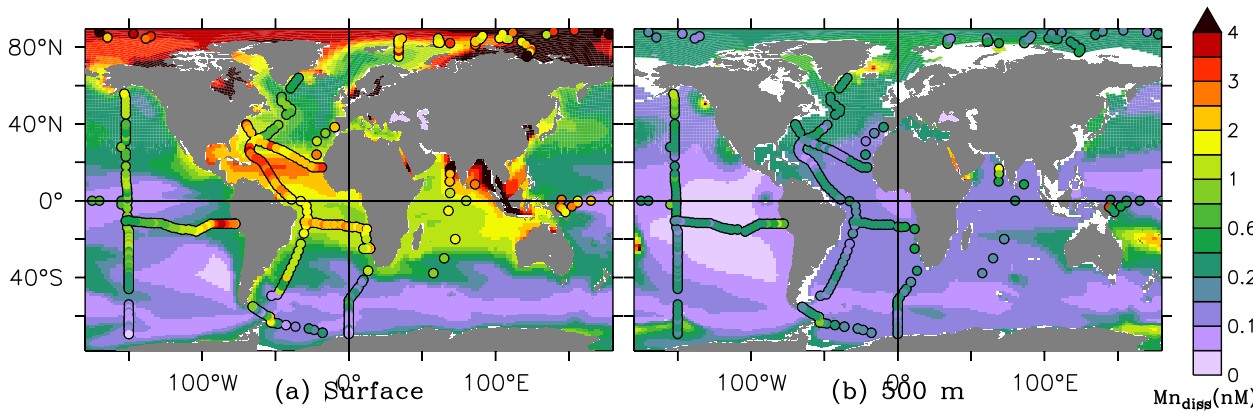

**Figure 11.** The effect of biological incorporation on $[\mathrm{Mn_{diss}}]$ at four depths: $(Reference - NoBio)/NoBio$. Grey is the land mask of the model domain; the black contour is that of the real continental coasts.

**Figure 12.** Annual average of $[\mathrm{Mn_{diss}}]$ (nM) at two depth in the world ocean for the simulation without a Mn biological cycle (*NoBio*) of year 500 after forking from *Reference*.

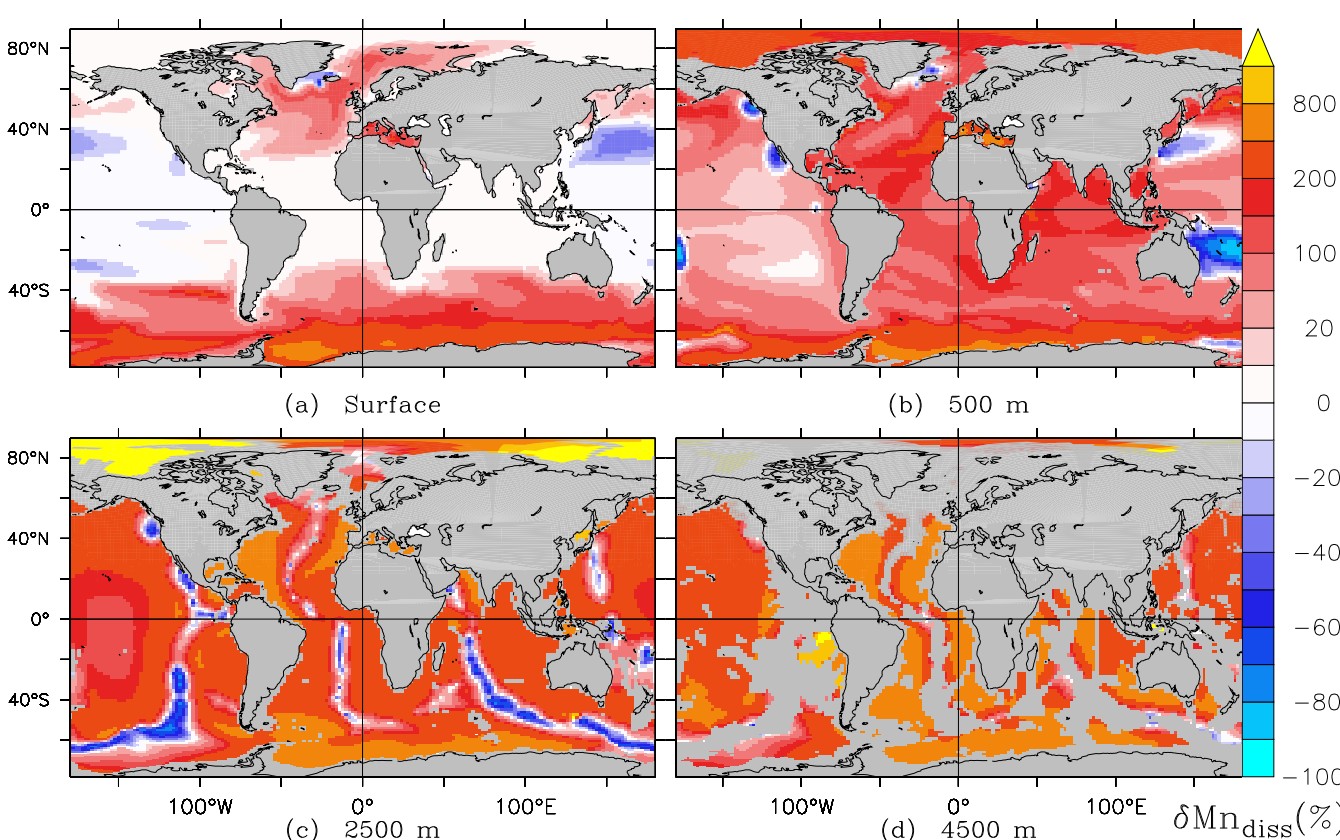

**Figure 13.** Result from the simulation with hydrothermal vent Mn input decreased and settling velocity decreased (*LowHydro*). Relative difference in $[\mathrm{Mn_{diss}}]$ (%) between *Reference* and *LowHydro*: (*LowHydro–Reference*) / *Reference*. To represent the changes of much larger than 100 %, the scale is increased from +50 % upwards.

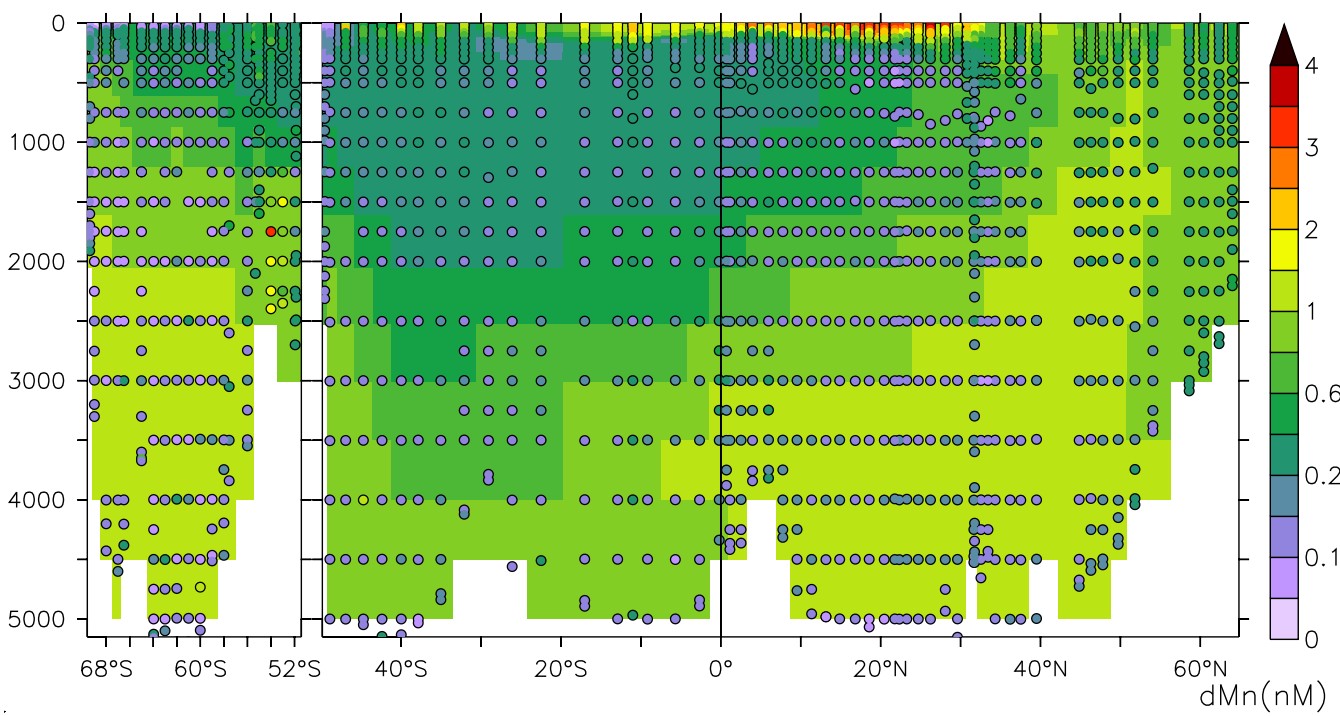

**Figure 14.** $[\mathrm{Mn_{diss}}]$ (nM) from the simulation with hydrothermal vent Mn input decreased and settling velocity decreased (*LowHydro*) at the West Atlantic GA02 and at the Southern Ocean Zero-Meridian part of the GIPY5 GEOTRACES transects.

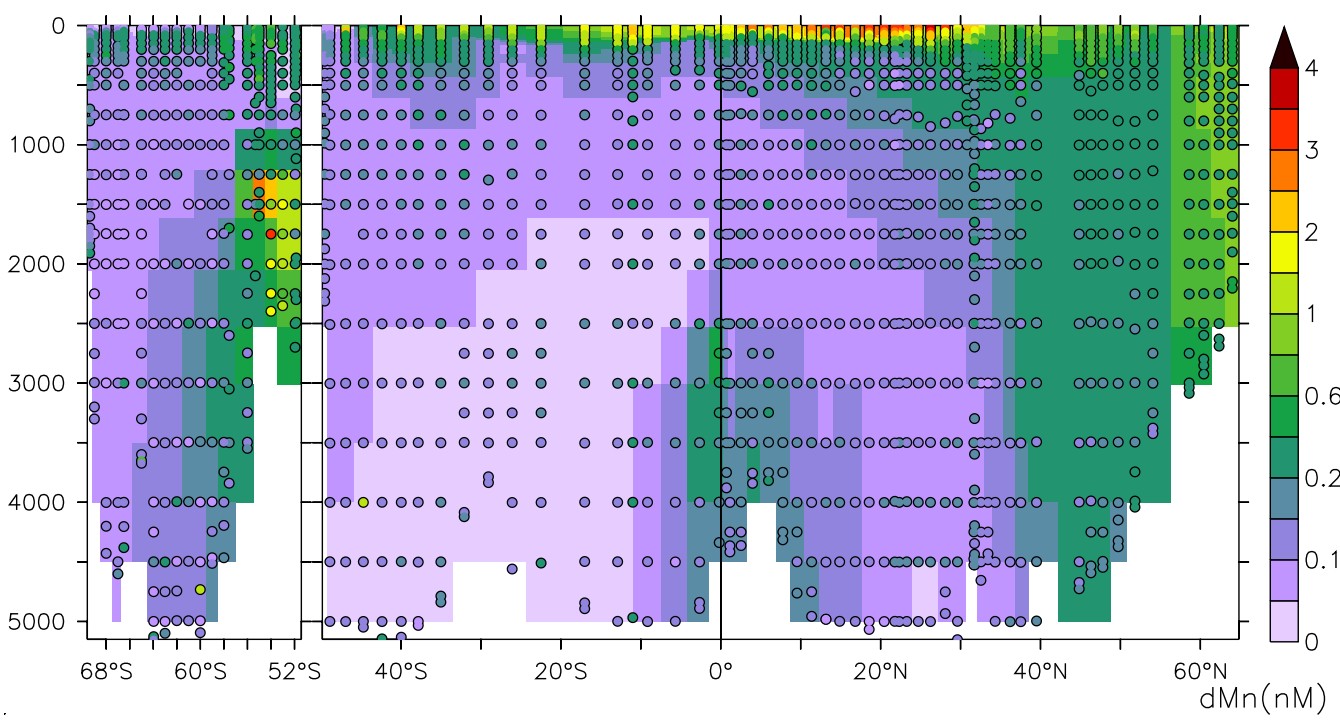

**Figure 15.** $[\mathrm{Mn_{diss}}]$ (nM) at the West Atlantic GA02 and at the Southern Ocean Zero Meridian part of the GIPY5 GEOTRACES transects. Simulation without threshold (*NoThreshold*).

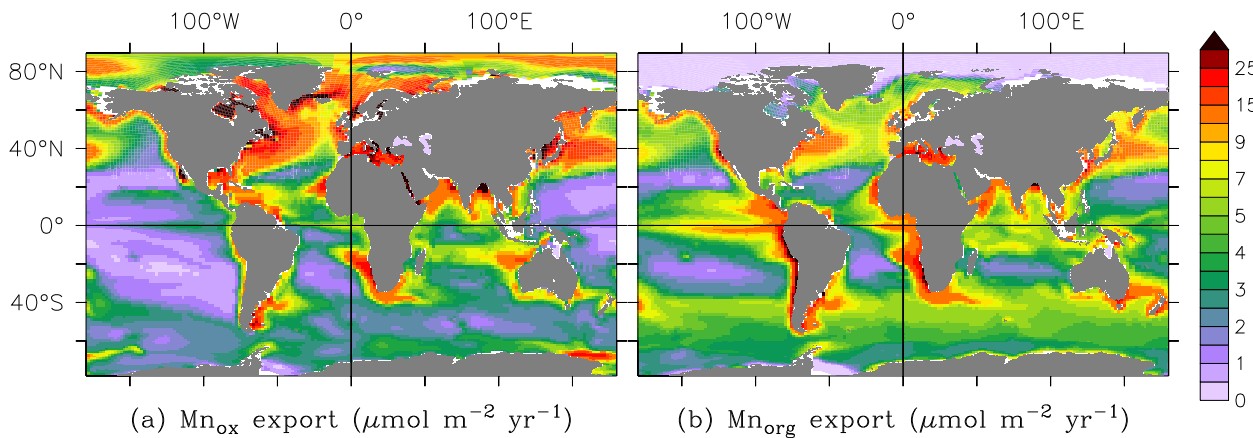

**Figure 16.** $[Mn_{ox}]/[Mn_{diss}]$ in the Pacific Ocean at the VERTEX-IV station in the upper 110 m (left) and at full depth (right). Blue squares are observations, the green line is the *Reference* simulation, and the red dashed line is the ratio from *OxidThreshold*.

(a) $Mn_{ox}$ export ($\mu$mol m$^{-2}$ yr$^{-1}$)

(b) $Mn_{org}$ export ($\mu$mol m$^{-2}$ yr$^{-1}$)

**Figure 17.** Manganese fluxes at 100 m depth ($\mu$mol m$^{-2}$ yr$^{-1}$): (a) oxidised Mn, and (b) biologically incorporated Mn.