# Peer review of "Manganese in the West Atlantic Ocean in context of the first global ocean circulation model of manganese"

_Biogeosciences, 2016_

## Referee Comment (RC1) · Anonymous Referee #1 · 4 Aug 2016

The threshold Mnox that is used to account for the homogeneous background concentration of dissolved Mn of about 0.10 nM to 0.15 nM observed throughout most of the deep ocean, may result from over simplification of the model. There might be very different Kox and Kred values at upper (above ∼300 m) and deeper (below ∼300 m) part of the ocean. Mndiss may be mainly derived from remineralization of sinking organic matter in the upper ocean, and from an equilibration with colloidal or fine particles via absorption/colloid formation processes in the deeper ocean. One would expect very different kred values at different depths. In addition, Mndiss may be mainly removed from the water column via oxidation to insoluble Mn(IV) with a rate that decreases with increasing depth due to lower dissolved oxygen concentration and lower pH at deeper depths, leading to very different kox values at these different depths. Thus both k values and their ratios Kred/kox are not homogeneous throughout the water column. Such

difference may cause lower modelled Mndiss than the observed values, thus requiring a threshold Mnox to account for higher Mndiss at deeper depths.

---

## Referee Comment (RC2) · Anonymous Referee #2 · 11 Aug 2016

Modelling Mn in the ocean is of great interest and is very timely. The Mn cycle is very complex and modelling it is not easy. This study presents the first results from a few model simulations, one of which the authors argue is in good agreement with the few dissolved Mn concentration measurements available.

While I am very supportive of the objectives of this work in general, the scope of the discussion, the wording and overall presentation of this manuscript gives the impression that the model is not sufficiently mature: at this stage of development, this paper reads more like a progress report than a fine-tuned, well-thought-out, scientific manuscript. The story in this manuscript announces a great start for Mn modelling, but comes across a bit short. Specifically, excluding the biological aspect of Mn cycling is a major omission (the authors themselves acknowledge that this is a major aspect of Mn cycling).

[Figure]

As a reader, it is not totally clear what has been learned about the Mn cycle by reading this manuscript. Interesting points are made in the last few lines of the abstract, but these are unfortunately not developed fully in the text. Most of the discussion focuses instead on technical limitations of the model and on a handful of parameterizations used to modify the model to make it fit data – with variable success. Since the bulk of the model presented here is essentially a scaled version of the existing companion Fe-model, the scope of the discussion presented here is somewhat limited.

One aspect of the writing that is not helping convincing the reader that this is a conclusive piece of work, is the over-reliance of subjective statements to describe the simulations and on imprecise/loose statements. This undermines the results. For example, the model fields are repeatedly qualified as "realistic", but this really doesn't mean anything, as there are just as many paragraphs in the paper outlining model flaws. The figures also reveal quite a few regions where the model does not fit the data. There is also no real set of criteria that are presented to define what "realistic" means. The authors should instead define the features their model fits or doesn't fit directly, without trying to oversell their product as "realistic", and explain in a bit more detail why the model fits or doesn't fit particular features. Other examples are: "The model reproduces observations accurately", "This is mostly captured by the model", "It is probably difficult to improve this feature of the model", "Reference simulation gives a reasonably realistic distribution", "LowHydro appears to give a much worse prediction", "This first-order approach works well mostly, but shortcomings can be identified", "the flux may be overall underestimated", "this may still contribute to the dissolved pool in the bottom water", "underestimation may be due to too strong vertical mixing", etc... Some of these propositions could be tested explicitly by the authors using additional sensitivity studies or appropriate diagnostics to produce objective conclusions instead of suggestions.

One way forward could also be to develop a section that discusses the observations in more detail (i.e. expand what is currently in the Introduction and Figure 1 and tie

this more closely to what the modelling section aims to achieve). Appendix A spends quite a bit of effort describing how measurements were made. Showcasing these (unpublished?) measurements in more depth early in the manuscript would help build a sense of expectations with regards to what the model is expected to do or not do. That could also be used as a "roadmap" to explain how the paper is organized and why. It is also not totally clear why the authors focus on only a few data sections in their discussion. Figure 5a&b show that the model struggles to fit observation in the upper Pacific on P16. This seems certainly worthy of further examination and commenting.

Since the authors emphasize goodness of fit as the main theme throughout their paper, it is not clear what is gained from the simulations in sections 3.2 & 3.3. (why these experiments were chosen and not others?) It is clear from Table 5 what the results of these sensitivity experiments are. In that sense, little new insight is gained from reading section 4.2 and 4.5. Discussion section 4 is also generally focused on outlining model flaws and omissions. While this honesty from the authors is appreciated, this discussion section has less scientific value than it could have as it caters more to the model developers themselves than to the general public interested in the Mn cycle.

Specific comments

- Please refrain from listing all the "in preparation" papers in the Introduction. The general public doesn't have access to these manuscripts at this point and there is no guarantee "in prep" papers will be published. Since these "in prep" papers seem to be review papers, a few well-chosen published references would be much more appropriate.

- The first line of the Introduction argues that Mn update by phytoplankton is important, even "crucial for photosynthesis". Yet, the argument for not modelling the biological aspect of Mn cycling is that "not enough data is available to constrain the processes (L22-23, p22)" and because "there is no clear evidence for typical updake-remineralization processes" (L15, p26). It seems that testing these ideas is very much what the value

of a Mn model would be? Also, how can you say Mn is "crucial for photosynthesis" in the Introduction if there is no information available on biological uptake?

- P22, L10: "the k values in our model are optimised to get values...". How is this "optimisation" done? Is optimisation really what was done?

- For figures 5, 6, 7, 9 that compare simualtions to data, consider adding panels showing the relative errors achieved by the model simualation with respect to the data. This would be helpful to interpret the meaning of the bulk statistical fit diagnostics reported in Table 5 and understand misfit patterns in more detail. The color scheme chosen is also not that great for these figures as it is very hard to distinguish dark blues from dark greens, even if these colors would imply errors of order 200-300%.

- How does the Mn inventory evolve from one sensitivity study to the next? These sensitivity runs would make more sense if the global inventory of Mn was kept constant. Maybe discuss the inventories as part of Table 3.

- Appendix B, p29. From what I understand in this section, it seems that the interpolation of the model simulations to the sampling location is done is different ways. If that's in fact the case, why?

- Why is the dependency of Mn on O2 not considered? (p11, L9)

- Section 2.2.6. Can Mn scavenge on other particles besides autigenically formed Mn oxides in this model? Is the lack of particles in deep water what you are trying to approximate with the "aggregation threshold"?

- On p11, L21, you say "manganese oxide is buried when arriving at the ocean floor, which means that it is removed from the model domain". If that is the case, what is the source of the sedimentary Mn from section 2.2.3, eq (3)? Is the amount of Mn buried substantially less/more(?) than the sedimentary Mn source? how do these compare? Is the sediment source totally decoupled from the burial sink in this model? Is that decoupling justified?

---

## Referee Comment (RC3) · Anonymous Referee #3 · 3 Sep 2016

The manuscript by Hulten et al. presents the first global model of oceanic manganese cycling and compares the model output with two high-quality sections of dissolved manganese concentrations in the Atlantic and Southern Ocean. Since manganese is a biologically important element and can be limiting growth in some regions of the ocean, this is an important step forward and certainly of interest to the readers of Biogeosciences and should be published.

In constructing their model, the authors follow the principle that models should start as simple as possible, and include complex processes — even if it is known that they occur — only if the observational data directly shows evidence for these processes. This approach of starting simple and increasing complexity only after thorough analysis of model deficiencies is laudable. But it also means that this work is likely to be just a first step, and that more complex models will follow.

[Figure]

The main victim of this approach of starting simple is that the model disregards the biological cycling of manganese, although it is known that manganese availability is necessary for oxygenic photosynthesis, and can be (co-)limiting to phytoplankton growth at times (Brand et al. 1983, Bruland et al. 1991). The authors argue that in the data they use for validating the model, there is no strong signal of uptake and remineralisation of Mn.

I must admit that I am sceptical about this assumption:

- firstly, is the argument to neglect of biological Mn cycling maybe biased by using primarily Atlantic data? The secondary maximum of dissolved Mn in the North Pacific, shown in Bruland et al. 1991 seems hard to explain without remineralisation of Mn in biological material, and — in addition — a slower Mn(II) reduction in the oxygen minimum. This at least is what I have gotten out of the 1-d modelling study by Johnson et al. 1996.

- secondly, how consistent is the neglect of biological cycling with the magnitude of other vertical fluxes of Mn in the model? At least an estimate for the magnitude of the vertical export of Mn through biology could be obtained by multiplying Mn:P ratios in phytoplankton (Sunda and Huntsman, 1998, Twining 2013) with common estimates of global export production. The vertical sinking flux of particulate Mn (only the precipitated form, not the non-dissolved lithogenic part) from the model could be compared to that number.

There are other parameterisations in this model that are less ad-hoc than they appear at first sight: The authors use a constant rate for Mn(II) oxidation and a reduction rate for Mn(IV) that switches abruptly from a high value in the euphotic zone to a low value below. Although this is qualitatively justified by the existence of photoreduction, one wonders where the values come from, until one finds in the discussion that this simple parameterization was chosen to reproduce the particulate/dissolved Mn ratio at the

Vertex site. I think figure 11 and the corresponding explanation should be shifted to the model description. Instead, in the discussion, I think the authors could perhaps discuss what they think is the driving process for this reduction and whether it is likely that the Mn(IV)/Mn(II) ratio in the deep ocean is the same in different ocean basins.

My last criticism concerns the parameterisation of the sinking removal of oxidised Mn; contrary to the spirit of starting a model with simple assumptions, I find this parameterisation not simple at all. At the surface and at concentrations below 25 pM, Mn particles are assumed to sink at 1 m/d. Has this value been estimated from some assumption on the size of the particles, using Stokes' law, or it it just an arbitrary choice? Most of the intially formed particles probably are so small they don't sink at all on their own and need aggregation with other organic particles to sink, but most aggregates sink quite a bit faster. The model assumes that above the threshold od 25 pM, Mn particles aggregate and sink faster, but the increasing sinking rate above the threshold is made dependent on water depth, so that it does not affect the removal of Mn near the surface. Is there any justification for the depth dependency of the sinking rate? Several biogeochemical models use a vertically increasing sinking rate for its detritus compartment, as an implicit way to account for aggregation, see e.g. Kriest and Oschlies 2010, but the absolute sinking rates for aggregates in these models are much higher than assumed here. I found the choices here quite ad-hoc and wondered how strongly the model results are affected by them.

Summarizing, I think that the authors need to discuss the processes that lead to the vertical transport and ultimately removal of Mn in their model more, before the paper can be published. They have shown with their sensitivity runs that aggregagtion seems to be important for the removal of Mn from hydrothermal plumes, and that a lower threshold for this aggregation seems to be important for explaining the relatively uniform Mn background in the deep ocean. But if indeed aggregation is the main vecor of vertical transport, then one might suspect that near the surface, biological export production plays a large role in determining the vartical Mn flux, either directly from incorporated Mn, or indirectly from driving aggregation. That would lead to a much less uniform vertical transport velocity and it would be good to have an idea how much that could influence model results.

**Specific comments:**

page 3, line 3-4: My suspicion is that the good correlation between lithogenic particles and Mn may be caused by Mn inside lithogenic material, not so much by scavenged Mn. I would therefore be cautious to cite that as an evidence for that 'lithogenic particles are likely to play a significant role in the removal .. on Mn'.

Equation 3 for the Middelburg et al. empirical formula for denitrification is incomplete, as the units of neither the denitrification rate nor the water depth are given. Probably it is metre for the latter, but for the first?

**Minor comments:**

p.4, line 17: It it on purpose that the concentration of O2 is denoted by square brackets, but that of the OH- ion by curly brackets?

In my printout of the pdf file several of the equations appear mangled, with mathematical symbols replaced by small blocks, vertical lines replaced by some dots, the vertical bracket in eq. 20 missing etc. I think this is the consequence of using the a formula editor, which relies on software-specific fonts. The formulae should be checked out not only on a computer screen, but also in a printout.

Also, in my printout of Figure 2, all arrows are replaced with strange symbols, the ligated letters ff in 'sediment diffusion' are lost etc. I think this means that the pdf relies on the presence of some fonts that are present some type of computers, but not elsewhere. The ligated 'fl' in flux is also lost in the captions of the subfigures in figure 3.

It is very convenient that the caption to Figure 4 marks the cruise names in the same colours as the dots within the figure, but is this feature supported in the final journal

form?

Several of the citations are incomplete, e.g. Charette et al.

**References not present in the original manuscript:**

Kriest, I., Oschlies, A. (2011). Numerical effects on organic-matter sedimentation and remineralization in biogeochemical ocean models. Ocean Modelling, 39(3-4), 275–283. doi:10.1016/j.ocemod.2011.05.001
* * *

---

## Author Comment (AC1) · 4 Oct 2016

**General response**

All three reviews contain useful comments that resulted in discussion among the authors. Important themes of the discussion were the ommision of a biological cycle and O₂ and pH dependency, as well as moving the focus on the observational data.

The observations at the GA02 transect have not been published and already give some of the insights reached or confirmed by the model runs. For this reason we plan to restructure the paper, giving more space for a discussion on these observations.

As the reviewers correctly remark, and as we acknowledge in our discussion paper, Mn

does play an important role in biology. Therefore, we will also be adding a biological cycle but a dependency of Mn redox on pH and $O_2$ would not be possible to do within a reasonable amount of time. Here we depend on a carbon cycle and circulation model that may not be adequate for this purpose. This may already be true for the biological incorporation/remineralisation, but we do realise that a simulation with a constant Mn/P incorporation rate within our current model may be useful.

We have chosen a different dust deposition field (Mahowald et al. 1999) for our new simulations; one that has a bigger overall flux, especially in the Pacific Ocean, but is still within the observational uncertainty. This was necessary because the addition of a Mn biological cycle decreased dissolved Mn concentrations significantly in some parts of the ocean. Obviously the model-derived concentration of Mn in surface waters is a balancing act between supply mostly from (i) dust influx but also (ii) the Mn emanating from reducing sediments, and loss due to (iii) the biological Mn uptake and export, and (iv) the a-biological (i.e. adsorptive scavenging with or without oxidation) export flux. One of our sensitivity simulations will be with the original Hauglustaine et al. (2004) flux, which clarifies the change in dust field from the discussion paper to the final paper.

We are all very much aware that the reality in the oceans is far more complicated than what is actually being measured by seagoing oceanographers, and hence being simulated in models like ours. Most notably, all colleagues are aware that there exists for Mn, and other metals like Fe, a spectrum of colloids in a suite of size classes. If field data were available for at least one pool of colloidal Mn, then it would be worth considering to include such a separate pool in the model design. However, the datasets available thus far have been almost exclusively only for 0.2 micron or 0.4 micron seawaters filtered, i.e. operationally defined dissolved Mn, and one small dataset of particulate Mn, that is the Mn captured on a filter. Accordingly, the simulation model includes dissolved Mn and particulate Mn but, as yet, no colloidal Mn pool. More generally, the complexity of the model should not be (much) larger than what the observations can constrain. Of

course, more extensive modelling and observational studies should be done to arrive at a more complete knowledge of Mn, but the current study is only a first model.

---

## Author Comment (AC2) · 4 Oct 2016

**Response to Anonymous Referee #1**

We wish to thank the reviewer for the succinct analysis and criticism on our model.

The threshold Mnox that is used to account for the homogeneous background concentration of dissolved Mn of about 0.10 nM to 0.15 nM observed throughout most of the deep ocean, may result from over simplification of the model. There might be very different Kox and Kred values at upper (above ~300 m) and deeper (below ~300 m) part of the ocean. Mndiss may be mainly derived from remineralization of sinking organic matter in the upper ocean, and from an equilibration with colloidal or fine particles via absorption/colloid formation processes in the deeper ocean.

[Figure]

One would expect very different kred values at different depths.

The reduction and oxidation processes are indeed simplified a lot. We decided to devise a model that is as simple as possible, but reproduces reasonable dissolved Mn concentrations, and at the same time teaches us something useful. For this we have set $k_{ox}$ to a constant, but realised on the outset that $k_{red}$ must be much larger in the photic zone. Thus our most simple and still reasonable model was one where the oxidation rate is a constant and the reduction rate a two-valued step function (Eqs 7–9). Still, we do realise that bacterial activity, oxygen minimum zones and colloids influence the $k$ values. We wrote that the dependence of oxygen would be advisable for future development, but decided that this was not needed for this first study, nor necessary to arrive at our conclusions. The influence by colloids would be difficult to verify, because the observations that we used do not include colloidal and "truly dissolved" fractions: the "dissolved Mn" in both the observations and the model includes both fractions. This means that the $k$ values are really the effective $k$'s between the operationally defined dissolved and particulate pools.

Concerning the relatively homogeneous background concentration, this could not be reached by only setting constant values for $k_{ox}$ and $k_{red}$ in the deep ocean. It is rather a consequence of a Mn oxide concentration threshold on the increased settling velocity (Eq. 11). There are two ways to look at this part of the model. One is to see it as a trick to get the right deep-ocean dissolved Mn concentration. The other is the interpretation of the threshold being a necessary minimum concentration of Mn oxides before aggregation and efficient settling can occur. Both are true, but the second is the interesting one here because it gives an actual, potential explanation for the constant background concentration.

In addition, Mndiss may be mainly removed from the water column via oxidation to insoluble Mn(IV) with a rate that decreases with increasing depth due to lower dissolved oxygen concentration and lower pH at deeper depths, leading to very

different kox values at these different depths. Thus both k values and their ratios Kred/kox are not homogeneous throughout the water column. Such difference may cause lower modelled Mndiss than the observed values, thus requiring a threshold Mnox to account for higher Mndiss at deeper depths.

Yes, in areas with lower oxygen concentrations and lower pH, the $k_{ox}$ would be lower, resulting in a higher dissolved concentration which is what we want when the threshold were to be removed from the model. However, at the moment we think that nowhere in the deep ocean $k_{ox}$ is actually low enough to accomplish this in our model. This may very well be related to the high deep-ocean settling velocity we chose in our model. However, if we would decrease this velocity, the hydrothermal plumes would extend too far (Section 4.2). If we would then furthermore reduce the hydrothermal input, the hydrothermal signals would not be reproduced by the model. Furthermore, as Fig. 11 shows, we have chosen at least the $k_{ox}/k_{red}$ ratio quite well, meaning that while minor features are likely to be improved by using an inhomogeneous $k_{ox}$, it could never capture the much larger effect that we achieve by the threshold on the increased settling velocity.

---

## Author Comment (AC3) · 4 Oct 2016

**Response to Anonymous Referee #2**

We wish to thank the reviewer for the analysis and suggestions on our manuscript.

> While I am very supportive of the objectives of this work in general, the scope of
> the discussion, the wording and overall presentation of this manuscript gives the
> impression that the model is not sufficiently mature: at this stage of development,
> this paper reads more like a progress report than a fine-tuned, well-thought-out,
> scientific manuscript. The story in this manuscript announces a great start for
> Mn modelling, but comes across a bit short. Specifically, excluding the biological
> aspect of Mn cycling is a major omission (the authors themselves acknowledge

that this is a major aspect of Mn cycling).

While the model does not include all the interesting processes, we believe that the model is sufficiently mature to publish as a first study. Concerning the biological Mn cycling, we agree and now have included a biological module in our simulation.

> As a reader, it is not totally clear what has been learned about the Mn cycle by reading this manuscript. Interesting points are made in the last few lines of the abstract, but these are unfortunately not developed fully in the text. Most of the discussion focuses instead on technical limitations of the model and on a handful of parameterizations used to modify the model to make it fit data – with variable success. Since the bulk of the model presented here is essentially a scaled version of the existing companion Fe-model, the scope of the discussion presented here is somewhat limited.

We believe that our model can be used as a starting point for studies that teach us about the Mn cycle. In other words, this model is a first model that should be primarily considered as a proof of concept and basis for further study. Such further study is seen as doing more, and other type, of field measurements, as well as further development of the simulation model. These should go hand in hand, given the currently available type and amount of field data a more complex model would be overkill. However, we do put forward several insights in the cycling of Mn in the ocean (which should be considered as further supporting the proof of concept). This could still be considered somewhat limiting, which is why we decided to include more discussion in the paper by a more extensive discussion on the GA02 transect that has not been published in a paper before.

The four bullet points at the end of the abstract follow from, or are strongly suggested by, the manuscript. Our first point concerning the high concentration in the upper ocean was already to some extend established in the literature. Our second point on the

deep AMOC transport is illustrated by both the observations and the model; the model makes it more intuitive because of the high deep settling velocity. Point three on the background concentration states what the model shows, and then goes on that this supports the idea that a minimal concentration of Mn oxides are needed before significant removal occurs (see simulation *NoThreshold*). It is outside the scope of this modelling study to test this; rather laboratory and/or field experiments are needed for this (and maybe different type of modelling that includes a more mechanistic description of aggregation). Point four on the hydrothermal signal also follows from the paper (see simulation *LowHydro*).

One aspect of the writing that is not helping convincing the reader that this is a conclusive piece of work, is the over-reliance of subjective statements to describe the simulations and on imprecise/loose statements. This undermines the results. For example, the model fields are repeatedly qualified as "realistic", but this really doesn't mean anything, as there are just as many paragraphs in the paper outlining model flaws. The figures also reveal quite a few regions where the model does not fit the data. There is also no real set of criteria that are presented to define what "realistic" means. The authors should instead define the features their model fits or doesn't fit directly, without trying to oversell their product as "realistic", and explain in a bit more detail why the model fits or doesn't fit particular features. Other examples are: "The model reproduces observations accurately", "This is mostly captured by the model", "It is probably difficult to improve this feature of the model", "Reference simulation gives a reasonably realistic distribution", "LowHydro appears to give a much worse prediction", "This first-order approach works well mostly, but shortcomings can be identified", "the flux may be overall underestimated", "this may still contribute to the dissolved pool in the bottom water", "underestimation may be due to too strong vertical mixing", etc. . . Some of these propositions could be tested explicitly by the authors using additional sensitivity studies or appropriate diagnostics to produce objective conclusions instead of suggestions.

[Figure]

In the final version we will try to refrain from subjective statements, and make them more objective with statistics or other analyses, or remove them completely. The statement "LowHydro appears to give a much worse prediction" is subjective on its own, but it is supported by our statistical analysis.

> One way forward could also be to develop a section that discusses the observations in more detail (i.e. expand what is currently in the Introduction and Figure 1 and tie this more closely to what the modelling section aims to achieve). Appendix A spends quite a bit of effort describing how measurements were made. Showcasing these (unpublished?) measurements in more depth early in the manuscript would help build a sense of expectations with regards to what the model is expected to do or not do. That could also be used as a "roadmap" to explain how the paper is organized and why. It is also not totally clear why the authors focus on only a few data sections in their discussion. Figure 5a&b show that the model struggles to fit observation in the upper Pacific on P16. This seems certainly worthy of further examination and commenting.

It would be a good idea to discuss all the observations, and also using all of them for the statistics; but we decided to focus on the GA02 transect. You are right that we can discuss the observations in more detail. The GA02 transect on which we focus has not been published before (except as a data product on the GEOTRACES website). It now has been decided among the co-authors, that the field data and its primary interpretations, now will become part of this simulation modelling paper, instead of thus far chosen by a separate paper. We will restructure the paper such that the discussion of these observations is more extensive and prominent than it is now in the paper.

Other observations are used in a visual (more subjective) way, and detailed discussion and statistics on comparing those with the model (and GA02 observations) is not straightforward because they use other methods or are not intercalibrated. Future studies could include many more measurements for a statistical comparison; this would imply the prerequisite of evaluating every dataset for accuracy and consistency

with the others. In the future there will also be more GEOTRACES sampling cruises in the Pacific Ocean and other basins. Analyses from those samples will be following the same quality protocols, thus be more suitable for model–data comparison than many of the already existing observations.

> Since the authors emphasize goodness of fit as the main theme throughout their paper, it is not clear what is gained from the simulations in sections 3.2 & 3.3. (why these experiments were chosen and not others?) It is clear from Table 5 what the results of these sensitivity experiments are. In that sense, little new insight is gained from reading section 4.2 and 4.5. Discussion section 4 is also generally focused on outlining model flaws and omissions. While this honesty from the authors is appreciated, this discussion section has less scientific value than it could have as it caters more to the model developers themselves than to the general public interested in the Mn cycle.

Sections 3.2 shows that we need the combination of high hydrothermal input and high export to get such a good simulation as described in Section 3.1. Section 3.3 shows that it is not as simple as that: the high export needs to be limited to get the correct background concentration. We could have chosen other simulations as well to test further properties of the model, but we think that the chosen sensitivity simulations are the ones that provide the most important insight in the model. A general note on this: it is often useful to start with a realistic simulation, and do sensitivity simulations based on this (good) reference simulation.

**Specific comments**

> - Please refrain from listing all the "in preparation" papers in the Introduction. The general public doesn't have access to these manuscripts at this point and there is no guarantee "in prep" papers will be published. Since these "in prep" papers

seem to be review papers, a few well-chosen published references would be much more appropriate.

Most "in preparation" papers in the Introduction are to be published in a special issue of the recent Royal Society GEOTRACES conference that connects closely with the goals of this paper. That special issue is very likely to be published before our Mn modelling paper. We also think that for the Introduction these review papers are the most useful for the reader, because they give an overview to all state-of-the art knowledge, and contain useful references. We will remove the citation to the Middag et al. "in preparation" paper from the manuscript, and instead what was aimed to be in that separate paper will now also be added to the current paper.

- The first line of the Introduction argues that Mn update by phytoplankton is important, even "crucial for photosynthesis". Yet, the argument for not modelling the biological aspect of Mn cycling is that "not enough data is available to constrain the processes (L2223, p22)" and because "there is no clear evidence for typical updake-remineralization processes" (L15, p26). It seems that testing these ideas is very much what the value of a Mn model would be? Also, how can you say Mn is "crucial for photosynthesis" in the Introduction if there is no information available on biological uptake?

We know that Mn is needed in organisms, and it plays multiple roles in photosynthesis. We also have estimates of inner-cell Mn from Twining and Baines (2013). This means that Mn is incorporated into the cell during growth, but we do not know how this process takes place and what happens with the Mn after it is incorporated. Possibly it is very similar to what happens to phosphate, so we decided to simulate a biological Mn cycle by following P, using a Mn:P ratio from Twining and Baines (2013), consistent with Middag et al. (2013). In this way, we have tested the effect of the trivial case of an uptake-remineralisation proportional to phosphate. First results of this simulation show

that it may be relevant for the equatorial region and at mid-latitudes, especially in the Pacific Ocean. We will include and discuss further this result in the paper.

> - P22, L10: "the k values in our model are optimised to get values. . .". How is this "optimisation" done? Is optimisation really what was done?

No, we did not use the right terminology here. We set $k_{red,light}$ simply to the mean value found by Sunda and Huntsman (1994), then used the dissolved and particulate profiles in Fig. 4 of Bruland, Orians and Cowen (1994) to derive $k_{ox}$ and $k_{red,dark}$.

> - For figures 5, 6, 7, 9 that compare simualtions to data, consider adding panels showing the relative errors achieved by the model simualation with respect to the data. This would be helpful to interpret the meaning of the bulk statistical fit diagnostics reported in Table 5 and understand misfit patterns in more detail. The color scheme chosen is also not that great for these figures as it is very hard to distinguish dark blues from dark greens, even if these colors would imply errors of order 200–300 %.

We already spent quite some time in producing good comparison plots, and do not think we can improve on these notably. The statistics support the comparison in a quantitative manner.

> - How does the Mn inventory evolve from one sensitivity study to the next? These sensitivity runs would make more sense if the global inventory of Mn was kept constant. Maybe discuss the inventories as part of Table 3.

We will add the inventories to Table 3. If one does a sensitivity simulation, it is not straightforward what is the best way to keep the inventory invariant compared to the reference simulation, or if this would be very useful. In the case of *NoThreshold* we want to see the change of inventory.

- Appendix B, p29. From what I understand in this section, it seems that the inter-polation of the model simulations to the sampling location is done in different ways. If that's in fact the case, why?

Yes, for the horizontal coordinates we used the sampling locations, while for the vertical we did either nothing (for the plots) or used the model grid (for statistical comparison). For the plotting the reason for interpolating the model to the the station coordinates was that there needed to be some kind of interpolation onto a grid that includes the sampling locations; the simplest choice was the minimal one where the new grid is defined by the station coordinates. The reason for the statistics is that while we preferred a comparison of the model at the sampling locations, this would be less trivial in the vertical, since the sample depths vary per station. Since the vertical spacing of the model is similar to that of the samples, we decided to interpolate the observations onto the vertical model grid.

- Why is the dependency of Mn on O2 not considered? (p11, L9)

It would be better to include such a dependency, as the effect is clear in, for instance, the Pacific Ocean. Johnson et al. (1996) had tested this on a local scale. We decided not to repeat this exercise, even though it would be sensible to include it at some point in global Mn models. We found it not to be essential for our purposes at this point. The downside of adding such a dependency is that it would make the model more complicated. It would imply relying on the circulation and oxygen distribution of a biogeochemical carbon model that has uncertainties of its own. So far it has shown quite difficult to simulation OMZs and its effects in an accurate way. However, as a first test we have introduced biological cycling of Mn.

- Section 2.2.6. Can Mn scavenge on other particles besides autigenically formed Mn oxides in this model? Is the lack of particles in deep water what you are trying to approximate with the "aggregation threshold"?
This may be the case. We believe that a more mechanistic model than the one presented in the current manuscript would be much more difficult, as it should include several size classes of particles, different types of manganese and other complications. This should be done, but it is unfeasible for the current study.

- On p11, L21, you say "manganese oxide is buried when arriving at the ocean floor, which means that it is removed from the model domain". If that is the case, what is the source of the sedimentary Mn from section 2.2.3, eq (3)? Is the amount of Mn buried substantially less/more(?) than the sedimentary Mn source? how do these compare? Is the sediment source totally decoupled from the burial sink in this model? Is that decoupling justified?

It is indeed decoupled. Coupling with a sediment model may be useful for future studies. We will compare burial with forced Mn input and say something about it in the updated manuscript. If useful, we might add a burial figure as well. However, the expectation is that the correlation with the forced flux is very bad: sediment source is large at shallow sediments, while the hydrothermal source would generate most of the Mn burial. This is probably realistic, and suggests that coupling water column and sediment may not be useful, or very difficult, for a global study.

---

## Author Comment (AC4) · 4 Oct 2016

**Response to Anonymous Referee #3**

We wish to thank the reviewer for the critical but also positive comments on our manuscript. The reviewer said that this

> approach of starting [with a] simple [model] and increasing complexity only after thorough analysis of model deficiencies is laudable. But it also means that this work is likely to be just a first step, and that more complex models will follow.

This is absolutely the idea. He or she continues:

The main victim of this approach of starting simple is that the model disregards the biological cycling of manganese, although it is known that manganese availability is necessary for oxygenic photosynthesis, and can be (co-)limiting to phytoplankton growth at times (Brand et al. 1983, Bruland et al. 1991). The authors argue that in the data they use for validating the model, there is no strong signal of uptake and remineralisation of Mn. I must admit that I am sceptical about this assumption: - firstly, is the argument to neglect of biological Mn cycling maybe biased by using primarly Atlantic data? The secondary maximum of dissolved Mn in the North Pacific, shown in Bruland et al. 1991 seems hard to explain without remineralisation of Mn in biological material, and – in addition – a slower Mn(II) reduction in the oxygen minimum. This at least is what I have gotten out of the 1-d modelling study by Johnson et al. 1996.

We now have decided to include a module for biological cycling in the model, and results of first simulations including such a biological module are encouraging. Yes, right now we are missing the secondary maximum. This is especially clear in the Pacific Ocean where there is not only a secondary maximum around 800 m, but also a higher concentration in the upper 250 m; neither are captured by our model. There should really be a study more dedicated to these issues, including indeed the uptake-remineralisation processes as well as the effect of oxygen minima into consideration. Future GEOTRACES sampling expeditions will also help in that they would be following the same quality standards as, amongst other GEOTRACES transects, the GA02. Nonetheless, there are a large number of processes and uncertainties when doing a first model. Thus, we believe that for a first global model study it is not too bad that we chose a more simple approach.

- secondly, how consistent is the neglect of biological cycling with the magnitude of other vertical fluxes of Mn in the model? At least an estimate for the magnitude of the vertical export of Mn through biology could be obtained by multiplying Mn:P ratios in phytoplankton (Sunda and Huntsman, 1998, Twining 2013) with common estimates of global export production. The vertical sinking flux of particulate Mn

(only the precipitated form, not the non-dissolved lithogenic part) from the model could be compared to that number.

Indeed the biological module that we now have included is based on a typical or world-average Mn/P element ratio of plankton.

> There are other parameterisations in this model that are less ad-hoc than they appear at first sight: The authors use a constant rate for Mn(II) oxidation and a reduction rate for Mn(IV) that switches abruptly from a high value in the euphotic zone to a low value below. Although this is qualitatively justified by the existence of photoreduction, one wonders where the values come from, until one finds in the discussion that this simple parameterization was chosen to reproduce the particulate/dissolved Mn ratio at the Vertex site. I think figure 11 and the corresponding explanation should be shifted to the model description. Instead, in the discussion, I think the authors could perhaps discuss what they think is the driving process for this reduction and whether it is likely that the Mn(IV)/Mn(II) ratio in the deep ocean is the same in different ocean basins.

Yes, the abrupt change from high to low reduction from the photic to the aphotic zone is ad hoc. In retrospect, we could have used a continuous function of depth, which would probably even gotten a higher correlation index, but the choice of the function is not constrained by the literature. (Linear in light intensity may be a good approximation.)

> My last criticism concerns the parameterisation of the sinking removal of oxidised Mn; contrary to the spirit of starting a model with simple assumptions, I find this parameterisation not simple at all. At the surface and at concentrations below 25 pM, Mn particles are assumed to sink at 1 m/d. Has this value been estimated from some assumption on the size of the particles, using Stokes' law, or it it just an arbitrary choice? Most of the intially formed particles probably are so small they don't sink at all on their own and need aggregation with other organic particles to sink, but most aggregates sink quite a bit faster. The model assumes that above

the threshold od 25 pM, Mn particles aggregate and sink faster, but the increasing sinking rate above the threshold is made dependent on water depth, so that it does not affect the removal of Mn near the surface. Is there any justification for the depth dependency of the sinking rate? Several biogeochemical models use a vertically increasing sinking rate for its detritus compartment, as an implicit way to account for aggregation, see e.g. Kriest and Oschlies 2010, but the absolute sinking rates for aggregates in these models are much higher than assumed here. I found the choices here quite ad-hoc and wondered how strongly the model results are affected by them.

The removal of oxidised Mn is somewhat arbitrary indeed. Most Mn probably sinks slowly with small lithogenic particles (p. 25, lines 11–15). This could not explain the strong removal rate near hydrothermal vents. One approach would be to include more species of particulate (and maybe dissolved) manganese. We chose to stick with only (observationally defined) two pools, namely dissolved and particulate manganese. This meant that we needed to set a high velocity near hydrothermal vents, but the removal needed to be small enough to maintain high surface concentrations and the (lower) background concentration. Our parameterisation works for this.

Summarizing, I think that the authors need to discuss the processes that lead to the vertical transport and ultimately removal of Mn in their model more, before the paper can be published. They have shown with their sensitivity runs that aggregagtion seems to be important for the removal of Mn from hydrothermal plumes, and that a lower threshold for this aggregation seems to be important for explaining the relatively uniform Mn background in the deep ocean. But if indeed aggregation is the main vecor of vertical transport, then one might suspect that near the surface, biological export production plays a large role in determining the vartical Mn flux, either directly from incorporated Mn, or indirectly from driving aggregation. That would lead to a much less uniform vertical transport velocity and it would be good to have an idea how much that could influence model results.

In the photic zone the sinking speed equals 1.0 m/d in the model, while higher speeds can be found around and below 100 m depth. That said, there may already be some aggregation directly below the photic zone in the model, because accumulated (dissolved) Mn in that region quickly oxidises, hence exceeding the 25 pM threshold and having a higher speed than in the photic zone. This may be expected in the real ocean as well, but we cannot be sure whether our model does it in a correct way. The model suggests that aggregation would only be important in the deeper ocean.

We will address these issues in the next version of the manuscript.

**Specific and minor comments**

> page 3, line 3-4: My suspicion is that the good correlation between lithogenic particles and Mn may be caused by Mn inside lithogenic material, not so much by scavenged Mn. I would therefore be cautious to cite that as an evidence for that 'lithogenic particles are likely to play a significant role in the removal .. on Mn'.

Lithogenic particles contains Mn, but in Roy-Barman et al. (2005) the correlation is between the authigenic Mn concentration and lithogenic tracer concentration such as $^{232}$Th in sediment. In this study, the authigenic Mn content exceeds largely the lithogenic Mn content, so that the correlation is indeed between authigenic Mn and lithogenic particles.

We will rephrase this accordingly in the new version of the paper.

> Equation 3 for the Middelburg et al. empirical formula for denitrification is incomplete, as the units of neither the denitrification rate nor the water depth are given. Probably it is metre for the latter, but for the first?

We will add the units. It is actually the logarithm of the denitrification flux; we'll make that clear in the text.

> p.4, line 17: It it on purpose that the concentration of O2 is denoted by square brackets, but that of the OH- ion by curly brackets?

Yes, this is how Johnson et al. (1996) have it. The [ ] is a concentration, {} activity. Since it is used only once, we will just use the words there, and not the symbols.

> In my printout of the pdf file several of the equations appear mangled, with mathematical symbols replaced by small blocks, vertical lines replaced by some dots, the vertical bracket in eq. 20 missing etc. I think this is the consequence of using the a formula editor, which relies on software-specific fonts. The formulae should be checked out not only on a computer screen, but also in a printout.

Next time I will make a print-out of the final version/proofs and check the typesetting on paper before confirming it. At first, I also thought it would be a font issue, but all fonts in the document are either standard or embedded (that's what Evince tells me). It may be a pdf library or printer driver problem. We didn't use a special formula editor, just LATEX.

> Also, in my printout of Figure 2, all arrows are replaced with strange symbols, the ligated letters ff in 'sediment diffusion' are lost etc. I think this means that the pdf relies on the presence of some fonts that are present some type of computers, but not elsewhere. The ligated 'fl' in flux is also lost in the captions of the subfigures in figure 3.

If we cannot find (nor solve) the problem ourselves, we will ask the editors to have a look at this problem at the time of submission. Thank you for raising this issue.

Just to get the issue clear: do all these problems only occur when printing the document, or are some of them also visible when viewing the pdf file? Any other information (software used) may be of use as well.

> It is very convenient that the caption to Figure 4 marks the cruise names in the same colours as the dots within the figure, but is this feature supported in the final journal form?

Good question. They state in the instructions that it is not allowed in tables, nothing about figures/captions. We'll just see what the editor says.

> Several of the citations are incomplete, e.g. Charette et al.

We will check and update our citations (like this one) where needed.

---

## Editor Decision (ED1)

*Dear Marco,*
*many thanks for your revised manuscript. It has been reviewed by two scientists again. It has improved a lot. But I would like you to carry out a few further amendments (see below). I look forward to your revised version of this pioneering article and appreciate your endurance.*
*Best wishes, Christoph*

**REVIEWER #1**

General comments
 The manuscript is overall much better than the first version and, in my opinion, only requires a few minor adjustments before publications.

 The introduction reads substantially better than version 1 and, this time, really sets the stage to an interesting paper. One major improvement over the last version is the way the various experiments are used and presented. This is now much more logical and helpful. There are still a few sections where additional explanations would be desirable (see specific comments), but overall, the structure is good.

 I still think that reliance on only the Atlantic data for assessing statistical model fit is a poor choice and this leads to some convoluted explanations in the main text, but this is no a major issue at this point. I respect this is the author's choice, even I believe it is a bad choice. I would simply suggest that the authors make it as clear as possible throughout the manuscript that all fits pertain to one single section in the Atlantic only.

 One remaining frustrating detail is that the authors seem to have replaced the word "realistic" with the word "accurate". Unfortunately, replacing a word doesn't fix the problem, which is to overly rely on highly subjective statements of quality. Please reduce this further by being more specific or more quantitative in your statements.

 Specific comments
 P5, l13, remove "on" from "on shipboard"

 P11, footnote (2). Just add that to the main text directly.

 P14, l24: "recently measurements" change to "recent"

 P15, l16: consider rephrasing this ": very soon the [Mndiss] reaches the typical background concentration", maybe " as Mndiss approaches near-constant deep background concentrations quickly such that the NADW plume is no longer discernable."

Section 3.3, p17. Expand this section – this is interesting. Please provide an interpretation as to why Atlantic value don't change but Pacific results do.

 P18, l15. Table 6 shows a reliability intex of 1.88 for NoThresh, not 2.77.

 P17, l20: What is your definition of "reasonably accurate"?

 P18, l27: overstatement "only the Reference simulation is accurate". What is your definition of accurate?

P19, l4-5: I don't understand the logic behind that argument about "usefulness": "We think that coupling the model to the sediment would … maybe not that useful, because the sediment source is large at shallow sediments, while most of the Mn burial occurs near the hydrothermal vents"

P19, l10-11: Does the model handle "small" shelf regions well? Can you make any quantitative argument about how well the model resolution handles shelves?

P19 ,l12: Why only report on Slomp's maximum values?

P19, l16: "out of proportion in some regions of the ocean". Where is that specifically? Is that only in the Arctic, as alluded to in the next sentences?
P19, l20-21: add a comma in "…Pacific Ocean, where…"

P19, l22: "In the East Pacific Ocean the California Current induces Ekman transport and hence equatorial upwelling". Very puzzling bit of physical oceanography? Some references in support of that statement would be very welcome. I believe wind induces Ekman transport, not the California current. I'm also not aware of physical theories of equatorial upwelling that argue the California current induces it.

P19, l23: "upwelling from OMZ sediments". Maybe "upwelling of water that has been in contact with OMZ sediments"

P19, l24: "This is partly captured by our model". Which part?

P19, l25: "In the South Pacific Ocean this effect is more clear in the data of Resing et al. (2015) (Fig. 8a,b, East Pacific around 20S)."… and in the model?

P19, l27-29: Better would be to provide a back of the envelop estimate of how much bias may come from not representing fluxes from OMZ sediments.

P20, l21: "and especially at low latitudes". Please substantiate this with a few sentences. The previous discussion was all about the Southern Ocean, not on low latitudes"

P20, l23: What is "the most settling Mn"? Do you mean the particulate Mn fraction that contributes most to the sinking Mn flux is from biological particles?

P20, l28: include, not includes

P22, l29: "because Mn redox does not depend on O2". Rewrite sentence. What is "Mn redox"?

P22, l31-43: "…For this reason we have not included a dependency on [O2] to the model…." Consider rewriting these few sentences in a less convoluted way.

P24, l6: remove "e.g." – say what you mean in words instead.

P25, l3: "for an accurate simulation of [Mndiss]". What is your definition of accurate? Replacing "realistic" with "accurate" doesn't remove the problem of relying on subjective statements.

Appendix A, p27,l3: do you refer here as the "Pearson correlation coefficient"? please specify.

Appendix A – Table 6. Why are there only errors for the Reference case and not all cases? For comparison purposes, errors should be calculated on all cases.

Figure 7, caption. What do you mean by the word "by in "the by red" or "the by blue" lines? Probably remove this. Also, would be good to make these lines thicker on the figure. They are very thin, even when the figure is full screen.

Figure 8: x-axis labels and sub-plot titles overlap. Fix spacing.

Figure 5 and 9. Choose a consistent name between GIPY5 or GIPY5_e.

Figure 13, explicitly state in the caption if relative difference is (ref-low hydro)/low hydro or (ref-low hydro)/ref?

Figure 16, make the colored lines thicker

**REVIEWER #2**

The manuscript by Hulten et al. has changed considerably in the revised version, with the two most important changes being that

- a biological cycling of Mn has now been implemented into the model, as a response to remarks by the reviewers. The cycling is parameterized as following the uptake and release of phosphorous from a global biogeochemical model, assuming a constant Mn:P ratio from Twining and Baines (2013). The modeled Mn distribution, as far as I can see, has no effect of phytoplankton growth in the model, i.e. Mn limitation is not included in the model. This is a reasonable first step, but should me mentioned in the model description.
- mostly in response to the second reviewer, the paper now contains a much more detailed description of the manganese observations along the dutch Geotraces section GA02, including a discussion of the methods for these observations.

The inclusion of a biological cycling of Mn in the model is reasonable, and I think it strengthens the paper a lot. I have, however, a bit mixed feeling about the new focus of the manuscript on the GA02 section: Reviewer 2 suggested "showcasing these (unpublished?) measurements in more depth early in the manuscript would help build a sense of expectations with regard to what the model is expected to do or not to do. That could also be used as a 'roadmap' to explain how the paper is organized and why." To me this aim has not yet been reached fully, the observational results (3.1) and the modelling results, especially section 3.2 still stand side by side in a too unconnected manner.

One example for this is the elevated value of Mn in the DSOW overflow, which is visible quite clearly in Fig 7 and discussed over a few sentences in section 3.1. In the modelling part, this feature is never mentioned again, and indeed the color scale in Fig. 9 is chosen in

such a way that it is not even visible in the observations anymore. My expectation is that the model does not reproduce this feature, and that is not even bad; it probably just highlights that the model is missing sediment resuspension, a locally inportant but probably globally unimportant process. In my opinion, often the most important information in model-data comparisons is where the two do not agree because here one learns about processes.

This is just one example, but my general impression is that the present manuscript does not integrate the observational and modelling parts enough and thus misses what the second reviewer had in mind when he suggested to focus more on observations. The model-data comparison could be made much more precise, and I would argue that the authors should try to do that in a second revision.

One aspect to improve is the "over-reliance on subjective statements to describe the simulations and on imprecise/loose statements" that was already mentioned by the second reviewer. I think the present manuscript still contains many too colloquial statements, but now also in the description of the observations. A few examples are that "very soon the Mndiss reaches the typical background concentration" when talking about NADW or "slightly elevated concentrations in the subsurface are also observed in ... AAIW .. but once again concentrations reach the typical background concentration". Both statements in section 3.1 would be much more precise with some indication where something happens. Likewise, "for a small part" later on the same page is not a helpful description. When comparing the NoBio with the reference run, it is stated that "NoBio generaly compares better with the observations in the Pacific Ocean especially with the US EPZT transect" Again, it would be helpful if the authors could be a bit more specific: Where is the improvement, and in what aspect? At the surface or at 500m depth?

Minor comments

Section 2.2.7: the value of 0.4 'derived by Middag et al. (2013)' should probably be $0.4 \cdot 10^{-3}$.

Section 3.1: "Whereas: on line 13 should probably be "As"

Page 16, line 17: maybe include "hydrothermal" before "forcing field"

Page 20, line 23ff: To me this is one of the most interesting results of the new model runs including biology; but why just state this here without giving any numbers? In my first review I already suggested that it would be informative to have an idea on the relative magnitude of the sinking fluxes of authigenic and ofbiologically incorporated Mn; The authors would make that point much stronger when calculating e.g. the globally integrated fluxes of Mn from Mnox and from the biological compartment, maybe at 100m depth and at some depth deeper in the water column. Just a suggestion..

The authors might consider using the same software for plotting observational and modelling results; Fig 7 is made using ODV, Fig 9 using ferret, I believe.

---

## Author Response (AR2)

**Response to Referees, 2nd round**

Marco van Hulten et al.

24th January 2017

Dear editor and reviewers,

Thank you for your input. We have considered all your suggestions and adjusted our manuscript accordingly.

The manuscript comprises much originality in presenting the new very large Mn dataset of the thus far longest ocean section GA02 in combination with a novel developed first-ever world ocean Mn simulation model. This work was done by the lead author in the Ph.D. period and subsequent postdoc time in Paris, next to other projects and tasks. In those years the GA02 dataset was available, and by re-organising this data into the grid of the simulation model, a statistical comparison exercise was feasible.

Only recently the USA Atlantic GA03 dataset and even more recently the USA Pacific GP16 dataset have become available. This did allow to show qualitatively in the various graphics an indication as to how the simulation compares with these very new datasets. For a more rigorous quantitative statistical comparison there simply is right now no labour time and computation finances available to do all this extra work of converting the dataset into the model grid, and doing all the statistical calculations. Also and perhaps even more important, the simulation of OMZ regions is not a simple and straightforward task, and also would require the new findings of the Pacific GA16 section to become available in new publications, in order to be able to cleverly draft the right type of refined resolution model boxes, and the right type of model simulation equations. For example the recent articles of Hawco et al. (2017) and Ohnemus et al. (2016) do provide a wealth of new insights, that first need to be studied and contemplated upon, before beginning to make modifications to the simulation.

As a matter of fact it may well be that the only feasible approach is to make separate models only for each one of the OMZ regions, and run these off-line from the overall world ocean model. Also one should not underestimate that for a world ocean model, as this is, one cannot simply solve one problem in one ocean, without the major risk of making at the same time the simulation worse in other oceans. Also fine colleague Resing has kindly made his dataset available, but intends to first write his own paper on Mn processes in the OMZ region off Peru, before this dataset can be utilised in a more rigorous statistical intercomparison with the simulation model.

All this would require much extra work and finances for computing. The lead author now having completed the period for work on this and other projects, and starting a new job quite soon, simply cannot do this right now.

We relabelled the reviewers arbitrarily to A and B, because the 1 and 2 were interchanged in the merged report (where rather the report number was used as it were the reviewer number and Reviewer 2 happened to respond earlier this time than Reviewer 1).

**Response to REVIEWER A (report 1)**

**General comments**

> The manuscript is overall much better than the first version and, in my opinion, only requires a few minor adjustments before publications.

> The introduction reads substantially better than version 1 and, this time, really sets the stage to an interesting paper. One major improvement over the last version is the way the various experiments are used and presented. This is now much more logical and helpful. There are still a few sections where additional explanations would be desirable (see specific comments), but overall, the structure is good.

We have done our best in amending these issues.

> I still think that reliance on only the Atlantic data for assessing statistical model fit is a poor choice and this leads to some convoluted explanations in the main text, but this is no a major issue at this point. I respect this is the author's choice, even I believe it is a bad choice. I would simply suggest that the authors make it as clear as possible throughout the manuscript that all fits pertain to one single section in the Atlantic only.

This should now be clear throughout the manuscript. Otherwise see the above introductory response on reasons why it is right now not possible to do more statistics on data-model veracity than the GA02 intercomparison thus far.

> One remaining frustrating detail is that the authors seem to have replaced the word "realistic" with the word "accurate". Unfortunately, replacing a word doesn't fix the problem, which is to overly rely on highly subjective statements of quality. Please reduce this further by being more specific or more quantitative in your statements.

We have now realised that it is confusing to use the word accurate for both (i) the actual dataset of dissolved Mn as measured by author Rob Middag, and (ii) for the agreement of model output with this dataset. Upon ample consideration we decided to maintain the word accurate for the actual measurements, and we use the word *veracity* for the agreement of the model output with this dataset. In the first pages these concepts have been defined, *accuracy* is defined at bottom of page 4 lines 31–33, veracity is defined at the top of page 5 line 6. Obviously, these definitions also apply to the words *accurate* and *veracious*, respectively.

The precision and accuracy of the measurements of dissolved Mn of the new published GA02 section dataset are presented as exact as can be in the main text of the methods and the related Table 1 and Figures 1 and 2.

Thus far a statistical data-model intercomparison has been done only for the GA02 dataset, see Table 6 and Appendix A. The Reliability Index RI (Leggett and Williams, 1981) provides a relative comparison of the veracity between the various model runs. Here the *Reference* run with lowest Reliability Index $1.76 \pm 0.08$ is deemed to have the best veracity. However as mentioned deviations of two other runs *NoBio* and *OxidThreshold* from the value of the Reliability Index of the *Reference* run are insignificant. Only the two remaining runs *LowHydro* and *NoThreshold* have significantly worse Reliability Index, as indicated by bold print of these values. For the run *LowHydro* this is further confirmed by the significantly worse Pearson Correlation and RMS deviation, their values 0.64 and 0.60, respectively, also given in bold print.

**Specific comments**

P5, l13, remove "on" from "on shipboard"

Done.

P11, footnote (2). Just add that to the main text directly.

Done.

P14, l24: "recently measurements" change to "recent"

Done.

P15, l16: consider rephrasing this ": very soon the [Mndiss] reaches the typical background concentration", maybe " as Mndiss approaches near-constant deep background concentrations quickly such that the NADW plume is no longer discernable."

Done.

Section 3.3, p17. Expand this section – this is interesting. Please provide an interpretation as to why Atlantic value don't change but Pacific results do.

We have added a Section 4.2.3 which discusses this more.

P18, l15. Table 6 shows a reliability index of 1.88 for NoThresh, not 2.77.

Indeed, fixed.

P17, l20: What is your definition of "reasonably accurate"?

We have included a definition on page 5 of the now chosen word "veracity". As it is not an exact notion, and in this context there is not such a thing as absolute veracity, we qualified this with an adverb. We tried to make it as clear as possible. The sentence has been improved as follows:

We have shown that the *Reference* simulation gives a reasonably veracious representation of the effects of hydrothermal vents and the background concentration in the deep ocean.

P18, l27: overstatement "only the Reference simulation is accurate". What is your definition of accurate?

We have now chosen to strictly use the word veracity for the model output, and this veracity is defined now on page 5. For the data–model intercomparison of only section GA02, the veracity is quantified by the three statistical entities Pearson correlation, RMS deviation and Reliability Index given in Table 6. For all the other data and including the GA02 section, there is qualitative data-model comparison in the various graphs of Figures 8, 9, 10, 12, 14, 15, as well as 16.

P19, l4-5: I don't understand the logic behind that argument about "usefulness": "We think that coupling the model to the sediment would . . . maybe not that useful, because the sediment source is large at shallow sediments, while most of the Mn burial occurs near the hydrothermal vents"

With "usefulness" we referred to the ideas that the model could practically be spun up and also that it would yield values close enough to the true value (or observations) that it may be useful for certain studies. We decided not to discuss coupling with the sediment in detail, and rewritten this (see Section 4.1).

> P19, l10-11: Does the model handle "small" shelf regions well? Can you make any quantitative argument about how well the model resolution handles shelves?

The sediment source was derived from Aumont et al. (2015). They note that "as a consequence of the relatively coarse resolution of ORCA2, the model bathymetry is not able to correctly represent the critical spatial scales of the ocean bathymetry. An example is the continental shelves, which typically have a width scale of 10–30 km". An algorithm was developed to account for this, but it clearly still has shortcomings. The resulting parameterisation overestimates large shelf regions (and underestimates certain small shelf regions where oxygen is low). We removed the word "large":

> [...] the low model resolution does not handle  shelf regions well.

> P19 ,l12: Why only report on Slomp's maximum values?

Looking again at the literature, we remembered that the choice was more subtle than just taking this maximum value. We changed the text, and more clearly refer to literature values now.

> P19, l16: "out of proportion in some regions of the ocean". Where is that specifically? Is that only in the Arctic, as alluded to in the next sentences?

It should be more clear now by mentioning the regions and adding references describing the same issue (page 20, lines 9–13 in marked up version).

> P19, l20-21: add a comma in "...Pacific Ocean, where..."

Done (plus semicolon earlier in sentence for clarity).

> P19, l22: "In the East Pacific Ocean the California Current induces Ekman transport and hence equatorial upwelling". Very puzzling bit of physical oceanography? Some references in support of that statement would be very welcome. I believe wind induces Ekman transport, not the California current. I'm also not aware of physical theories of equatorial upwelling that argue the California current induces it.

You are right. We repaired this.

> P19, l23: "upwelling from OMZ sediments". Maybe "upwelling of water that has been in contact with OMZ sediments"

Yes, and we have further reformulated this part.

> P19, l24: "This is partly captured by our model". Which part?

At the specified location (20° N, East Pacific boundary) there is an elevated value for dissolved Mn. But we rephrased this paragraph for clarity.

P19, l25: "In the South Pacific Ocean this effect is more clear in the data of Resing et al. (2015) (Fig. 8a,b, East Pacific around 20S)."... and in the model?

Much more clear in the data than in the model, actually. We rephrased this.

P19, l27-29: Better would be to provide a back of the envelop estimate of how much bias may come from not representing fluxes from OMZ sediments.

No measured Mn fluxes can easily be applied to a coarse model. Even if we have those fluxes, we cannot do a simple, reasonable calculation how much impact it would have on Mn concentrations, e.g. compared with other sources. The sediment source was derived from Aumont et al. (2015).

P20, l21: "and especially at low latitudes". Please substantiate this with a few sentences. The previous discussion was all about the Southern Ocean, not on low latitudes"

Done.

P20, l23: What is "the most settling Mn"? Do you mean the particulate Mn fraction that contributes most to the sinking Mn flux is from biological particles?

Reformulated.

P20, l28: include, not includes

Done.

P22, l29: "because Mn redox does not depend on O2". Rewrite sentence. What is "Mn redox"?

Done. "Redox" is defined at first use at page 2 in relation to the given Reaction (R1) equation. Redox is a very common word in research articles on Mn and other 'redox' chemical elements in the oceans, such as Fe, Co, nitrate/nitrite, sulphate/sulphide.

P22, l31-43: "... For this reason we have not included a dependency on [O2] to the model...." Consider rewriting these few sentences in a less convoluted way.

Done.

P24, l6: remove "e.g." – say what you mean in words instead.

Done.

P25, l3: "for an accurate simulation of [Mndiss]". What is your definition of accurate? Replacing "realistic" with "accurate" doesn't remove the problem of relying on subjective statements.

We have defined "accurate" before.

Appendix A, p27,l3: do you refer here as the "Pearson correlation coefficient"? please specify.

Done. It is indeed the Pearson correlation we use here, this is now stated as follows:

In addition to classical statistical indices (Pearson correlation index $r$, root mean square error RMS), another [...]

Appendix A – Table 6. Why are there only errors for the Reference case and not all cases? For comparison purposes, errors should be calculated on all cases.

This is not really needed but has now been added upon the request of the referee. As a matter of fact, the comparison is only with *Reference*. In Appendix A we explain that the error of only the *Reference* run is needed to decide on the sensitivity simulation's significance. We realised that we had not used the very last timestep (year 480 instead of 600) of the simulations for the statistics. Therefore we updated the statistics. The RI increased a bit for all simulations, so we decided to have another look at how close the simulations are to a steady state. Whereas at year 480 there was a decadal relative change of Mn content of 83 ppm in the surface (upper 100 m) and of 28 ppm below 100 m, this was only 24 ppm and 7 ppm, respectively, in the last decade (year 600 minus year 590). We shortly discuss this now at the beginning of Section 3.2. There are no consequences for the significances between the different simulations or any of the discussion.

Figure 7, caption. What do you mean by the word "by in "the by red" or "the by blue" lines? Probably remove this. Also, would be good to make these lines thicker on the figure. They are very thin, even when the figure is full screen.

We made the letters thicker, and remove/rephrased the colour referencing.

Figure 8: x-axis labels and sub-plot titles overlap. Fix spacing.

Done.

Figure 5 and 9. Choose a consistent name between GIPY5 or GIPY5_e.

Done. We now only use the name "GIPY5", and specify simply that it pertains the Zero Meridian part alone.

Figure 13, explicitly state in the caption if relative difference is (ref-low hydro)/low hydro or (ref-low hydro)/ref?

Done.

Figure 16, make the colored lines thicker

Done.

**Response to REVIEWER B (report 2)**

The manuscript by Hulten et al. has changed considerably in the revised version, with the two most important changes being that

- a biological cycling of Mn has now been implemented into the model, as a response to remarks by the reviewers. The cycling is parameterized as following the uptake and release of phosphorous from a global biogeochemical model, assuming a constant Mn:P ratio from Twining and Baines (2013). The modeled Mn distribution, as far as I can see, has no effect of phytoplankton growth in the model, i.e. Mn limitation is not included in the model. This is a reasonable first step, but should me mentioned in the model description.

We have added this.

> - mostly in response to the second reviewer, the paper now contains a much more detailed description of the manganese observations along the dutch Geotraces section GA02, including a discussion of the methods for these observations.

Yes, indeed that was our intention based on the previous reports.

> The inclusion of a biological cycling of Mn in the model is reasonable, and I think it strengthens the paper a lot. I have, however, a bit mixed feeling about the new focus of the manuscript on the GA02 section: Reviewer 2 suggested "showcasing these (unpublished?) measurements in more depth early in the manuscript would help build a sense of expectations with regard to what the model is expected to do or not to do. That could also be used as a 'roadmap' to explain how the paper is organized and why." To me this aim has not yet been reached fully, the observational results (3.1) and the modelling results, especially section 3.2 still stand side by side in a too unconnected manner.

We modified the text in order to improve the integration of model and data. In line with the recommendation of 'minor revisions', we kept changes to a minimum.

> One example for this is the elevated value of Mn in the DSOW overflow, which is visible quite clearly in Fig 7 and discussed over a few sentences in section 3.1. In the modelling part, this feature is never mentioned again, and indeed the color scale in Fig. 9 is chosen in such a way that it is not even visible in the observations anymore. My expectation is that the model does not reproduce this feature, and that is not even bad; it probably just highlights that the model is missing sediment resuspension, a locally inportant but probably globally unimportant process. In my opinion, often the most important information in model-data comparisons is where the two do not agree because here one learns about processes.

It does seem to be reproduced to some extend, even with our colour scale. Still, how well the DSOW features are reproduced is not of ultimate interest, because the resolution of our model does not describe the DSOW well.

> This is just one example, but my general impression is that the present manuscript does not integrate the observational and modelling parts enough and thus misses what the second reviewer had in mind when he suggested to focus more on observations. The model-data comparison could be made much more precise, and I would argue that the authors should try to do that in a second revision.

This issue is discussed at the beginning of our response.

> One aspect to improve is the "over-reliance on subjective statements to describe the simulations and on imprecise/loose statements" that was already mentioned by the second reviewer. I think the present manuscript still contains many too colloquial statements, but now also in the description of the observations. A few examples are that "very soon the Mndiss reaches the typical background concentration" when talking about NADW or "slightly elevated concentrations in the subsurface are also observed in ... AAIW .. but once again concentrations reach the typical background concentration". Both statements in section 3.1 would be much more precise with some indication where something happens. Likewise, "for a small part" later on the same page is not a helpful description. When comparing the NoBio with the reference run, it is stated that "NoBio generaly compares better with the observations in the Pacific Ocean especially with the US EPZT transect" Again, it would be helpful if the authors could be a bit more specific: Where is the improvement, and in what aspect? At the surface or at 500m depth?

We have improved a lot of these subjective statements. However, sometimes this was quite difficult for reasons mentioned above.

**Minor comments**

Section 2.2.7: the value of 0.4 'derived by Middag et al. (2013)' should probably be $0.4 \cdot 10^{-3}$.

Corrected.

Section 3.1: "Whereas: on line 13 should probably be "As"

Corrected.

Page 16, line 17: maybe include "hydrothermal" before "forcing field"

Corrected.

Page 20, line 23ff: To me this is one of the most interesting results of the new model runs including biology; but why just state this here without giving any numbers? In my first review I already suggested that it would be informative to have an idea on the relative magnitude of the sinking fluxes of authigenic and ofbiologically incorporated Mn; The authors would make that point much stronger when calculating e.g. the globally integrated fluxes of Mn from Mnox and from the biological compartment, maybe at 100m depth and at some depth deeper in the water column. Just a suggestion..

We have included 100 m flux plots, and discuss them in the paper.

The authors might consider using the same software for plotting observational and modelling results; Fig 7 is made using ODV, Fig 9 using ferret, I believe.

We did consider this, and acknowledge that ODV and Ferret present different-looking plots. We do still use the two different programs, because there are different communities working on this study (modellers and observationalist) who use different tools. Furthermore, possibly, ODV is the best tool for interpolated observational data and Ferret is the best tool to make these kind of model–data comparison plots (see Appendix A for the description). In addition, Ferret handles the irregular ORCA grid. In short, it is a practical consideration why we use different software for plotting.

---

## Author Response (AR3)

**Final response to Editor**

Marco van Hulten et al.

8th February 2017

Dear editor,

Thank you for accepting the manuscript. We have made the necessary changes as proposed.

> Please, correct the following: - Abstract, p.2 l.4: What is meant by "the complete model"?
> Would not "the model" suffice here?

Indeed, this was sufficient. We have removed "complete" from both occurences of this phrase in the manuscript.

> - You use now "veracious/veracity" for "accurate/accuracy". I think this makes things worse as "veracious/veracity" is not used in our field in this context. Please, choose another wording. I suggest to use "adequate", "reasonably accurate", "as close as possible to the observations", "in a quantitatively appropriate way", or something like this. Please, replace "veracious-(ly)/veracity" by something else wherever it occurs in the text.

We have followed this suggestion. It is also still clear to what type of "accuracy" we refer to.

> - P.14 l.5: Replace "we will mainly use" by "we mainly use" (use correct tense as in rest of paragraph). - P.19 l.28: Perhaps replace "properties" by "features"?

We have applied those two minor improvements.

Thank you again.

On behalf of the authors, sincerely,

Marco van Hulten